# Influenza A virus rapidly adapts particle shape to environmental pressures

Edward A. Partlow[1], Anna Jaeggi-Wong[1], Steven D. Planitzer[1,2], Nick Berg[1,3], Zhenyu Li[4] & Tijana Ivanovic [1]✉

Enveloped viruses such as influenza A virus (IAV) often produce a mixture of virion shapes, ranging from 100 nm spheres to micron-long filaments. Spherical virions use fewer resources, while filamentous virions resist cell-entry pressures such as antibodies. While shape changes are believed to require genetic adaptation, the mechanisms of how viral mutations alter shape remain unclear. Here we find that IAV dynamically adjusts its shape distribution in response to environmental pressures. We developed a quantitative flow virometry assay to measure the shape of viral particles under various infection conditions (such as multiplicity, replication inhibition and antibody treatment) while using different combinations of IAV strains and cell lines. We show that IAV rapidly tunes its shape distribution towards spheres under optimal conditions but favours filaments under attenuation. Our work demonstrates that this phenotypic flexibility allows IAV to rapidly respond to environmental pressures in a way that provides dynamic adaptation potential in changing surroundings.

Mixed-shape strategy (pleomorphy) is employed by many enveloped viruses despite extreme diversity in structure, composition and entry strategy. Many pleomorphic viruses, such as IAV, respiratory syncytial virus[1], measles virus[2], Ebola virus[3], Nipah virus and Hendra virus[4] pose great threats to human health. IAV, which is responsible for periodic pandemics, produces virions ranging from small, 100 nm spheres to filaments with lengths of several microns. Building larger virions requires greater cellular resources, thus it remains an enigma why filaments are a conserved shape found among diverse virus families. For IAV, filament formation is favoured in animal infection, but not in tissue culture[5–8], suggesting that shape is tuned to extrinsic host pressures. However, the mechanisms by which influenza adopts its shape are not clear. Studying virus filaments is challenging because structural approaches require extensively processed and concentrated samples. Importantly, our work has shown that filaments can resist inactivation by antibodies and withstand other cell-entry pressures[9], highlighting the importance of understanding virion shape regulation.

Previous work on the assembly of influenza filaments has centred around genetic determinants of shape[10–17]. Mutations in the *M1* gene,

coding for the matrix protein 1 (M1), have been found to alter the morphology of assembled virions[10–13]. M1 coats the lumen of virions, and electron microscopy of influenza shows that while it appears disordered in spheres, it assembles into an ordered helix in filaments[14,15]. Mutations affecting shape have also been reported in *NA* and *NP* genes coding for neuraminidase (NA)[16] and nucleoprotein (NP)[17], respectively. This suggested that other viral proteins and ribonucleoprotein (RNP) packaging are involved in determining virion shape[18]. Overall, the mechanisms of how viral mutations alter shape remain unexplained.

Here we reveal a previously unrecognized ability of IAV to dynamically tune assembly to favour either spheres or filaments depending on infection conditions. We perform highly sensitive flow virometry on minimally processed infection supernatants intractable to measurement by existing methods. We find that the shape distribution of actively assembling IAV rapidly adjusts in response to conditions that reduce infection efficiency, such as virus–host incompatibility or the presence of antiviral antibodies. Indeed, antibody effects can be measured within minutes of binding to an infected cell. This ability to adapt in real-time to replication inefficiencies or attenuation may

[1]Laboratory of Viral Diseases, National Institute of Allergy and Infectious Diseases, National Institutes of Health, Bethesda, MD, USA. [2]Martin A. Fisher School of Physics, Brandeis University, Waltham, MA, USA. [3]Department of Biochemistry, Brandeis University, Waltham, MA, USA. [4]Center for Virology and Vaccine Research, Harvard University, Boston, MA, USA. ✉e-mail: tijana.ivanovic@nih.gov

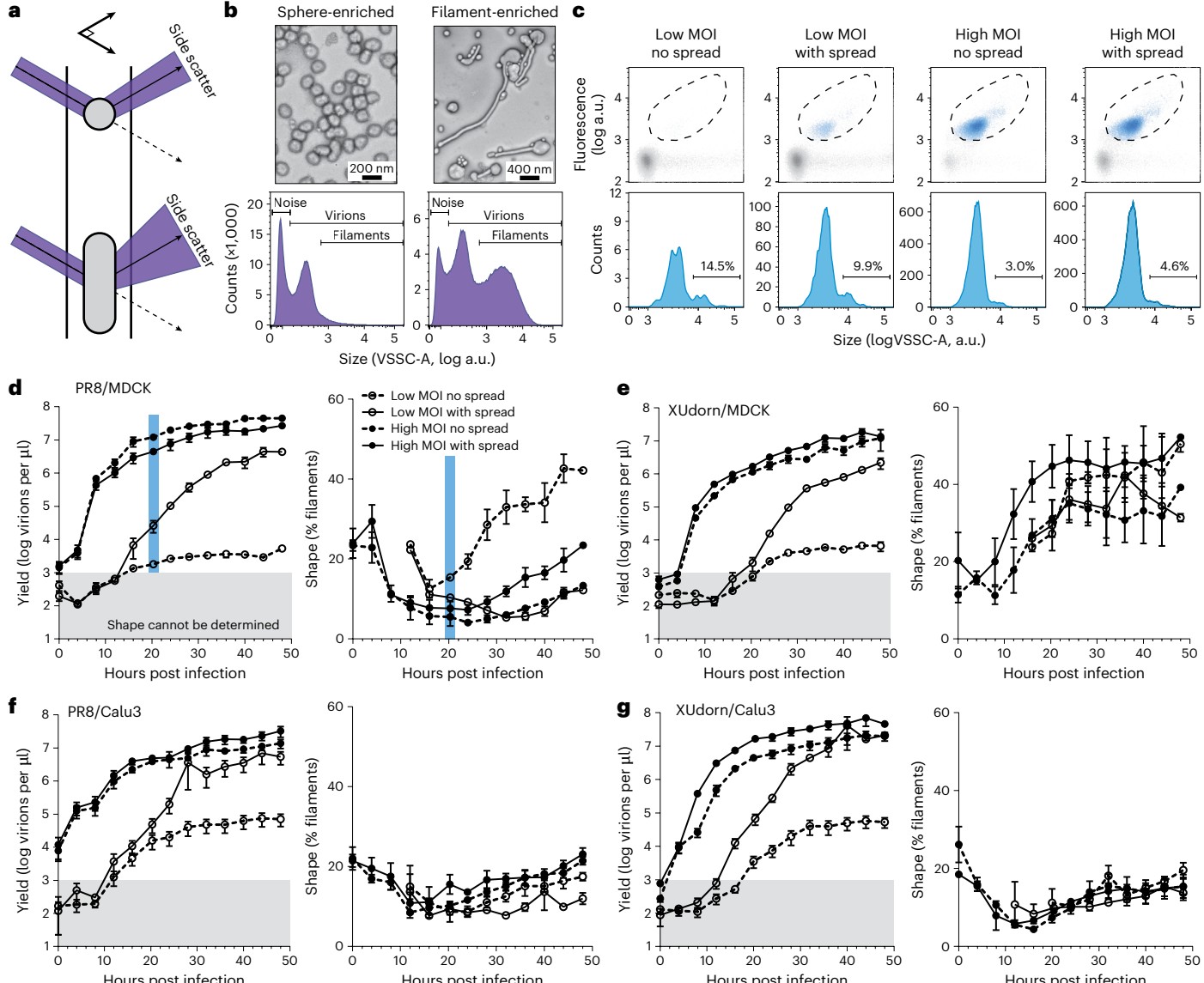

**Fig. 1 | IAV shape distribution is dynamic. a**, A schematic illustrating the VSSC of virions of varying sizes. Large virions such as filaments scatter violet light more than smaller ones such as spheres. **b**, Top: representative electron micrographs of IAV virion samples sphere-enriched or filament-enriched by sucrose gradient ultracentrifugation; five independent samples containing a range of shape profiles were analysed. Bottom: corresponding VSSC histograms of the above samples. Violet side scatter of virions is mostly resolved from noise inherent to small-particle flow cytometry. **c**, Representative flow virometry data for unpurified infection supernatants. To distinguish highly dilute virus signal from noise, virions are bound with a fluorescently labelled HA antibody (Sb H36-26) to enable separation in two detection channels. Displayed are scatterplots of VSSC versus fluorescence with gates enclosing virions (dashed area, top), along with

corresponding VSSC histograms of the gated population (bottom). Percentage of filamentous virions for each sample are indicated. **d**, Time courses of virion yields (left) and shape (right), determined by flow virometry of supernatants from MDCK cells infected with PR8 at MOI 0.006 (Low MOI, open symbols) or MOI 6 (High MOI, filled symbols). No-spread infections (dashed lines) were achieved by omitting trypsin. Mean and s.e.m. are plotted. The blue bars highlight the samples with raw data shown in **c**. Grey shading indicates yields too low to reliably measure shape. **e–g**, Same as **d**, but for XUdorn infections in MDCK cells (**e**), PR8 infections in Calu3 cells (**f**) and XUdorn infections in Calu3 cells (**g**). For Calu3 infections, no-spread infections were achieved by adding ammonium chloride at 4 h post infection (h.p.i.). **d–g**, Data plotted are from three biological replicates.

inform how viruses persist in the population, evade the immune system and acquire adaptive mutations.

## Results

### IAV shape distribution is dynamic

Determining shape of virions produced under attenuating conditions requires a high-sensitivity approach. Here, 'virion' refers to any virus particle regardless of its infectivity. We employed flow virometry to measure the violet side scatter (VSSC) of virions to determine both overall virion concentrations and the relative fraction of filaments (Fig. 1a).

We used diluted samples of concentrated purified virions to enable correlation with electron microscopy (EM) (Fig. 1b and Extended Data Fig. 1a–c). IAV virions produced detectable scatter over inherent instrument background, and filamentous virions scattered violet light more strongly than spherical ones. Samples enriched for filaments revealed two subpopulations in VSSC (Fig. 1b and Extended Data Fig. 1b). The population with greater VSSC correlated with counts of filamentous virions longer than 130 nm derived from EM (Extended Data Fig. 1c). Furthermore, virion counts from flow virometry correlated with haemagglutination (HA) assays (Extended Data Fig. 1d) (HA unit (HAU) is

proportional to virion counts and independent of virion shape[9]). In sum, established assays—EM and HA assay—validated this novel approach for obtaining fast shape and yield measurements of IAV virions.

To extend the sensitivity of flow virometry to dilute samples, including attenuated infections, we used a fluorescent haemagglutinin (HA) antibody that separates virions from noise (Fig. 1c). Using this approach, we were able to determine count and shape for samples as dilute as 1,000 virions per µl. This concentration is typical for majorly attenuated infections in cell culture and ~100,000 times lower than required for EM[9].

We used flow virometry to measure virion yield and shape over time for infections by two IAV strains, A/Puerto Rico/8/34 (PR8) and a variant of A/Udorn/72 containing the *HA*-gene segment of A/Aichi/2/1968-X31 (XUdorn) (Fig. 1d–g). PR8 is reported to be spherical, while XUdorn is reported to be filamentous[19,20]. We performed the time-course analysis in two cell lines, Madin-Darby canine kidney (MDCK)-2,6-sialtransferase (SIAT1) cells and Calu3 cells. We infected cells at high and low multiplicities of infection (MOIs), MOI 6 and 0.006 infectious units (IUs) per cell, to either infect most cells by multiple infectious units, or to ensure that each infected cell receives at most one infectious unit. We additionally compared infections where we allowed or prevented spread of infections (Methods and Extended Data Fig. 2).

The shape distribution of produced virions was dynamic over the course of infection at both MOIs. Infections generally produced a somewhat filamentous population first, followed by a period dominated by spherical virion production, with an eventual increase in filament production. Surprisingly for PR8, low-MOI infections displayed only a transient dip in filamentous virion production, which was followed by a large increase in the proportion of filaments (Fig. 1d). While XUdorn produced an expected high proportion of filaments in MDCK cells[9,20], even XUdorn infections produced spherical populations early in infection (Fig. 1e). Moreover, unlike in MDCK cells, both PR8 and XUdorn were predominantly spherical in Calu3 cells (Fig. 1f,g). These combined data demonstrate that the relationship between viral genotype and shape is not strictly defined but is influenced by the cellular environment of infection. The throughput and sensitivity provided by flow virometry thus revealed an overlooked feature of IAV shape dynamics and set the stage for defining the contribution of the cellular environment to virion shape.

## Infection efficiency influences virion shape

To explore the relationship between the cellular environment and shape, we expanded the analysis of shape for low and high-MOI infections to a panel of four virus strains: PR8, XUdorn, A/Hong Kong/1968 H3N2 (HK68) and A/California/07/2009 H1N1 (Cal0709), in four cell lines: MDCK, Calu3, A549 and Caco2 (Extended Data Fig. 3a–d). The low MOI was 0.006 and the high MOI was 6 IUs per cell except Cal0709, where high MOIs ranged from 1 to 6 IUs per cell due to limited titre. Spread was prevented in all infections.

We explored multiple factors that could influence virion shape and found that while cell line and viral strain biased the shape of the produced virions, no single factor was decisive, and infections produced a range of shapes (Fig. 2a,b). Calu3 cells generally favoured production of spherical virions and MDCK cells tended to produce more filaments, but the cell line was not a lone determinant of shape (Fig. 2a). Similarly, all virus strains could produce either spherical or filamentous populations (Fig. 2b). We found weak correlation between the infectivity of each virus in each cell line and the resulting virion shape (Fig. 2c), but none between virion shape and the total virion yield (Fig. 2d). Increasing MOI often promoted spherical virion production measured at 24 h post infection (h.p.i.), but this was not universal (Fig. 2e and Extended Data Fig. 4). We measured viral genome replication in cells using quantitative PCR with reverse transcription (RT–qPCR), but replication rates did not correlate with shape (Fig. 2f and Extended Data Fig. 3e–h). Remarkably, when we calculated the efficiency of infection

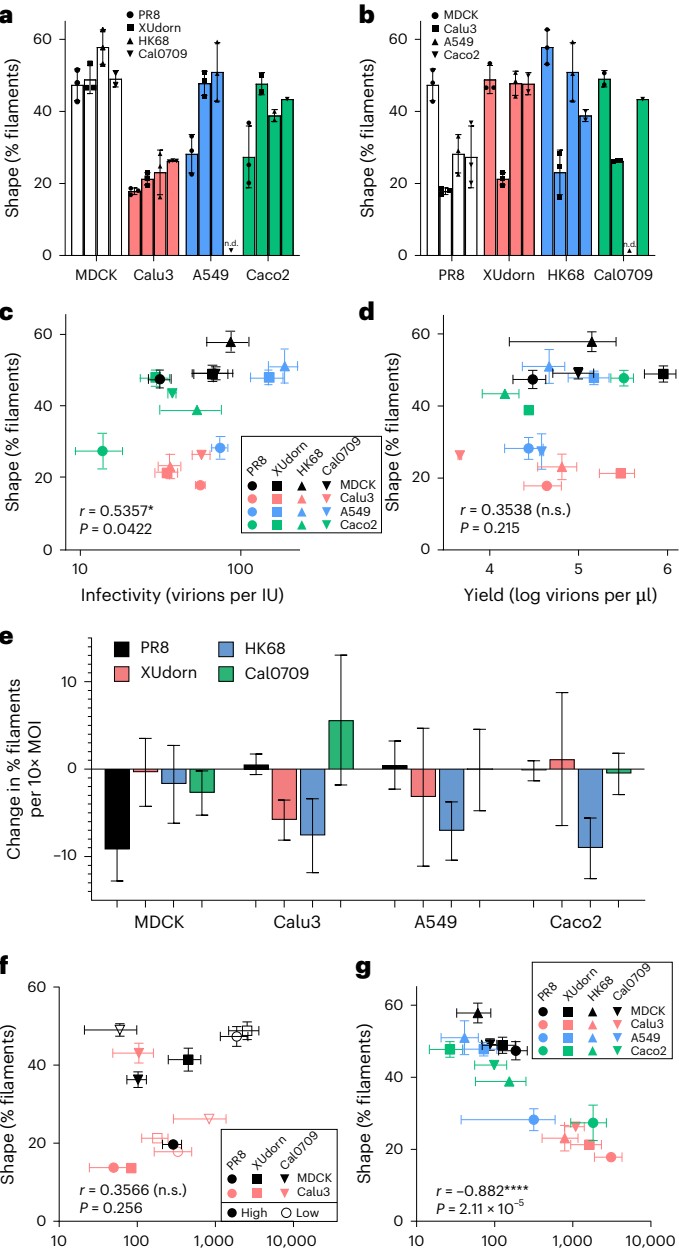

**Fig. 2 | Infection efficiency predicts virion shape. a**, Virion shapes determined by flow virometry at 50 h.p.i. for low MOI (0.006) infections, coloured by cell line. IAV strains are indicated by symbols. n.d., no data. **b**, The data plotted in **a** but coloured by viral strain. Cell lines are indicated by symbols. **c**, Virion shapes for the samples in **a**, plotted against the infectivity (virions per IU) for the given virus–cell combination. **d**, Virion shapes for the samples in **a**, plotted against virion yield produced in the infection, determined by flow virometry. **e**, The four indicated virus strains were used to infect the four indicated cell lines at a range of MOI (0.016–2). Plotted is the average change in shape distribution for a 10-fold increase in MOI, calculated from log-linear regression analysis, derived from 15–48 data points. Error bars represent the 95% confidence intervals of the regression. **f**, RT–qPCR for viral genomes (NP segment) were performed on cells at 0, 4, 20 and 21 h.p.i. Plotted is the genome replication versus shape of virions produced by these infections, where average genome replication was calculated as the ratio of genomes at 20.5 h.p.i. (average of values at 20 and 21 h.p.i.) to those at $t_0$. **g**, Efficiency versus shape plots at low MOI. For all panels, plotted data are mean and s.e.m. of three biological replicates. Spread was prevented in all shown infections. **c,d,f,g**, *r* and *P* values shown are from two-tailed Spearman's correlation analyses. n.s., not significant, *P* > 0.05; \**P* < 0.05 and \*\*\*\**P* < 0.0001 by Spearman's correlation.

at low MOI, defined as the ratio of virions produced to virions initially added, we observed a strong correlation with the shape of produced virions (Fig. 2g). This correlation persisted at high MOI but was less distinct, probably due to complementing or antagonistic interactions between co-entering genomes depending on the cell–virus combination[21] (Extended Data Fig. 3i). Thus, cell permissiveness to infection appears to be a robust inducer of filaments conserved across decades of influenza evolution.

### Attenuation after attachment drives filament assembly

To define the stage of infection at which attenuation increases the proportion of filaments, we titrated inhibitors of distinct early infection steps to achieve the same range of virion outputs and compared their effects on shape at 24 h.p.i. (Fig. 3a–e). Spread was prevented in these infections. The antibody Sb H36-26, included from 0–4 h.p.i., inhibits PR8 virion attachment; the antibody MEDI8852, included from 0–4 h.p.i., allows attachment and internalization but inhibits endosomal membrane fusion; and the small molecule baloxavir, included throughout infection, allows entry and fusion but inhibits the viral polymerase. When maintaining constant virion input, all treatments increased the proportion of filaments (Fig. 3e). The two treatments that permit virion endocytosis, MEDI8852 and baloxavir, induced filaments similarly to each other and to a stronger extent than inhibition of attachment with Sb H36-26. This showed that preventing infection before attachment, analogous to decreasing MOI (Fig. 2e), affects virion shape less than attenuating infection after attachment.

The stronger induction of filaments by inhibiting steps after attachment suggests that cellular sensing of impending or active infection, relative to assembled virion output, may contribute to apparent efficiency. We next altered the balance between entering and progeny virions by infecting MDCK cells with PR8 at either a high or a low MOI (10 or 0.3) and equalizing virion output using baloxavir (Fig. 3f,g). Spread was prevented in these infections. Indeed, the attenuated high MOI infection (lower virion output-to-input ratio) produced a more filamentous population compared with the low MOI infection (higher virion output-to-input ratio) (Fig. 3g). This is an opposite result to the control, unattenuated infections, where high MOI results in a greater proportion of spheres. Notably, in unattenuated infections, the high MOI case has a higher virion output-to-input ratio, supporting the notion that apparent efficiency results from the balance between impending or early and successful infection. We confirmed these findings in Calu3 cells with both PR8 and XUdorn strains (Extended Data Fig. 5).

### Antibodies added after entry potently induce filaments

Our antibody inhibition experiments suggest that influenza shape responds to attenuation, including antibody pressure, early in infection. To identify any generalizable trends between attenuation and shape, we extended antibody treatments to include a panel of 12 monoclonal antibodies under 3 incubation conditions. Antibody was present only during attachment and entry until 4 h.p.i. (PRE); only after entry, starting at 4 h.p.i. (POST); or throughout the infection (ALL). We measured shape at 24 h.p.i. in both MDCK (Fig. 4a and Extended Data Fig. 6) and Calu3 cells (Fig. 4b and Extended Data Fig. 7a). We additionally performed analogous experiments with XUdorn in Calu3 cells using an overlapping set of 9 antibodies as permitted by strain antigenic differences (Fig. 4c and Extended Data Fig. 8). To mimic natural infections, spread was not prevented in these experiments. We also included the modern strain, Cal0709 using a single HA antibody added at 3 h.p.i. in embryonated chicken eggs (Extended Data Fig. 7b). Eggs were chosen because egg-grown Cal0709 virus was spherical enough to assay for changes in morphology.

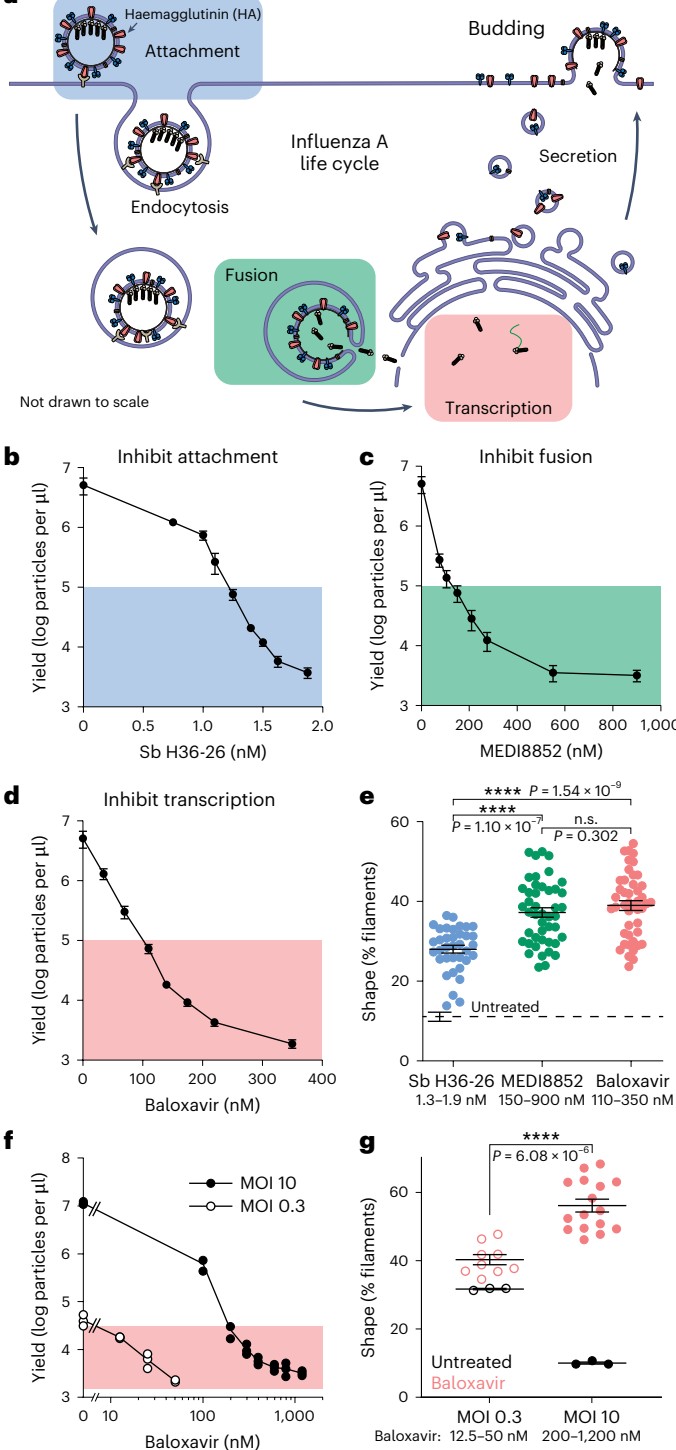

**Fig. 3 | Attenuation of infection past attachment drives downstream filament assembly. a**, Schematic of the influenza life cycle. Stages targeted by antibodies (Attachment and Fusion) or inhibitors (Transcription) are shaded. **b**, Virion yields for MOI 3 infections of MDCK cells with PR8, performed with varying concentrations of Sb H36-26 used to pre-incubate input virus and then also included until 4 h.p.i. **c**, As in **b**, but with MEDI8852. **d**, As in **b**, but with baloxavir added after virion attachment and present throughout infection. **b–d**, Plotted are mean and s.e.m. **e**, Virion shapes for a subset of points in **b–d**, as indicated by the shaded boxes and shown with corresponding colours. Points selected are for overlapping yield regimes. **f**, Virion yields for MOI 10 (filled circle) and MOI 0.3 (open circle) infections of MDCK cells with PR8 at various baloxavir concentrations. Red box indicates region of overlapping yields. Plotted are mean and s.e.m. **g**, Shape of virions produced in **f** for either baloxavir-treated (red) or untreated (black) samples. Spread was prevented in all shown infections. Plotted are mean and s.e.m. **b–g**, Data plotted are from three biological replicates. ****P < 0.0001 using two-sided unpaired t-test.

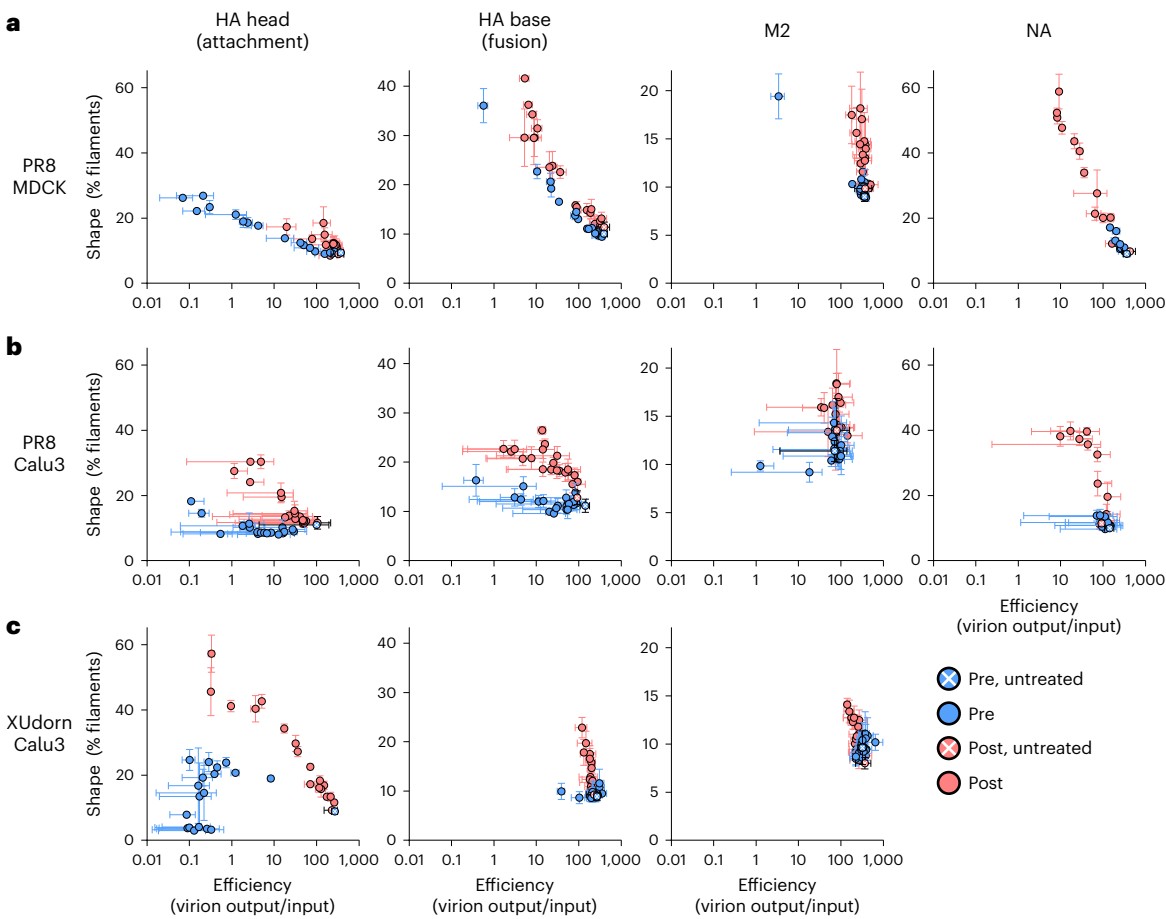

**Fig. 4 | Antibody pressure drives filament assembly. a**, Plots of virion efficiency (virions produced/virions input) versus shape, measured by flow virometry of supernatants from MDCK cells at 24 h post-PR8 infection at MOI 0.6, treated with antibodies at a range of concentrations targeting the labelled antigenic site (see Extended Data Figs. 6–8 for individual inhibitor titration curves). Blue: antibody present until 4 h.p.i. to test entry effects. Red: antibody added at 4 h.p.i. to test assembly effects. Plotted are mean and s.e.m. **b**, As in **a**, but for PR8 virus in Calu3 cells. **c**, As in **a**, but for the H3N2 virus XUdorn in Calu3 cells. Trypsin was not included in infections, which fully prevents spread in **a** and mostly, but not completely in **b** and **c** (Extended Data Fig. 2). **a–c**, Data plotted are from three biological replicates.

The combined antibody-sweep data corroborated our earlier conclusions and new patterns emerged. For pretreatment, inhibition of attachment most strongly yielded filaments for PR8 MDCK infections, in agreement with our MOI data (Fig. 2e). Inhibition of fusion during entry induced filaments more strongly than attachment, and the effect was universal and proportional to attenuation of infection. This supports the notion that attenuation of early infection steps after virion attachment and before genome amplification, including by inhibiting membrane fusion, is a correlate of efficiency.

Adding antibodies after completion of entry universally and potently induced filaments (Fig. 4, Post). The magnitude of these effects (that is, change in shape as a function of virion release) was stronger than the effects of pretreatments, and different for antibodies targeting different viral components. For example, NA antibodies induced shape changes more strongly than HA antibodies for the same extent of attenuation, revealing that induction of filaments late in infection is uncoupled in magnitude from inhibition of particle release. Nonetheless, the late effects of anti-viral antibodies on shape also appear coupled to changes in efficiency. Importantly, the antibody-induced filaments retained their infectivity (Extended Data Fig. 9a), had comparable genome content to untreated virus (Extended Data Fig. 9b), and were not antibody-induced clumps of spheres[22] (Extended Data Fig. 9c,d). We also included a control antibody targeting a host transmembrane MHC class 1 protein, but saw no shape changes, suggesting that effects are specific to viral proteins (Extended Data Fig. 7c). In conclusion, antiviral antibodies are potent inducers of filaments even after virion entry is complete.

## Antibody-induced shape dynamics are fast

To explore the mechanism of antibody-induced shape changes, we performed a short time-course of shape measurements after addition of antibody to an already established infection. We chose to start the antibody treatment at 20 h post infection of MDCK cells with PR8 virus because this is when shape changes are minimal in our time courses (Fig. 1d). At the time of treatment, cells were washed to remove any already produced virus, and given either fresh media or media with various concentrations of an HA antibody. As expected, virion shape was stable in untreated media, slowly increasing from 7.5% to 10.1% filaments over the 4 h period (Fig. 5a, dotted line), while yield steadily increased (Extended Data Fig. 9e). In contrast, antibody treatment elicited a rapid and dose-dependent increase in the fraction of filaments (Fig. 5a). In a control experiment where fluorescently labelled PR8 was added along with antibody treatment, no change to added virion count or shape was observed (Extended Data Fig. 9f,g), demonstrating that measured shape changes (Fig. 5a) are due to newly produced filaments and not aggregation or adsorption of previously produced spheres.

To test whether fast antibody effects are generalizable to other strains and cell lines, we repeated these short-term antibody treatments

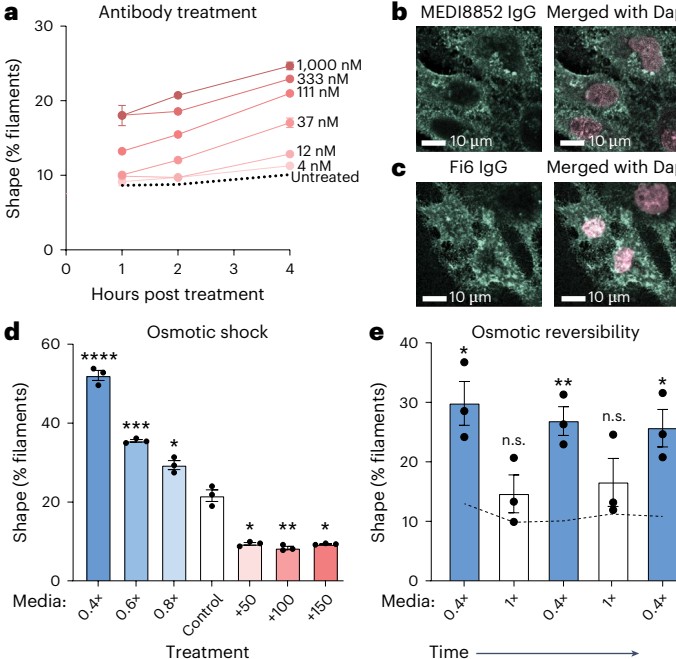

**Fig. 5 | Shape dynamics are fast. a**, MDCK cells infected with PR8 at MOI 0.6 for 20 h were washed into media containing varying concentrations of MEDI8852 antibody. Virion shapes at 1, 2 or 4 h following antibody treatment. **b**, MDCK cells infected with PR8 at MOI 0.6 were treated with MEDI8852 antibody at 21 h.p.i. Two hours after treatment, cells were fixed and permeabilized. MEDI8852 was stained with an AF647-labelled secondary antibody and nuclei were stained with Dapi (grey). **c**, As in **b**, but at MOI 0.3 with directly conjugated AF488-Fi6 IgG treatment at 22 h.p.i. Panels **b** and **c** are representative images of antibody internalization that has been observed in four biologically independent experiments. **d**, MDCK cells infected with PR8 at MOI 0.3 for 20 h were washed into media of varying osmotic strengths. Plotted are virion shapes at 30 min post media change. *P* values: 0.4×, 8.59 × 10⁻⁵; 0.6×, 7.60 × 10⁻⁴; 0.8×, 0.0138; +50, 1.24 × 10⁻³; +100, 9.60 × 10⁻⁴; and +150, 1.12 × 10⁻³. **e**, As in **d**, but just for 0.4× media. In addition, samples were alternated between treatment and untreated every 30 min. Plotted are virion shapes for each 30-min interval. *P* values from left to right: 0.0109, 0.468, 5.87 × 10⁻³, 0.402 and 0.0258. **a**,**d**,**e**, Plotted are mean and s.e.m. of three biological replicates. \**P* < 0.05, \*\**P* < 0.01, \*\*\**P* < 0.001 and \*\*\*\**P* < 0.0001 using unpaired *t*-test relative to control.

with PR8 in A549 and Caco2 cells, as well as with HK68 in Calu3 cells (Extended Data Fig. 10a–c). In all cases, antibody-induced filament production was observed within 2 h of treatment. These results highlight another conserved behaviour of IAV rapidly responding to antibody binding to infected cells by producing filaments.

The rapid nature of antibody-induced changes in viral assembly could implicate a biophysical effect driving filament assembly. To test this, we investigated whether antibodies targeting viral surface proteins are internalized by infected cells. We assayed the subcellular localization of anti-HA antibodies after transient treatment on MDCK cells infected with PR8. We treated infected cells separately with two HA antibodies for 2 h, then performed immunofluorescence to visualize the treatment antibodies. Indeed, both antibodies were observed in cytosolic and perinuclear puncta, showing that they had been internalized by the cell (Fig. 5b,c). These data suggest that antibodies binding to infected cells may be inducing endocytosis and directly or indirectly influencing viral assembly.

To test whether changes to membrane biophysics could drive shape changes in assembling virions, we subjected PR8-infected MDCK cells to osmotic shock and measured the shape of virions produced in the subsequent 30 min. Hypotonic media induce cell swelling by osmosis, while hypertonic media has the opposite effect. Hypotonic media

generated populations with greater than 50% filaments while hypertonic media suppressed filaments to below 10% (Fig. 5d), and these changes to virion shape were reversible (Fig. 5e and Extended Data Fig. 10k,l) and conserved in other cell lines and virus strains (Extended Data Fig. 10e,f,i,j). When we infected fresh MDCK cells with equivalent numbers of virions released under different media tonicities (3 virions per cell), we found little difference in per-virion infectivity, or the shape of virions produced from these infections, confirming that these are bona fide, infectious virions (Extended Data Fig. 10g,h). While from these results alone we cannot distinguish whether shape changes due to membrane tension, media salinity or effects on virus yield (Extended Data Fig. 10e), these experiments confirm that shape changes can occur quickly and reversibly. The data suggest that transient changes in membrane dynamics or other conditions could affect ongoing assembly.

## Discussion

This work reveals that influenza A virion assembly is dynamic and rapidly tuned to the infection environment rather than a fixed property of a given strain as commonly assumed (Fig. 6a). Assembly of filaments is responsive to the host-cell permissiveness as well as the presence of attenuating external factors including immune pressure from antibodies. Adaptive pleomorphy enables immediate response to a changing environment. This property of IAV infections probably has major consequences for evolvability of pleomorphic populations (Fig. 6b,c).

Filaments offer the greatest adaptive advantage if their production is phenotypically tunable. If shape distributions were purely genetically programmed, the selective survival of filamentous variants would bottleneck the population's genetic diversity, limiting the virus' ability to acquire true escape mutations (Fig. 6b). Further, even if an adaptive mutation was acquired, the virus would remain committed to producing filaments, which could be disadvantageous in lower-pressure environments, until spherical variants could be selected again. In contrast, phenotypic tuning allows the entire population, with all genetic variation, to shift to more beneficial forms, providing the best chance of evading selection pressures[23] (Fig. 6c). Once genetic change allows escape from ongoing pressure, a shift to the more efficient spheres is immediate.

Our findings reveal that shape is a conserved, rather than strain-specific, genetic feature across IAV strains. Sixteen different virus–cell combinations, each with different inherent efficiencies, produced shape profiles that followed predictable trends (Fig. 2g). Weak or lacking correlations between shape and any individual parameter that contributes to efficiency (Fig. 2c,d,f), such as infectivity, yield or replication can be interpreted as different early steps limiting infections for different virus–cell pairs. Despite these likely differences in the underlying mechanisms of reduced efficiency for different strains in different host environments, the overall replicative efficiency is a strong predictor of virion shape across strains. The strains tested here represent two subtypes and 80 years of viral evolution, yet our results show that none of them, even XUdorn, are true shape mutants, highlighting the conserved nature of this adaptive strategy. These insights and fast shape analysis enabled by flow virometry set the stage for reanalysis of previously published genetic shape mutants to distinguish shape changes deriving from changes in efficiency from direct effects on virion assembly and budding.

Antibody binding to cells after infection is already established triggers immediate and strong shape changes relative to early inhibition (Figs. 4 and 5). Fast responses were also observed for transient changes in media osmolarity (Fig. 5d,e). While small changes in yield were observed in those cases (Extended Data Fig. 10d), the produced virions exhibited the same per-particle infectivity (Extended Data Fig. 10g), confirming that genomes and viral proteins were not limiting during unperturbed assembly and suggesting that shape dynamics are regulated at the step of virion release. Indeed, osmotic pressure effects were fully reversible and could be cycled in rapid succession

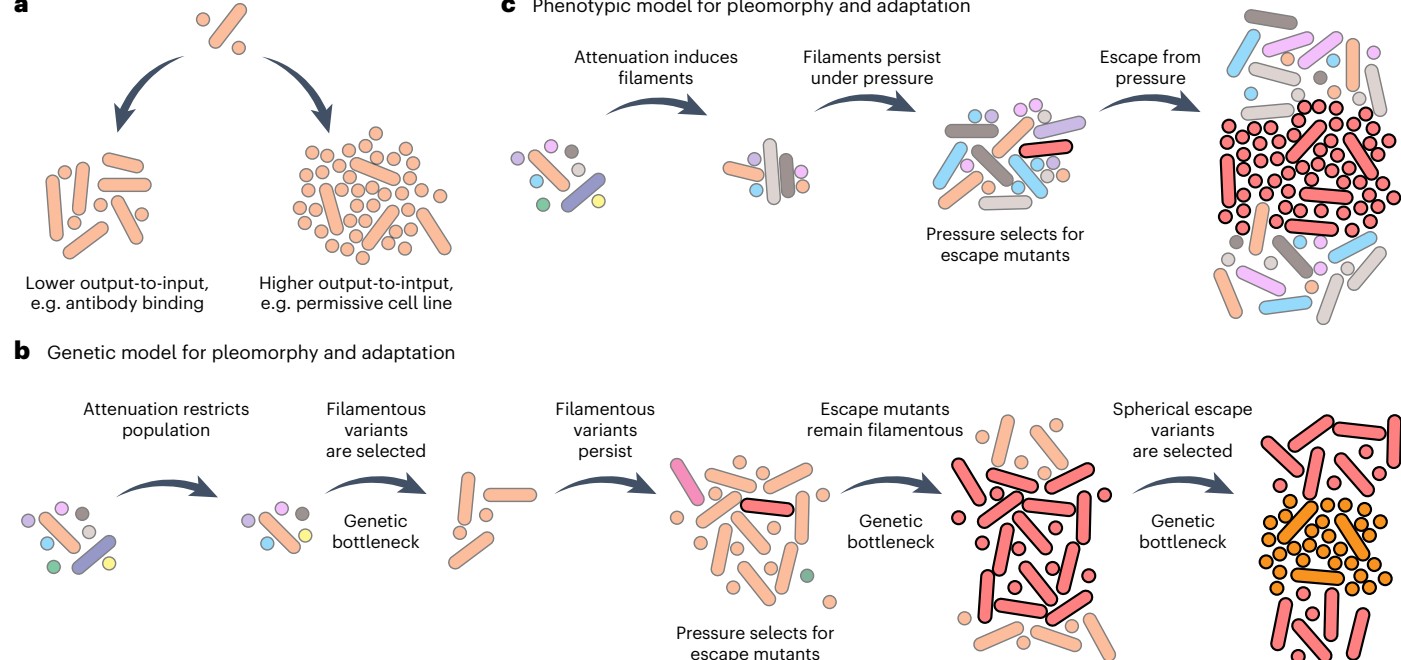

**Fig. 6 | Filaments as a mechanism for viral persistence and adaptation.**
**a**, IAV infections produce either more spherical or more filamentous populations independent of the input-virion shape and according to infection efficiency in each environment. **b**, Schematic of viral adaptation pathway showing if virion shape distributions were solely genetically encoded. A predominantly spherical strain infects an animal cell. Attenuation is permissive to filaments and selects for filamentous variants. Filaments persist, possibly adapting to overcome the attenuating pressure. In the absence of pressure, spheres have a fitness advantage and spherical variants are selected for. **c**, Model illustrating how dynamic virion shape distribution may promote viral persistence. A predominantly spherical population infects an animal cell. Attenuation induces filament production and any genetic variant has an equal probability of being enclosed in a filament, and thus population diversity is solely constrained by the extent of attenuation. These filaments persist under attenuating pressure, allowing further replicative cycles and possibly generating variants containing genetic changes that enable adaptation or immune evasion. Upon escaping attenuating pressure, the population immediately reverts to spheres without loss of genetic diversity.

(Fig. 5e and Extended Data Fig. 10k,l). Our preliminary experiments with transient antibody treatments suggest that antibody binding to infected cells could be inducing endocytosis and, in this way, affecting membrane dynamics (Fig. 5b,c). Importantly, our current experiments do not establish a causal link between antibody internalization and shape change. Moreover, we do not differentiate between a direct effect of antibody binding on membrane dynamics and consequently virion shape, and an indirect effect via changes in infection efficiency. The mechanism of antibody shape effects will be defined in future experiments.

In summary, pleomorphy in IAV is a conserved, tunable feature that provides an evolutionary advantage under immune and entry challenges. This plasticity enables IAV to rapidly respond to immune pressures or new hosts, facilitating both persistence of adapted strains and new adaptation after spillover.

## Methods
### Reagents
**Cells.** Madin-Darby canine kidney (MDCK)-2,6-sialtransferase (SIAT1) cells (Sigma-Aldrich, 05071502), MDCK.2 cells (ATCC strain CCL-34) and human embryonic kidney (HEK)293T cells (ATCC) were propagated in DMEM medium (Cytiva) supplemented with 10% FBS (Atlas Biologicals). Calu3 cells (ATCC), A549 cells (ATCC) and Caco2 (ATCC) cells were propagated in EMEM medium (Wisent) supplemented with 20% FBS. HEK293F cells, a gift from S. C. Harrison (Harvard Medical School), were propagated in FreeStyle 293 Expression Medium (Thermo Fisher) and were used for the expression of HC19, MEDI8852 and 14C2 antibodies.

**Antibodies.** Purified Fi6, CR9144, O19, W6/32, T2-5D and T2-7D antibodies, and hybridomas producing anti-influenza virus HA antibodies Sb H36-26, Sa Y8-1A6-6, Ca H2-4B1-14 and Cb H9-D3-4R2 and NA antibodies NA2-1c1 and NA2-10e10 were gifts from Jonathan Yewdell (NIH/NIAID/LVD/CBS). Hybridoma producing anti-influenza virus M2 monoclonal antibody 65 was a gift from Xavier Saelens (VIB-UGent Center for Medical Biotechnology). Hybridoma producing anti-influenza virus NP monoclonal antibody HB65 was obtained from ATCC (H16-L10-4R5, 58696953). The expression vectors (modified pVRC8400) for HC19 and MEDI8852 IgG heavy and light chains were a gift from S. C. Harrison (Harvard Medical School). The sequence for F045 was previously described[24]. For 14C2 IgG, the variable heavy and variable light chain sequences from a single-chain variable fragment[25] were incorporated into a mouse IgG expression plasmid. Expressed antibodies were produced by transient transfection of HEK293F cells with polyethylenimine (0.4 μg heavy chain DNA, 0.6 μg light chain DNA, 1.5 μg polyethylenimine, per $1 \times 10^6$ cells) and collected from the cell culture supernatant at 4–7 days post transfection. Antibodies from hybridomas, HC19, MEDI8852 and 14C2 IgG were purified using protein G resin (Cytiva). TriHSB.2 plasmid was a gift from David Baker (University of Washington). TriHSB.2 was expressed by autoinduction in bacteria[26]. F045 and TriHSB.2 constructs contain C-terminal 6×His tags and were purified by passage over a HisTrapFF column (GE Healthcare), followed by size exclusion chromatography using a Superdex 200 increase 10/300 column (Cytiva). Antibodies were stored in PBS or PBS containing 10% glycerol.

**Antibody labelling.** Lyophilised DyLight550 (DL550, Thermo Fisher), JaneliaFluor646 (JF646, Janelia) or AlexaFluor488 (AF488, Thermo Fisher) NHS ester dyes were resuspended in anhydrous dimethylsulfoxide (Sigma-Aldrich). Labelling was performed with a 100:1 (HC19, Sb H36-26, T2-5D, or T2-7D:DL550 or Fi6:AF488) or 20:1 (HB65:AF488,

DL550 or JF646) molar ratio of dye to antibody in 100 mM sodium bicarbonate buffer for 1 h at room temperature, followed by passage over a Macro SpinColumn packed with G25 packing material (Harvard Apparatus) in PBS. Labelled antibody concentrations were calculated using absorbance at 280 nm and 493 or 557 nm.

**Primers.** *NP*-gene segment primers used were oNB23: TACTGGGCC ATAAGGACCA and oNB24: TCCTCTGCATTGTCTCCGA. 7sk snoRNA primers used were oEP1591: CTGGCTGCGACATCTGTCA and oEP1592: GGAGGTTCTAGCAGGGGA.

## Viruses

A/Puerto Rico/8/1934 (PR8) wild-type influenza virus and A/Udorn/ 307/1972 containing the A/Aichi/1968 (X31) HA segment (XUdorn)[20] were passaged in Calu3 cells at MOI 0.002 with 1 µg ml$^{-1}$ TPCK-trypsin (Sigma-Aldrich) in OptiMEM (Thermo Fisher). The *HA*- and *NA*-gene segments were sequenced to ensure correctness in stock viruses. Viruses used here were passaged once from these stocks and behaviour was indistinguishable in our assays. Viruses were titred by haemag-glutination (HA), infectivity assay and flow virometry. HA assays were performed with turkey red blood cells. Viruses A/Hong Kong/1/1968 (HK68) and A/California/07/2009 (Cal0709) were provided by Jonathan Yewdell (NIH/NIAID/LVD/CBS). These were derived from chicken eggs and used without propagation. Work as described with IAV strains A/Puerto Rico/8/1934, A/Udorn/307/1972 containing the A/Aichi/1968 (X31) *HA*-gene segment, A/Hong Kong/1/1968 and A/California/ 07/2009 was approved by the Institutional Biosafety Committee (IBC) at the Biosafety Level 2 Laboratory (BSL-2; Registration #RD-23-IV-10).

**Infectivity assays.** For each cell line–virus strain combination, infectiv-ity (virions per IU) was calculated from the average result of infections at two MOIs, each performed in triplicate. Cells were infected at virion inputs ranging from 0.001–2 virions per cell (see 'Infections without spread' below). At 24 h.p.i., cells were fixed and permeabilized using 4% paraformaldehyde and 0.1% Triton X-100, and then stained with AF488- or JF646-labelled HB65, which recognizes NP. The number of infected cell singlets was determined by flow cytometry (NP deriving from input virions binding to cells at $t_0$ was not detected under these conditions). Each replicate derived at least 2 inputs with a percent of cells infected of 8 or less, ensuring that infected cells resulted from a single infectious unit.

**Virus purification.** XUdorn viruses were purified through a 20% sucrose cushion (SC), then either by passage over a 20–60% sucrose gradient[20] or by sequential centrifugation: viruses were processed for 8 cycles of centrifugation at 3,250 relative centrifugal force for 1.5 h at 4 °C[9]. Three fractions were collected from the sucrose gradient. 'Spherical' is a distinct band in the sucrose gradient, and 'Filamentous 1' and 'Fila-mentous 2' are virion samples deriving from diffuse-virus regions of the gradient of progressively higher sucrose density. From the sequential centrifugation, additional filament- (filamentous 3 and fil. other) and sphere- (sph. other) enriched fractions were collected from the pellet or the supernatant, respectively (Extended Data Fig. 1).

**Direct (antibody-independent) virus labelling.** Virus was fluores-cently labelled by conjugating viral surface proteins to AF488 via an NHS ester reaction. PR8 virions ($4 × 10^9$) in 0.1 M NaHCO$_3$ were com-bined with 5.6 nmol AF488 NHS ester dye in dimethylsulfoxide and incubated at room temperature for 1 h. Labelled viruses were purified using a PD-10 desalting column. Virus was quantified and confirmed to retain binding of Sb H36-26 IgG and MEDI8852 IgG by flow virometry.

## Electron microscopy

For electron microscopy, 'Filamentous 1', 'Filamentous 2' and 'Filamen-tous 3' were adjusted to $2 × 10^4$ HAU per ml and 'All' and 'Spherical' were

adjusted to $1 × 10^5$ HAU per ml. As a precaution, viruses were pretreated with 25 µM rimantadine (M2 inhibitor, Sigma-Aldrich) for 30 min at room temperature to prevent fragmentation of the filamentous viri-ons[27]. Grids were stained with 2% phosphotungstic acid (pH 7.5–8)[20]. Images were taken using a Philips Morgagni v.3.0 transmission electron microscope (80 kV) using an AMT NanoSprint5 camera. Virion size measurements were performed using custom MATLAB codes[20].

## Time-course experiments

MDCK-SIAT1, Calu3, A549 or Caco2 cells were grown in 24-well plates until they formed confluent monolayers. The cells were washed twice with HBSS solution (Thermo Fisher) before attachment of PR8 or XUdorn virus in OptiMEM at MOI 0.006 or 6 for 1 h at room tempera-ture. For Cal0709, titres could not reach MOI 6, so 35 µl undiluted virus was used for attachment. After attachment, unattached virus was removed, and the cells were washed twice with HBSS. Infection media (300 µl) with or without trypsin and with or without 20 mM ammonium chloride, were added. At 4 h.p.i., ammonium chloride was added to some wells. Samples (10 µl) were taken every 4 or 8 h until 48 h.p.i. Infected-cell supernatant aliquots were analysed by flow virometry. Samples were stored in PCR tubes at −80 °C until analysis.

## Infections in chicken eggs

The allantoic fluid of day-10 SPF premium eggs (AVSbio) was injected with $10^6$ Cal0709 virions. At 24 h.p.i., eggs were moved to 4 °C for at least 2 h. The allantoic fluid was collected, centrifuged at 4,000 $g$ for 10 min to remove debris, and the supernatant was analysed by flow virometry.

## Infections without spread

**General.** Cells were grown in 24-well plates until they formed con-fluent monolayers. After washing twice in HBSS, virus was added in 35 µl OptiMEM and incubated at room temperature for 1 h with fre-quent shaking. After attachment, cells were washed twice with HBSS and 300 µl OptiMEM was added. Trypsin was omitted and 20 mM ammonium chloride was added at 4 h.p.i. to inhibit spread. Infections were incubated at 34 °C with 5% CO$_2$ and 100% humidity. Infected-cell supernatant aliquots were analysed by flow virometry. Samples were analysed freshly or stored in PCR tubes at −80 °C until analysis.

**MOI experiments.** MDCK-SIAT1, Calu3, A549 or Caco2 cells were infected with PR8, XUdorn, HK68 or Cal0709 virus at MOI 0.0156–2, except for Cal0709, where 0.0078–0.979 was used because of limited titre. Infected-cell supernatants were collected at 24 h.p.i.

**Antibody-sweep experiments.** MDCK-SIAT1 cells were used for PR8, and Calu3 cells for PR8 and XUdorn. PR8 or XUdorn virus and antibody dilutions were made in OptiMEM. MEDI8852, Fi6, CR9114, HC19, monoclonalAb65, 14c2 and O19 antibodies were used for both viruses. Antibodies used only for PR8 were Sa Y8-1A6-6, Sb H36-26, Ca H2-4B1-15, Cb H9-D3-4R2, NA2-1c1 and NA2-10e10. XUdorn-specific antibodies were F045 and TriHSB.2. Cells were infected at MOI 0.6. For PRE and ALL, a range of the indicated treatment was included during the room-temperature incubation and first 4 h of infection. At 4 h.p.i., the media were removed and the cells washed with HBSS. New infection media containing untreated OptiMEM for PRE or OptiMEM with treatment for POST and ALL were added to the cells. Infected-cell supernatants were collected at 24 h.p.i. Ammonium chloride was not included in these experiments, but effects of spread were expected to be minimal.

**Inhibitor experiments.** For Fig. 3b–e, MDCK-SIAT1 cells were infected with PR8 virus at MOI 3. When included, Sb H36-26 and MEDI8852 were included during the room-temperature incubation and until 4 h.p.i. At 4 h.p.i., the media were removed and the cells washed with HBSS.

When included, baloxavir marboxil (Medchemexpress) was added to the indicated concentration from an 875 µM stock in cell culture dimethylsulfoxide (Sigma-Aldrich) at the end of the room-temperature incubation and after washing with HBSS at 4 h.p.i. Infected-cell supernatants were collected at 24 h.p.i. For Fig. 3f,g and Extended Data Fig. 5, MDCK-SIAT1 cells were infected with PR8 and Calu3 cells were infected with PR8 or XUdorn at MOI 0.3 or 10. Baloxavir was added at the end of the room-temperature incubation. Infected-cell supernatants were collected at 24 h.p.i.

**Transient antibody treatment experiment.** MDCK-SIAT1 cells were infected with PR8 virus at MOI 0.6. At 20 h.p.i., cells were washed into a range of concentrations of MEDI8852. Samples of the supernatant were collected at 1, 2 and 4 h post treatment with MEDI8852 (Fig. 5a,b) or 2 h post treatment with MEDI8852 (Extended Data Fig. 10a–c). Cal0709 transient antibody experiment (Extended Data Fig. 7b) was performed in eggs (see 'Infections in chicken eggs'). At 3 h.p.i., PBS or antibody was injected into the allantoic fluid. Concentrations were based on an assumed 50 ml per egg.

**Transient antibody genome quantification experiment.** MDCK or Calu3 cells were infected with PR8 or XUdorn at MOI 0.6 (Extended Data Fig. 9b). At 20 h.p.i., cells were washed into 250 nM MEDI8852 and incubated for 2 h. Resulting supernatants were analysed by flow virometry and RT–qPCR (see 'Viral genome quantification' below).

**AF488-virus reuptake control.** MDCK-SIAT1 cells were infected with PR8 virus at MOI 3. At 20 h.p.i., cells were washed into a range of concentrations of MEDI8852 along with a fixed concentration (~4,000 per µl) of AF488-labelled PR8 virus. Samples of the supernatant were collected at 2 h post treatment. We confirmed that the produced virus had the expected shape effects.

**Antibody internalization experiment.** For Fig. 5c, MDCK-SIAT1 cells were infected with PR8 virus at MOI 0.6. At 21 h.p.i., cells were washed into 500 nM MEDI8852 and incubated for 2 h. The cells were then collected, fixed, permeabilized and stained with 300 nM Dapi and 13.3 nM AF647-αMouse IgG secondary antibody. Samples were imaged on a Leica SP8 (690) DMI6000 confocal microscope using an HC PL APO CS2 ×40/1.30 oil objective. Imaging and image processing were performed in LAS X and Fiji, respectively. For Fig. 5d, MDCK-SIAT1 cells were infected with PR8 virus at MOI 0.3. At 22 h.p.i., cells were washed into 300 nM AF488-Fi6 and incubated for 1 h. The cells were then collected, fixed, permeabilized and stained with 300 nM Dapi. Samples were imaged as above.

**Osmolarity experiments.** MDCK-SIAT1 cells were infected with PR8 virus at MOI 0.6 (Fig. 5e) or Calu3 cells with PR8 or XUdorn virus at MOI 0.3 (Extended Data Fig. 8i,j). At 20 h.p.i., cells were washed into either OptiMEM diluted with ultrapure $H_2O$ (0.4×, 0.6×, 0.8×) or supplemented with NaCl (50, 100, 150 mM). Supernatants were collected at 30 min post treatment.

**Osmolarity experiments.** Calu3 cells were infected with HK68, and A549 and Caco2 cells were infected with PR8, at MOI 1 (Extended Data Fig. 8e,f). At 20 h.p.i., cells were washed into either OptiMEM diluted with ultrapure $H_2O$ (0.4×) or supplemented with NaCl (150 mM). Supernatants were collected at 60 min post treatment.

**Osmolarity reversibility experiments.** MDCK-SIAT1 cells were used for PR8, and Calu3 cells for PR8 and XUdorn. Cells were infected with virus at MOI 0.6. At 20.5, 21.5 and 22.5 h.p.i., cells were washed into either OptiMEM diluted with ultrapure $H_2O$ (0.4×, 0.6×, 0.8×) or supplemented with NaCl (50, 100, 150 mM), and the supernatant was collected at 30 min post treatment. At 21 and 22 h.p.i., cells were washed into 1× OptiMEM and the supernatant was collected at 30 min post treatment.

**Reinfection experiments.** MDCK-SIAT1 cells were infected with PR8 supernatants from the osmolarity or antibody treatment experiments diluted to 3 virions per cell in 35 µl. After 1 h attachment, cells were washed into fresh OptiMEM. Samples of the supernatant were collected at 24 h.p.i.

## Clumping experiments

An untreated supernatant from the antibody-sweep experiments (see above) was mixed with antibodies at the highest concentrations used in the antibody experiments. Mixtures were incubated at 34 °C for 1 h and then analysed by flow virometry.

## Flow virometry

DyLight550-labelled Sb H36-26 IgG, HC19 IgG, T2-5D IgG and T2-7D IgG stock solutions were diluted to 11.85 nM, 50 nM, 25 nM and 25 nM, respectively, in 0.2% BSA and HNE20 (20 mM HEPES NaOH pH 7.4, 150 mM NaCl and 0.2 mM EDTA). Infected-cell supernatants were undiluted or diluted up to 1:30 in HNE20 and combined 1:1 with antibody dilution in BSA. Sb H36-26 IgG was used to label PR8, HC19 IgG was used to label XUdorn and HK68, and T2-5D IgG or T2-7D IgG was used to label Cal0709. Binding reactions were incubated at room temperature for 30 min to 1 h, then diluted 1:250 in HNE20. Flow virometry was performed using the CytoFLEX S platform (Beckman Coulter). Laser powers were 70 mW for violet and 50 mW for yellow. Gain values were set to 300 for VSSC and 1,000 for RFP. Samples were triggered on violet side-scatter area (1,000–3,500 a.u. threshold) and RFP (300–500 a.u. threshold) and acquired for 600 s or 25,000 particles in the virion gate. Thresholds were optimized for each instrument. Unlabelled concentrated virus preparations were triggered on VSSC and acquired after 2 min until 500,000 particles in the virion gate (Fig. 1b and Supplementary Fig. 1). Virus samples at variable dilutions (1:10 to 1:40,000) were mixed with FluoSphere (Invitrogen) beads (170 nm, 505/515) at 1:300 and acquired by flow cytometry (Supplementary Fig. 1). All flow virometry samples were prepared in HNE20. Analysis was performed in Cytexpert 2.5 or FlowJo 10.9.0.

## Flow cytometry

Flow cytometry was performed using the CytoFLEX S platform (Beckman Coulter). Laser powers were 70 mW for violet and 50 mW for yellow. Gain values were set to 85–103 for FSC and 92–333 for SSC. For infectivity assays, gain values were set to 100 for FITC when using AF488-HB65 and 565 for APC when using JF646-HB65. For antibody internalization experiments, gain values were set to 3,000 for APC and 50 for RFP. Samples were triggered on FSC and acquired for 50,000 particles in the singlet-cell gate. All flow cytometry samples were prepared in PBS. Analysis was performed in Cytexpert 2.5 or FlowJo 10.9.0.

## Viral genome quantification

Infected-cell supernatants were combined 1:1 with 100 ng ml$^{-1}$ RNase A and incubated for 30 min at 37 °C before addition of 40 U of RNasin ribonuclease inhibitor. RNase-treated supernatants were purified using a New England Biolabs Luna Cell Ready One-Step RT–qPCR kit. Treated samples (1 µl) were added to 9 µl lysis reactions for 10 min at 37 °C, followed by addition of 1 µl stop buffer. Lysate mix (2 µl) was combined with 18 µl qPCR master mix, with primers amplifying a region of the *NP*-gene segment conserved across all our influenza model strains (oNB23 and oNB24). The genome content of experimental samples was determined by comparing cycle threshold ($C_t$) for detection with a standard curve of sucrose cushion-purified virions. The packaged genome per virion was determined by comparing genome titre by qPCR to virion titre by flow virometry.

## Cellular vRNA quantification

**RNA extraction.** Cell pellets containing ~$5 \times 10^5$ cells were resuspended in 500 µl of TRIzol, followed by 5 min incubation at room temperature.

Chloroform (100 µl) was added to each of the lysis mixtures. After 2 min, samples were centrifuged at 12,000 *g* for 15 min. Aqueous phases were collected and mixed with 250 µl of isopropanol. After 10 min at 4 °C, samples were centrifuged for 10 min at 12,000 *g*. Supernatant was removed and RNA was washed in 500 µl of 75% ethanol, followed by 5 min of centrifugation at 7,500 *g*. Supernatant was removed and pellets were air dried before dissolving in 50 µl of water.

**Two-step qPCR.** To amplify the (−) sense genomic vRNA within cells, a two-step RT–qPCR was performed on RNA extractions of infected cells. One microlitre of extracted RNA was combined with RT–qPCR master mix (New England Biolabs Luna Universal RT–qPCR kit). A primer specific to the (−) sense NP strand (oNB23) was added to the reaction for 10 min at 50 °C. Samples were placed on ice and the (+) sense NP primer (oNB24) was added. Samples were then PCR amplified. Viral genome content in each sample was compared to total RNA via a separate qPCR reaction using primers (oEP1591 and oEP1592) against 7sk (a non-mRNA control). All sample $C_t$ values were normalized to a model sample to obtain a relative quantification, with a standard curve constructed by 10-fold serial dilutions of the model sample.

### Statistics and reproducibility
No statistical methods were used to predetermine sample sizes, but our sample sizes are similar to those of ref. 9. The only measurements excluded from analysis were shape measurements of Cal0709 at <20,000 particles per µl or other viruses at <1,000 particles per µl, where noise measured in negative samples reached ~10% of the virion count, skewing shape measurements. Data distribution was assumed to be normal, but this was not formally tested. The experiments were not randomized. The investigators were not blinded to allocation during experiments and outcome assessment. Except for qualitative observations in immunofluorescence (Fig. 5b,c), all analysis was automated and independent of human interpretation. For example, gates in flow virometry were set using a control and applied to all samples.

### Reporting summary
Further information on research design is available in the Nature Portfolio Reporting Summary linked to this article.

## Data availability
The electron micrographs are presented and used for analysis in Fig. 1 and Extended Data Fig. 1, uncropped immunofluorescence images are presented and used for analysis in Fig. 5, and all other data plotted are available as Source data provided with this paper.

## Code availability
Custom Matlab code (R2016a) was used to analyse electron microscopy data, and is available on GitHub at https://github.com/tivanovic/Ivanovic-lab-analysis-codes (ref. 28).

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

## Acknowledgements

We thank J. Yewdell's Lab at the National Institutes of Health, S. Harrison's lab at Harvard University, and X. Saelens at VIB-UGent Center for Medical Biotechnology for providing hybridomas, antibody constructs and/or antibodies; J. Yewdell, A. Lee, G. Hollopeter and I. Kosik for their comments to aid in preparing this paper; and I. Kosik for teaching us how to propagate influenza in chicken eggs. We acknowledge support from the NIH Director's New Innovator Award 1DP2GM128204 (T.I.) and the NSF MRSEC DMR-1420382 (T.I.). This publication is based on research supported by The G. Harold and Leila Y. Mathers Charitable Foundation (T.I.). Funding for this study was in part provided by the Divisions of Intramural Research of the National Institute of Allergy and Infectious Diseases (E.A.P., A.J.-W., S.D.P., N.B., T.I.). The funders had no role in study design, data collection and analysis, decision to publish or preparation of the manuscript. The content of this publication does not necessarily reflect the views or policies of the US Government, and mention of trade names, commercial products or organizations does not imply endorsement by the US Government.

## Author contributions

E.A.P., A.J.W. and T.I. conceptualized the project. E.A.P., A.J.W., S.D.P., N.B. and T.I. developed the methodology. E.A.P., A.J.W. and T.I. performed validation. E.A.P., A.J.W., S.D.P. and T.I. conducted formal analysis. E.A.P., A.J.W., S.D.P., N.B. and T.I. conducted investigations. E.A.P., A.J.W., N.B., Z.L. and T.I. acquired resources. E.A.P. wrote the original draft of the paper. E.A.P., A.J.W., S.D.P. and T.I. reviewed and edited the paper. E.A.P., A.J.W., S.D.P., N.B. and T.I. performed visualization. E.A.P. and T.I. supervised and administered the project. T.I. acquired funding.

## Competing interests

T.I. is involved on a pending patent related to the flow virometry methodology: PCT/US2022/042125, status pending. The other authors declare no competing interests.

## Additional information

**Extended data** is available for this paper at https://doi.org/10.1038/s41564-025-01925-9.

**Correspondence and requests for materials** should be addressed to Tijana Ivanovic.

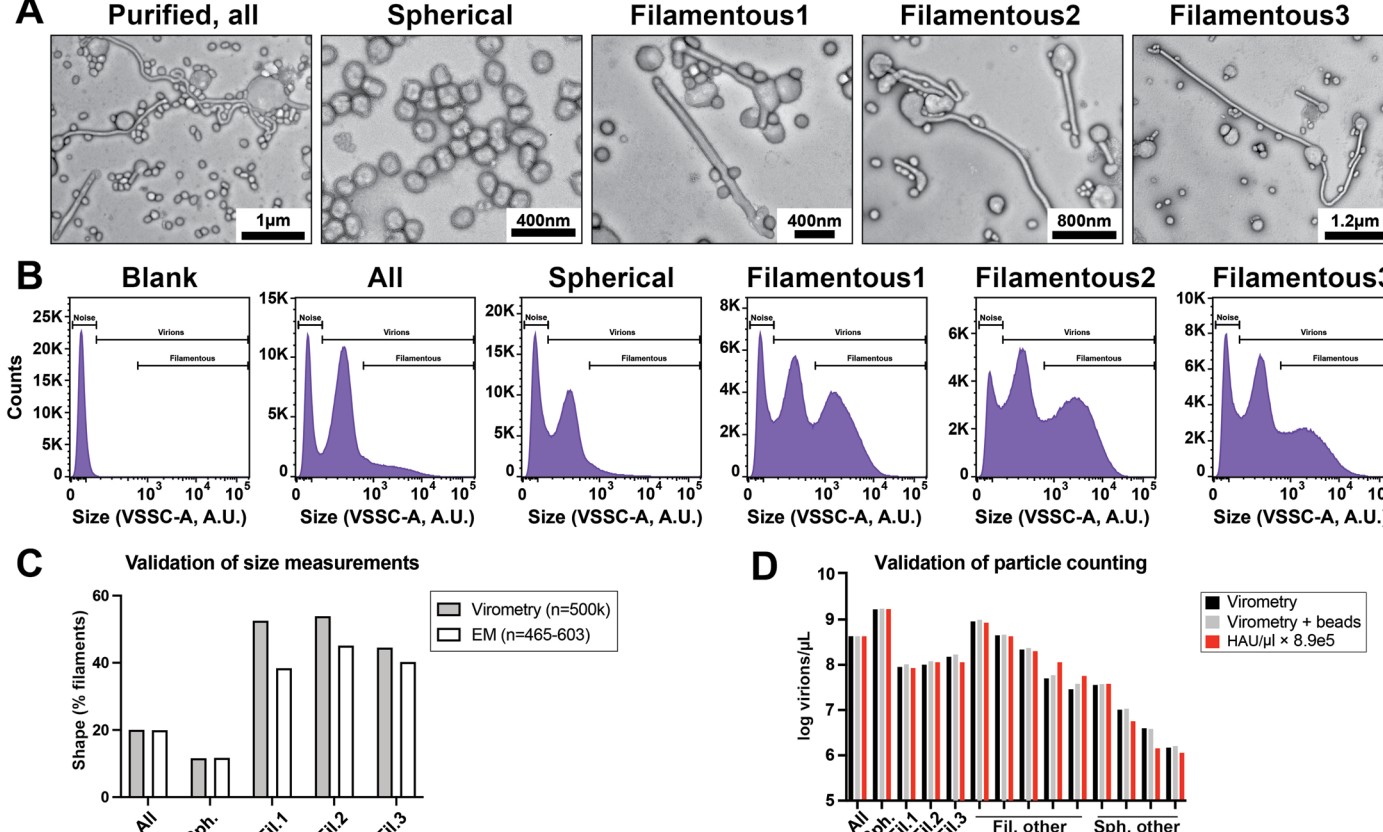

**Extended Data Fig. 1 | Validation of virion yield and size measurements.**
Unfractionated virus was purified by pelleting through a 20% sucrose cushion.
**a**. Representative electron micrographs of five biologically independent
samples. Samples include sucrose-cushion-purified virus (All), fractions
obtained by 20–60% sucrose gradient centrifugation of the sucrose-cushion-
purified virus (Spherical and Filamentous1-2), and a fraction obtained by
centrifugal fractionation of the sucrose-cushion-purified virus (Filamentous3).
**b**. Analysis of the 5 biologically independent samples in **a**, plus a buffer (Blank)
using flow virometry. **c**. Plot of the percentage of filamentous virions within
samples from **a** and **b**. Measurements were obtained either by flow virometry
(gray bars), using the Filamentous Violet Side Scatter (VSSC) threshold indicated
in **b**, or by identifying virions larger than 130nm by electron microscopy

(white bars). **d**. Virion counts were determined for the samples from **a** and
an additional nine fractions from the centrifugal fractionation experiments
enriched either in filaments (Fil. Other) or spheres (Sph. Other) to various
degrees. The counts were determined using flow virometry by either assuming a
constant flow rate (black bars) or by accounting for flow rate fluctuations using a
known concentration of fluorescent 170-nm beads (gray bars). Hemagglutination
(HA) assays were performed on all samples. The HA units (HAU) and virion counts
from the sucrose-cushion-purified virus were used to generate a scaling factor
of $8.9 \times 10^5$ virions/HAU. The scaling factor we obtained is consistent with
published literature that employed EM to correlate HAU to virion counts[29–31].
This factor was used to convert HAU values to estimated virion counts (red bars).

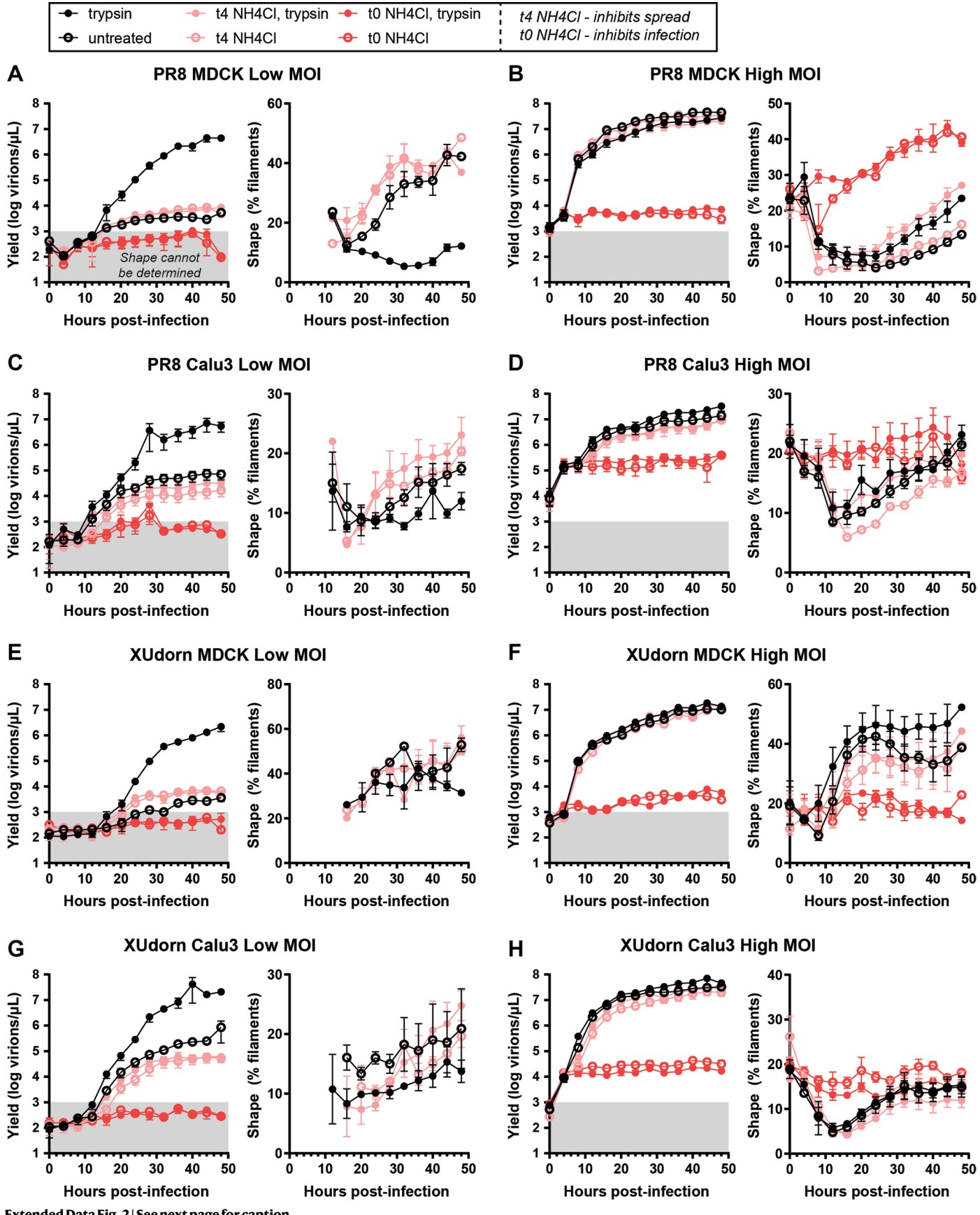

**Extended Data Fig. 2 | See next page for caption.**

**Extended Data Fig. 2 | Limiting infections to a single round by omitting trypsin or including ammonium chloride.** Yield and shape from time courses of infection with **a-d**. PR8 or **e-h**. XUdorn. High MOI is 6, and low MOI is 0.006. Ammonium chloride (NH4Cl) was added to infections either at the start, 0 h.p.i. (t0) to inhibit viral entry, or at 4 h.p.i. (t4) to inhibit viral spread. t0 ammonium chloride reveals detection of input virions that have detached from cells without becoming internalized. t4 ammonium chloride demonstrates that Calu3 cells permit limited activation of HA on released virions in the absence of trypsin, while omitting trypsin was sufficient to block new rounds of infection by released virions in MDCK cells. **All:** Plotted are mean and S.E.M. Data plotted derive from 3 biological replicates.

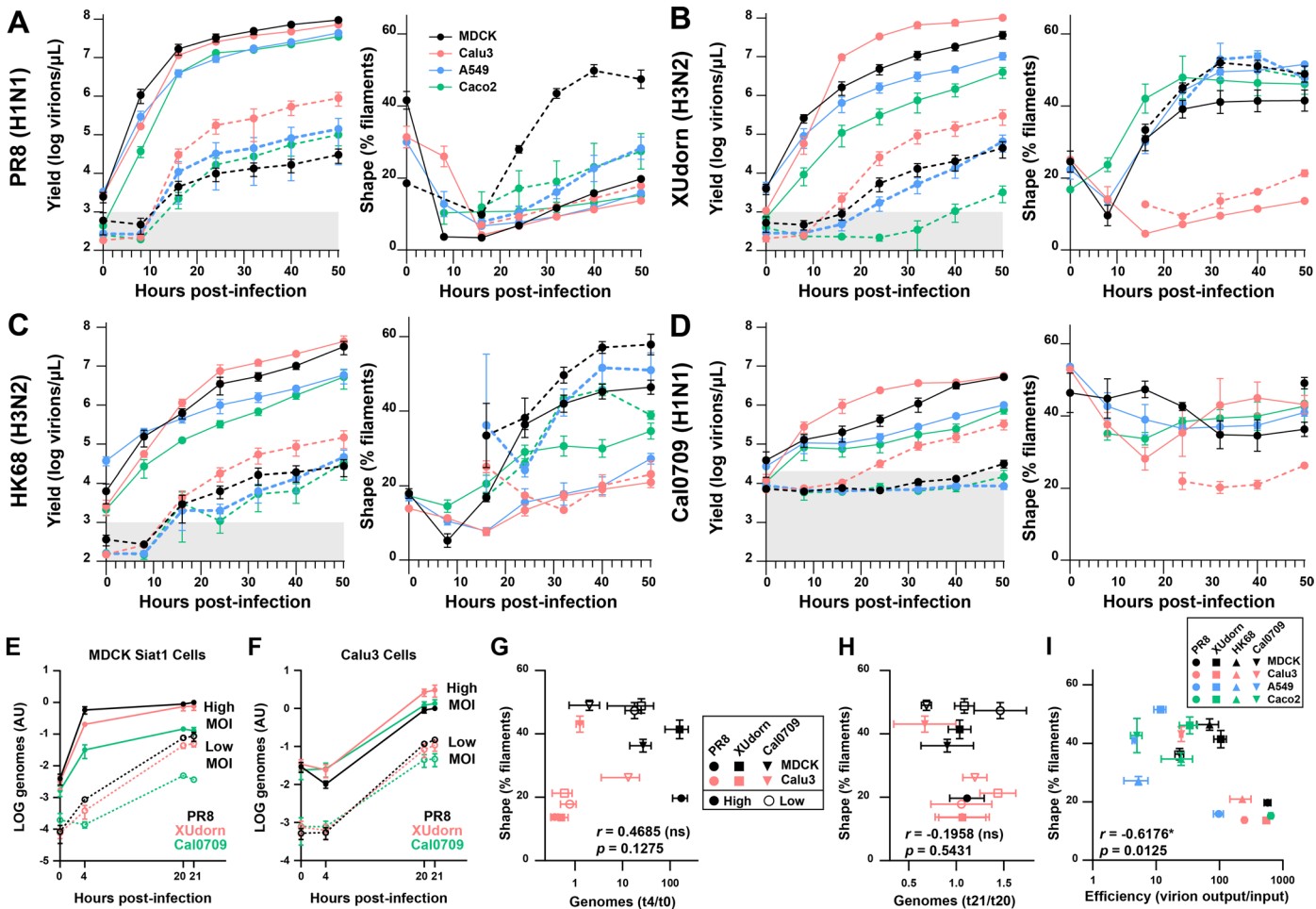

**Extended Data Fig. 3 | Supplement to Fig. 2. a-d.** Virion yields (left) and shape (right), determined by flow virometry of supernatants for the cell line indicated by color infected with **a.** PR8, **b.** XUdorn, **c.** HK68, and **d.** Cal0709 at high (solid line) and low (dashed line) MOI. Ammonium chloride was added at 4 h.p.i. to prevent spread in each case. **e.** Viral genomes (NP segment) within MDCK cells infected with PR8 (black), XUdorn (red), or Cal0709 (green) at high MOI (6, solid line) or low MOI (0.006, dashed line) measured by RT-qPCR at 0, 4, 20, and 21 h.p.i. **f.** As in **e**, except in Calu3 cells. **g.** Plotted is the shape of virions produced by

infections vs. the ratio of genomes at 4 h.p.i. to 0 h.p.i. (t4/t0), representing early amplification of viral genomes. Also shown are two-tailed Spearman's correlation *r* and *p* values. **h.** As in **g**, except plotting the ratio of genomes at 21 h.p.i. to 20 h.p.i (t21/t20), representing late changes in viral genomes. **i.** Efficiency vs. shape plots for the t50 time points at high MOI, along with two-tailed Spearman's correlation *r* and *p* values. All panels: Plotted are mean and S.E.M. **a-i.** Data plotted derive from 3 biological replicates.

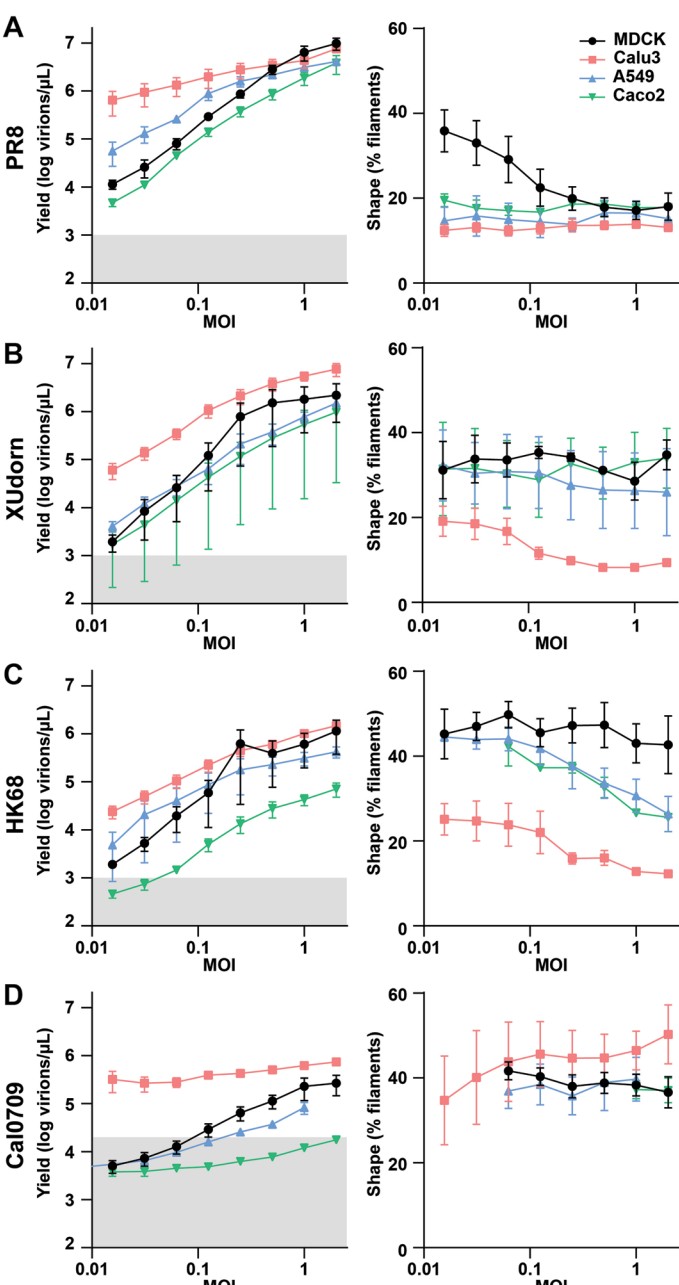

**Extended Data Fig. 4 | MOI and virion shape. a-d.** Virion yields (left) and shape (right) determined by flow virometry of supernatants at 24 h.p.i. for the cell line indicated by color infected with **a.** PR8, **b.** XUdorn, **c.** HK68, and **d.** Cal0709 at a range of MOIs. Plotted are means and S.E.M. **a-d.** Data plotted derive from 3 biological replicates.

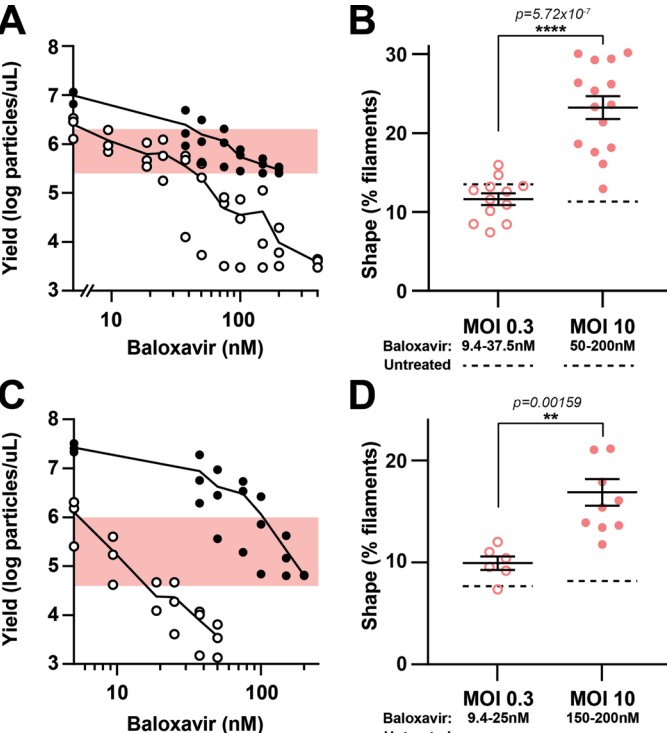

**Extended Data Fig. 5 | Supplement to Fig. 3. a.** Virion yields for MOI 10 (solid) and MOI 0.3 (open circle) infections of Calu3 cells with PR8 at various baloxavir concentrations. Red box indicates region of overlapping yields. **b**. Shape of virions from the shaded region in **a**. Dashed line: untreated. *p<0.05, **p<0.01, ***p<0.001, ****p<0.0001 by two-tailed unpaired t-test relative to an untreated control. **c**. As in **a** except for Calu3 cells infected with XUdorn. **d**. As in **b**, except for Calu3 cells infected with XUdorn. In **a** and **c**, raw data for independent replicates is shown instead of averages to facilitate selection of individual data points within the highlighter yield regimes. All error bars show S.E.M. **a-d**. Data plotted derive from 3 biological replicates.

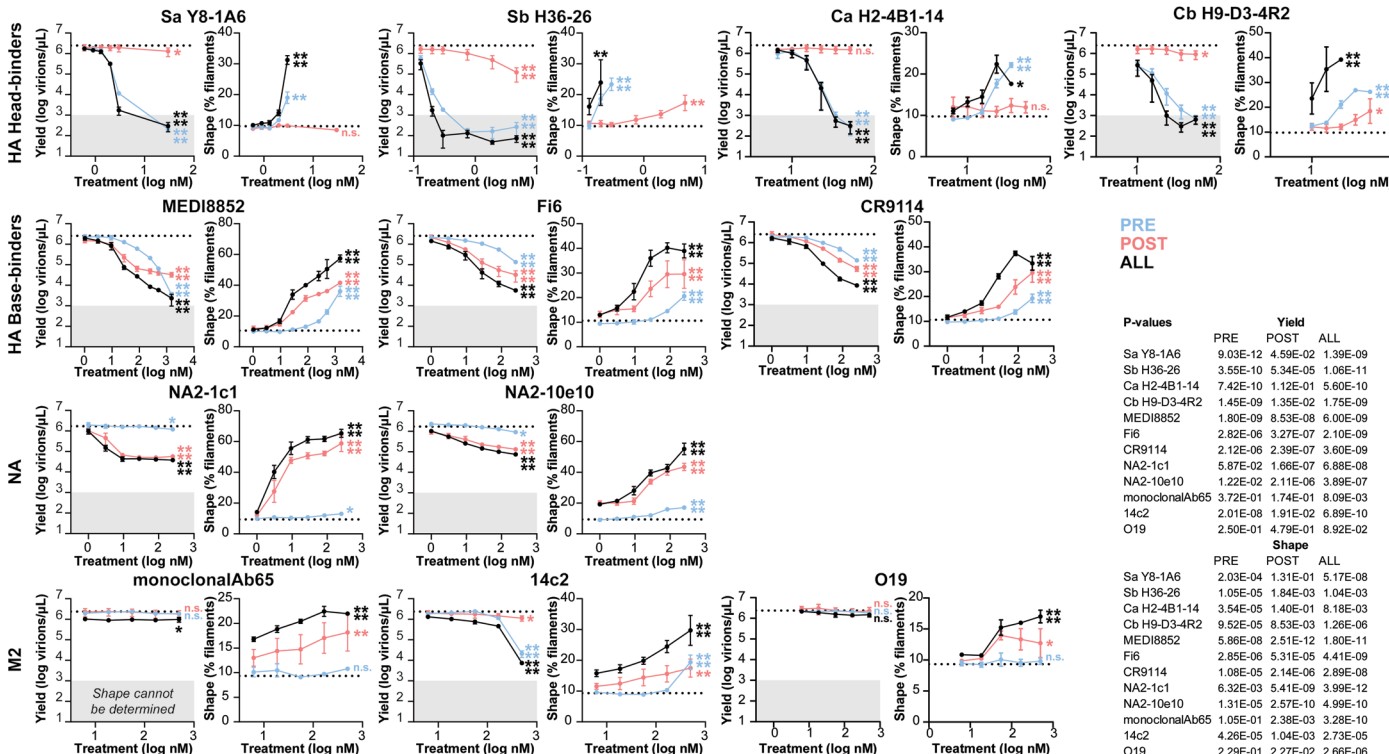

**Extended Data Fig. 6 | Antibody Pressure Drives Filament Assembly. Supplement to Fig. 4.** Pairs of plots showing viral yields (left) and shapes (right), measured by flow virometry of supernatants from MDCK cells, 24 hours post-PR8 infection at MOI 0.6, treated with antibodies at a range of indicated concentrations. Black: antibody present throughout infection. Blue: antibody present until 4 h.p.i. to test entry effects. Red: antibody added at 4 h.p.i. to test assembly effects. All antibodies are mouse IgG except Fi6, which is human IgG. Plotted are mean and S.E.M. *p<0.05, **p<0.01, ***p<0.001, ****p<0.0001 by one-tailed unpaired t-test relative to an untreated control for the highest antibody concentration. Gray shading indicates yields too low to reliably measure shape. Data plotted derive from 3 biological replicates.

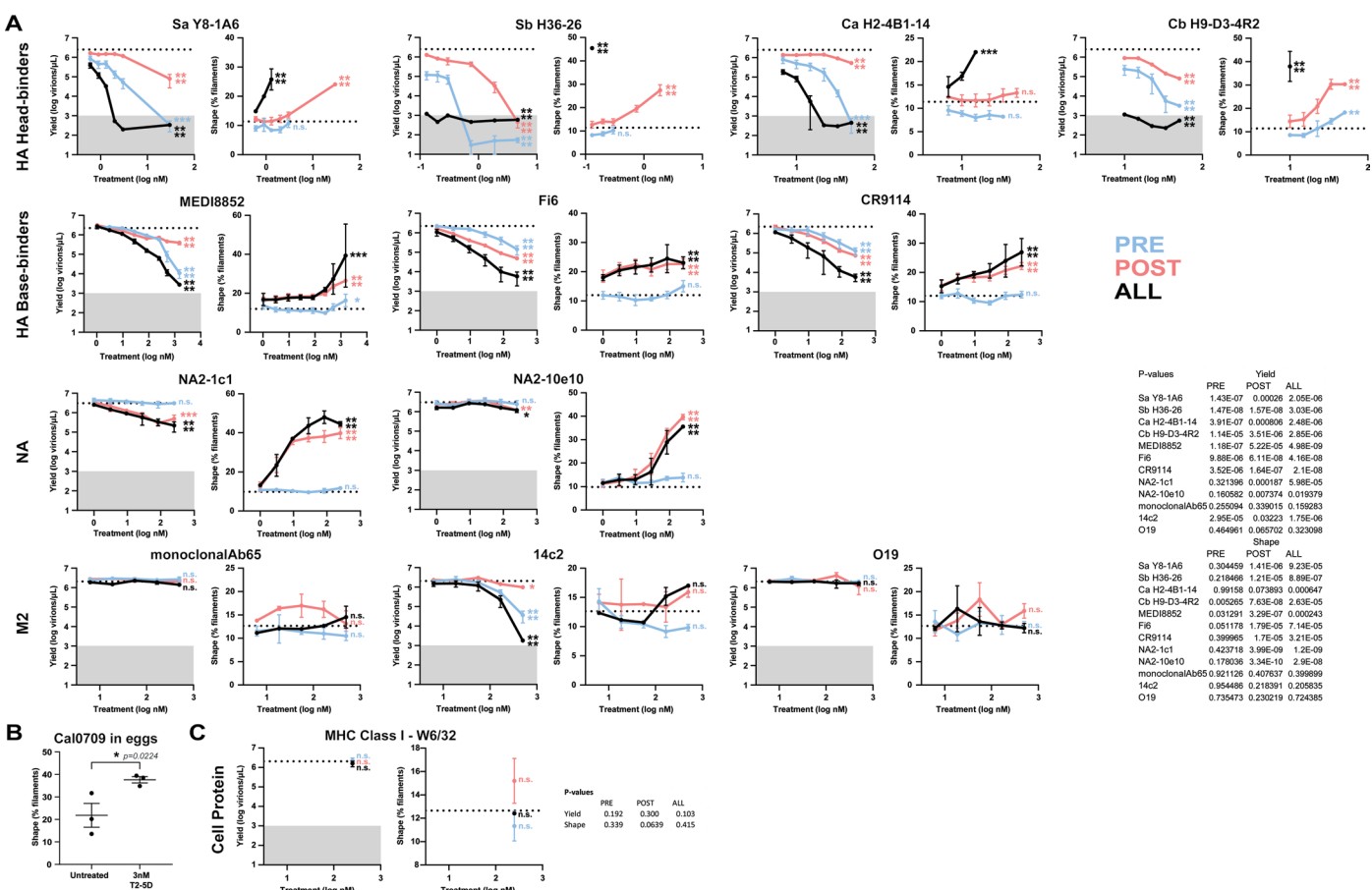

**Extended Data Fig. 7 | Antibody Pressure Drives Filament Assembly in Calu3 cells and eggs. Supplement to Fig. 4. a**. Pairs of plots showing viral yields (left) and shapes (right), measured by flow virometry of supernatants from MDCK cells, 24 hours post-PR8 infection at MOI 0.6, treated with antibodies at a range of indicated concentrations. Black: antibody present throughout infection. Blue: antibody present until 4 h.p.i. to test entry effects. Red: antibody added at 4 h.p.i. to test assembly effects. All antibodies are mouse IgG except Fi6, which is human IgG. **b**. Shape of virions measured by flow virometry of allantoic fluid from

embryonated chicken eggs, 24 hours post-Cal0709 infection at $10^6$ input virions, either untreated or treated with 150 pmol T2-5D antibody at 3 h.p.i. (estimated 3nM final concentration). **c**. As in **a**, except with a single concentration of an antibody targeting a cellular transmembrane protein. **All**: Plotted are mean and S.E.M. *p<0.05, **p<0.01, ***p<0.001, ****p<0.0001 by one-tailed unpaired t-test relative to an untreated control for the highest antibody concentration. Gray shading indicates yields too low to reliably measure shape. Data plotted derive from 3 biological replicates.

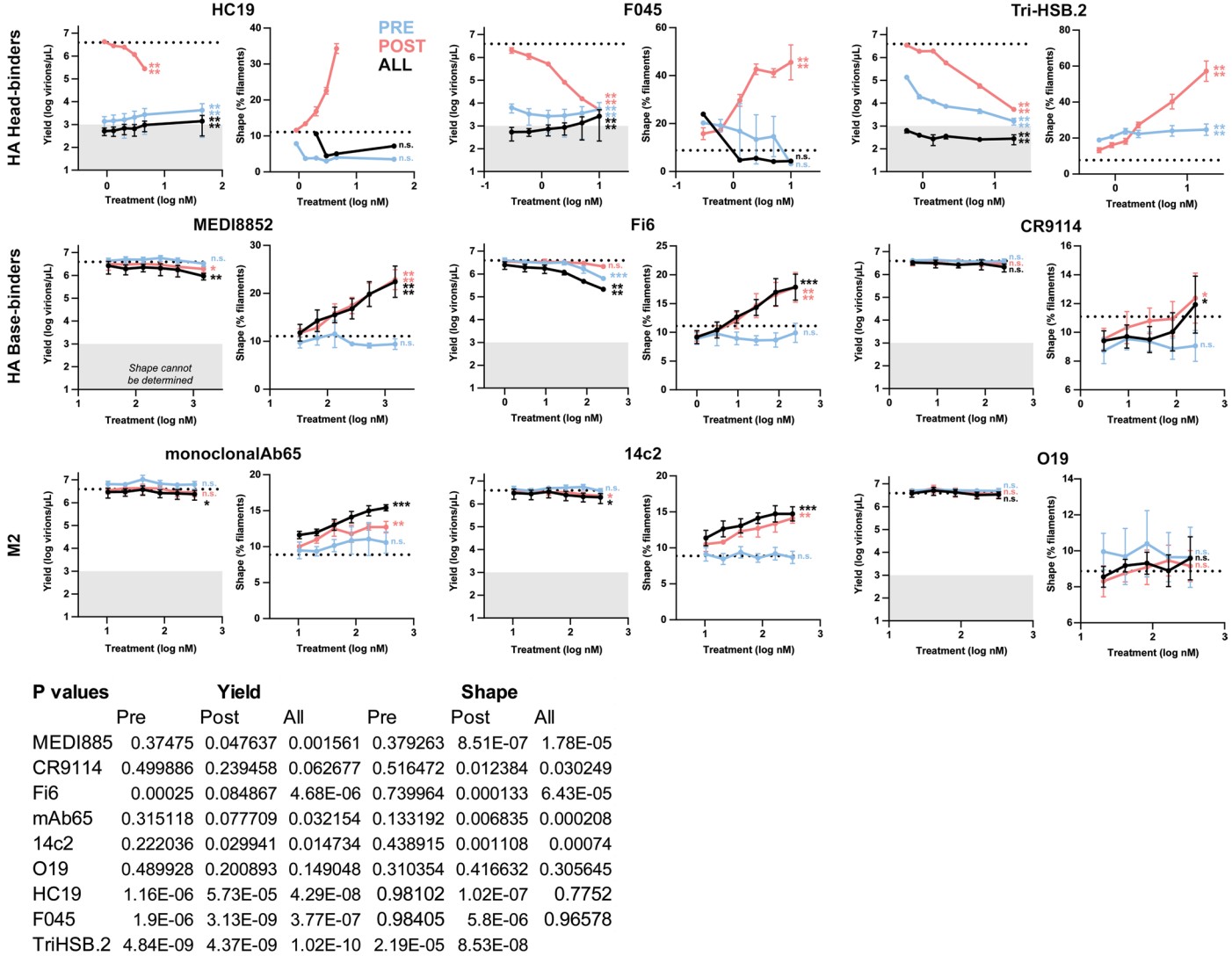

| P values | Yield | | | Shape | | |
|---|---|---|---|---|---|---|
| | Pre | Post | All | Pre | Post | All |
| MEDI885 | 0.37475 | 0.047637 | 0.001561 | 0.379263 | 8.51E-07 | 1.78E-05 |
| CR9114 | 0.499886 | 0.239458 | 0.062677 | 0.516472 | 0.012384 | 0.030249 |
| Fi6 | 0.00025 | 0.084867 | 4.68E-06 | 0.739964 | 0.000133 | 6.43E-05 |
| mAb65 | 0.315118 | 0.077709 | 0.032154 | 0.133192 | 0.006835 | 0.000208 |
| 14c2 | 0.222036 | 0.029941 | 0.014734 | 0.438915 | 0.001108 | 0.00074 |
| O19 | 0.489928 | 0.200893 | 0.149048 | 0.310354 | 0.416632 | 0.305645 |
| HC19 | 1.16E-06 | 5.73E-05 | 4.29E-08 | 0.98102 | 1.02E-07 | 0.7752 |
| F045 | 1.9E-06 | 3.13E-09 | 3.77E-07 | 0.98405 | 5.8E-06 | 0.96578 |
| TriHSB.2 | 4.84E-09 | 4.37E-09 | 1.02E-10 | 2.19E-05 | 8.53E-08 | |

**Extended Data Fig. 8 | Antibody Pressure Drives Filament Assembly in XUdorn. Supplement to Fig. 4.** Pairs of plots showing viral yields (left) and shapes (right), measured by flow virometry of supernatants from MDCK cells, 24 hours post-XUdorn infection at MOI 0.6, treated with antibodies or TriHSB at a range of indicated concentrations. Black: antibody present throughout infection. Blue: antibody present until 4 h.p.i. to test entry effects. Red: antibody added at 4 h.p.i. to test assembly effects. All antibodies are mouse IgG except Fi6, which is human IgG. TriHSB is a synthetically designed protein that binds H3-HA head domain. Plotted are mean and S.E.M. *p<0.05, **p<0.01, ***p<0.001, ****p<0.0001 by one-tailed unpaired t-test relative to an untreated control for the highest antibody concentration. Gray shading indicates yields too low to reliably measure shape. Data plotted derive from 3 biological replicates.

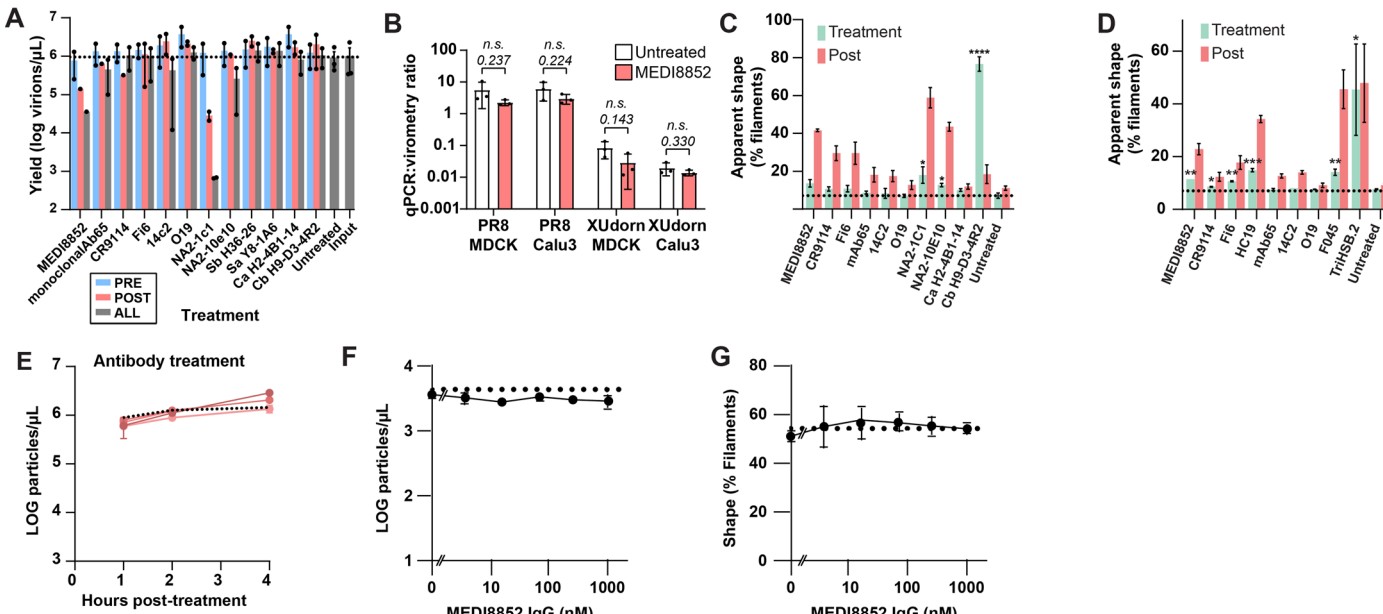

**Extended Data Fig. 9 | Induced filaments are infectious and not an artefact of antibody induced reuptake of spheres. a**. Supernatants from the antibody sweep experiment (Fig. 4) were used to infect fresh MDCK cells at 3 virions/cell, and yields were measured at 24 h.p.i. For each antibody, the highest concentration of antibody was chosen such that 1) virion concentration enabled reinfection at 3 virions/cell and 2) the amount of antibody in the supernatant (for 'POST' and 'ALL' supernatants) was not expected to be neutralizing. **b**. MDCK or Calu3 cells infected with PR8 or XUdorn at MOI 0.6 were treated with 250 nM MEDI8852 at 20 h.p.i. for two hours. Resulting supernatants were analysed for virion counts by flow virometry and for genome counts by qRT-PCR for the nucleoprotein gene segment. The mean ratio of qPCR intensity to virion count is plotted with S.E.M. **c**. Apparent shape of supernatants from untreated PR8 infections mixed with antibodies (green) compared to the shapes produced in infections treated with the same antibody during assembly (red) in Fig. 4a and Extended Data Fig. 6. **d**. Apparent shape of supernatants from untreated XUdorn

infections mixed with antibodies (green) compared to the shapes produced in infections treated with the same antibody during assembly (red) in Fig. 4c and Extended Data Fig. 8. **e**. MDCK cells infected with PR8 at MOI 0.6 for 20 hours were washed into media containing varying concentrations of MEDI8852. Viral yields 1, 2, or 4 hours following antibody treatment. Data from the same infections show in Fig. 5a. **f**. AF488-labelled PR8 virus was added to established PR8 infections along with varying concentrations of MEDI8852 IgG. After 2 hours, the AF488-labeled virus in the supernatant was measured by flow virometry. Plotted is the concentration of AF488-virus in the supernatant compared to the input (dashed line). **g**. As in **e**, except the shape of the AF488-virus is plotted. **All**: *p<0.05, **p<0.01, ***p<0.001, ****p<0.0001 by two-tailed unpaired t-test relative to an untreated control. Plotted are mean and S.E.M. Data plotted in **a**. derive from two biological replicates. All others panels, data plotted derive from 3 biological replicates.

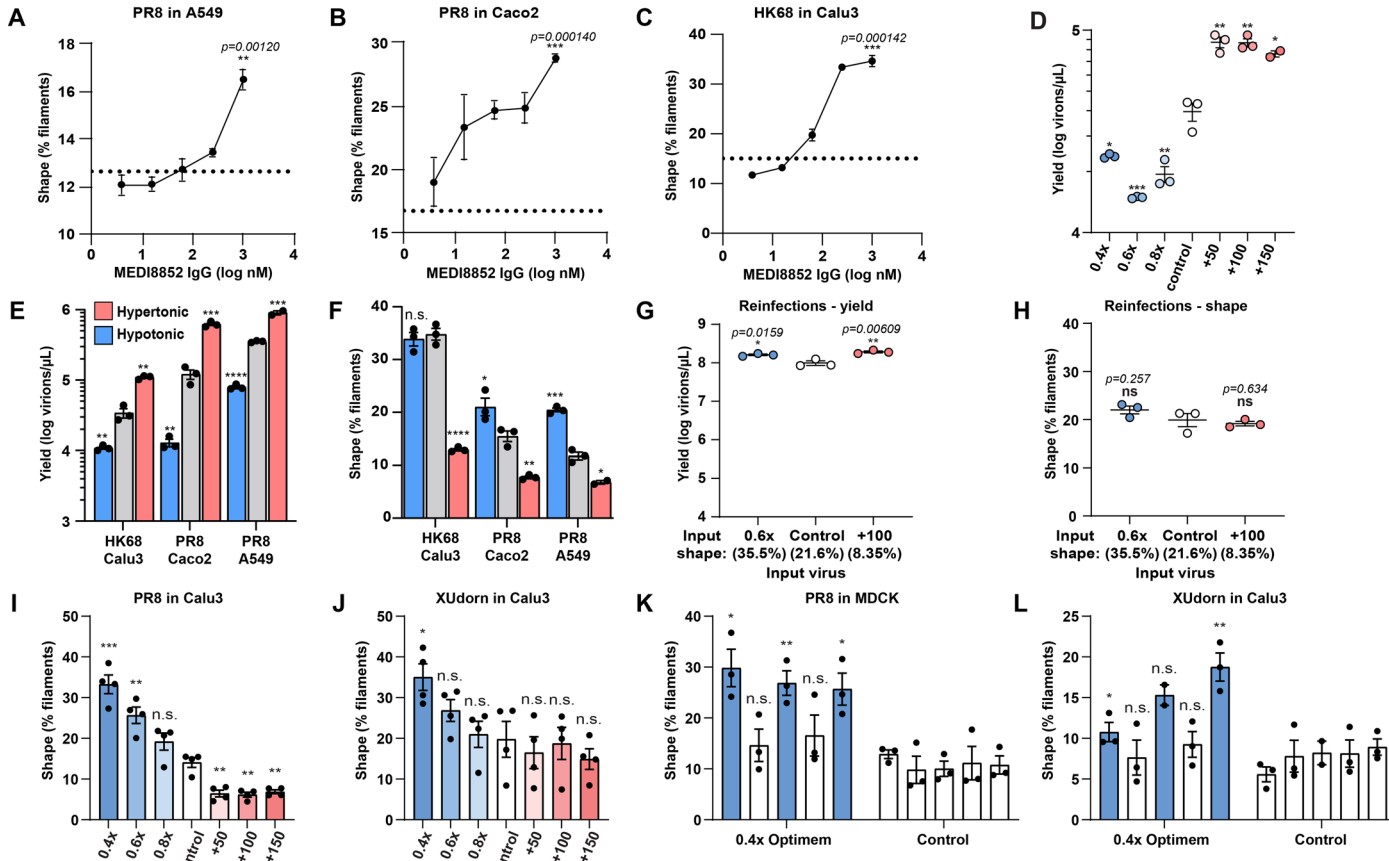

**Extended Data Fig. 10 | Transient effects extend to other cell lines and virus shapes. a**. A549 cells infected with PR8 at MOI 1 were washed and treated with MEDI8852 at 20 h.p.i. Supernatant was analysed for virion shape 60 minutes after treatment. **b-c**. As in **a**, except **b**. PR8 at MOI 1 in Caco2 cells and **c**. HK68 at MOI 3 in Calu3 cells. **a-c**. Dashed line represents the shape of an untreated sample. **d**. Particle yields from the osmotic shock treatments plotted in Fig. 5e. P-values (left to right): 0.0187*, 0.00382**, 0.0107*, 0.00199**, 0.00119**, and 0.00730**. **e-f**. The indicated cell line was infected with the indicated virus at MOI 1. 20 h.p.i., cells were washed into hypertonic (red), hypotonic (blue), or untreated media (grey). Supernatants were analysed for virion shape 60 minutes after treatment. **e**. Yields 24 h.p.i. of MDCK cells with the virus produced in Fig. 5c at 3 virions/ cell. P-values (left to right): 0.00877**, 0.00157**, 0.00505**, 0.000108***, $3.04 \times 10^{-6}$****, and 0.000823***. **f**. Shape of the samples in **e**. P values (left to right): 0.626, $4.54 \times 10^{-5}$****, 0.0457*, 0.00181**, 0.000374***, and 0.0127*. **g**. Yields at 24 h.p.i. of MDCK cells infected with the virus produced in Fig. 5e at 3 virions/cell. **h**. Shape of the virus produced in **g**. **i-j**. Calu3 cells infected with **i**. PR8 or **j**. XUdorn

at MOI 0.3 were washed into media of varying osmotic strengths at 20 h.p.i. Virion shape 30 minutes post media change. **i**. P-values (left to right): 0.000314***, 0.00269**, 0.0796(n.s.), 0.00204**, 0.00111**, and 0.00149**. **j**. P-values (left to right): 0.0317*, 0.218(n.s.), 0.830(n.s.), 0.597(n.s.), 0.870(n.s.), and 0.376(n.s.). **k**. Data also shown in Fig. 5f, here with all control points. MDCK cells infected with PR8 at MOI 0.6 were washed into media of varying osmotic strengths at 20.5, 21.5, and 22.5 h.p.i., and into normal media at 21 and 22 h.p.i. Virion shape 30 minutes post media change. P-values (left to right): 0.0109*, 0.311(n.s.), 0.00412**, 0.356(n.s.), and 0.0147*. **l**. As in **k**., except XUdorn in Calu3 cells. P-values (left to right): 0.0250*, 0.958(n.s.), 0.0659(n.s.), 0.650(n.s.), and 0.00788**. Plotted are mean and S.E.M. *p<0.05, **p<0.01, ***p<0.001, ****p<0.0001 by two-tailed unpaired t-test relative to an untreated control. For antibody treatments, the highest antibody concentration was used for comparison. **All**: Data plotted in **i** and **j** derive from 4 biological replicates. Data plotted in all other panels derive from 3 biological replicates.

# Reporting Summary

## Statistics

For all statistical analyses, confirm that the following items are present in the figure legend, table legend, main text, or Methods section.

| n/a | Confirmed | |
|---|---|---|
| ☐ | ☒ | The exact sample size (*n*) for each experimental group/condition, given as a discrete number and unit of measurement |
| ☐ | ☒ | A statement on whether measurements were taken from distinct samples or whether the same sample was measured repeatedly |
| ☐ | ☒ | The statistical test(s) used AND whether they are one- or two-sided *Only common tests should be described solely by name; describe more complex techniques in the Methods section.* |
| ☐ | ☒ | A description of all covariates tested |
| ☐ | ☒ | A description of any assumptions or corrections, such as tests of normality and adjustment for multiple comparisons |
| ☐ | ☒ | A full description of the statistical parameters including central tendency (e.g. means) or other basic estimates (e.g. regression coefficient) AND variation (e.g. standard deviation) or associated estimates of uncertainty (e.g. confidence intervals) |
| ☐ | ☒ | For null hypothesis testing, the test statistic (e.g. *F*, *t*, *r*) with confidence intervals, effect sizes, degrees of freedom and *P* value noted *Give P values as exact values whenever suitable.* |
| ☒ | ☐ | For Bayesian analysis, information on the choice of priors and Markov chain Monte Carlo settings |
| ☒ | ☐ | For hierarchical and complex designs, identification of the appropriate level for tests and full reporting of outcomes |
| ☒ | ☐ | Estimates of effect sizes (e.g. Cohen's *d*, Pearson's *r*), indicating how they were calculated |

*Our web collection on statistics for biologists contains articles on many of the points above.*

## Software and code

Policy information about availability of computer code

| Data collection | Flow virometry and flow cytometry data was collected in Cytexpert 2.5. |
|---|---|
| Data analysis | Flow virometry was analyzed in Cytexpert 2.5. Flow cytometry data was analyzed in Cytexpert 2.5 and FlowJo 10.9.0. Custom Matlab code (Matlab R2016a) was used to analyze electron microscopy data, and is available at https://github.com/tivanovic/Ivanovic-lab-analysis-codes |

For manuscripts utilizing custom algorithms or software that are central to the research but not yet described in published literature, software must be made available to editors and reviewers. We strongly encourage code deposition in a community repository (e.g. GitHub). See the Nature Portfolio guidelines for submitting code & software for further information.

## Data

Policy information about availability of data

All manuscripts must include a data availability statement. This statement should provide the following information, where applicable:
- Accession codes, unique identifiers, or web links for publicly available datasets
- A description of any restrictions on data availability
- For clinical datasets or third party data, please ensure that the statement adheres to our policy

The electron micrographs presented and used for analysis in figures 1 and extended data figure 1 are included in the source data. The uncropped immunofluorescence images presented and used for analysis in figure 5 are included in the source data. All other data plotted is available in source data.

# Research involving human participants, their data, or biological material

Policy information about studies with human participants or human data. See also policy information about sex, gender (identity/presentation), and sexual orientation and race, ethnicity and racism.

| | |
|---|---|
| Reporting on sex and gender | N/A |
| Reporting on race, ethnicity, or other socially relevant groupings | N/A |
| Population characteristics | N/A |
| Recruitment | N/A |
| Ethics oversight | N/A |

Note that full information on the approval of the study protocol must also be provided in the manuscript.

# Field-specific reporting

Please select the one below that is the best fit for your research. If you are not sure, read the appropriate sections before making your selection.

☒ Life sciences ☐ Behavioural & social sciences ☐ Ecological, evolutionary & environmental sciences

For a reference copy of the document with all sections, see nature.com/documents/nr-reporting-summary-flat.pdf

# Life sciences study design

All studies must disclose on these points even when the disclosure is negative.

| | |
|---|---|
| Sample size | For EM analysis we analyzed 465-603 particles deriving from 8-35 electron micrographs, as published previously for EM analysis of IAV shape(9). For other experiments three biological replicates were used as is standard in the field and in our previous work (9). Statistical methods were not used to predetermine sample size. The findings presented result from strongly significant differences evidenced by generally reproducible results (small error bars). |
| Data exclusions | In flow virometry, we excluded shape measurements from samples below 1000 virions/μL as they could not be reliably used to determine shape. For Cal0709, we excluded shape measurements from samples below 20,000 virions/μL for the same reason. Some measurements were uninterpretable due to occasional equipment malfunction. In these cases a second measurement was taken. |
| Replication | All experiments in the main figures were performed at least three times and found to be reproducible. The "no spread" samples shown in Figure 1D-G are repeated in figure 2. Data in Figure 3F was collected by AJW and was independently successfully replicated once by EP (data not shown). The data shown in Figure 4 are successful replicates of prior experiments (unpublished) that used a less-sensitive flow virometry method and a smaller panel of antibodies. The experiments shown in Figure 5 have now been repeated with additional cell lines and virus strains, shown in Extended Data Fig. 8. |
| Randomization | Samples were not randomized, however the arrangement of samples (in 24-well plates for example) were often scrambled to eliminate any systematic variations, such as from evaporation in edge wells. |
| Blinding | Blinding was not used, although our results depend on direct measurement and not on human classification or subjective analysis. |

# Reporting for specific materials, systems and methods

We require information from authors about some types of materials, experimental systems and methods used in many studies. Here, indicate whether each material, system or method listed is relevant to your study. If you are not sure if a list item applies to your research, read the appropriate section before selecting a response.

## Materials & experimental systems

| n/a | Involved in the study |
|---|---|
| ☐ | ☒ Antibodies |
| ☐ | ☒ Eukaryotic cell lines |
| ☒ | ☐ Palaeontology and archaeology |
| ☒ | ☐ Animals and other organisms |
| ☒ | ☐ Clinical data |
| ☒ | ☐ Dual use research of concern |
| ☒ | ☐ Plants |

## Methods

| n/a | Involved in the study |
|---|---|
| ☒ | ☐ ChIP-seq |
| ☐ | ☒ Flow cytometry |
| ☒ | ☐ MRI-based neuroimaging |

# Antibodies

**Antibodies used**

All monoclonal antibodies derived from hybridoma clones are referred to by their clone name.
H36-26 is an IAV H1 HA monoclonal mouse antibody. Source: Hybridoma from Jon Yewdell NIH/NIAID/LVD/CBS. Used at 0.5-10nM in infections or at 25nM for flow virometry.
HC19 is an IAV H3 HA monoclonal mouse antibody. Source: Purified from expression vectors from S. C. Harrison at Harvard Medical School. Used at 0.5-100nM.
HB65 is an IAV NP monoclonal mouse antibody. Source: Hybridoma from ATCC.Catalog number: H16-L10-4R5 Lot number:58696953. Used at 25nM.
Fi6 is an IAV broadly-neutralizing HA monoclonal human antibody. Source: Purified antibody from Jon Yewdell NIH/NIAID/LVD/CBS. Used at 1-1000nM.
CR9114 is an IAV broadly-neutralizing HA monoclonal human antibody. Source: Purified antibody from Jon Yewdell NIH/NIAID/LVD/CBS. Used at 1-250nM.
O19 is an IAV M2 monoclonal mouse antibody. Source: Purified antibody from Jon Yewdell NIH/NIAID/LVD/CBS. Used at 5-1000nM.
W6/32 is an HLA-ABC monoclonal mouse antibody. Source: Purified antibody from abcam..Catalog number: ab22432. Used at 250nM.
Y8-1A6-6 is an IAV H1 HA monoclonal mouse antibody. Source: Hybridoma from Jon Yewdell NIH/NIAID/LVD/CBS. Used at 0.1-100nM.
H2-4B1-14 is an IAV H1 HA monoclonal mouse antibody. Source: Hybridoma from Jon Yewdell NIH/NIAID/LVD/CBS. Used at 0.5-10nM.
H9-D3-4R2 is an IAV H1 HA monoclonal mouse antibody. Source: Hybridoma from Jon Yewdell NIH/NIAID/LVD/CBS. Used at 10-100nM.
NA2-1c1 is an IAV N1 NA monoclonal mouse antibody. Source: Hybridoma from Jon Yewdell NIH/NIAID/LVD/CBS. Used at 1-500nM.
NA2-10e10 is an IAV N1 NA monoclonal mouse antibody. Source: Hybridoma from Jon Yewdell NIH/NIAID/LVD/CBS. Used at 1-500nM.
monoclonalAB65 is an IAV M2 monoclonal mouse antibody. Source: Hybridoma from Xavier Saelens VIB-Ugent Center for Medical Biotechnology. Used at 1-500nM.
MEDI8852 is an IAV broadly-neutralizing HA monoclonal mouse antibody. Source: Purified from expression vectors from S. C. Harrison at Harvard Medical School. Used at 1-1000nM.
F045 is an IAV H3 HA monoclonal mouse antibody. Source: Purified from expression vectors cloned in-house. Used at 0.2-10nM.
14C2 is an IAV M2 monoclonal mouse antibody. Source: Purified from expression vectors cloned in-house. Used at 10-500nM.
T2-5D is an IAV HA monoclonal human antibody. Source: Purified antibody from Jon Yewdell NIH/NIAID/LVD/CBS. Used at 3nM.
T2-7D is an IAV HA monoclonal human antibody. Source: Purified antibody from Jon Yewdell NIH/NIAID/LVD/CBS. Used at 25nM.

**Validation**

H36-26 binds to H1 HA but not H3 HA by flow virometry, antigen mapped in Yewdell and Gerhard, 1981. This antibody binds to HA-expressing cells.
HC19 plasmids were sequence verified, by flow virometry, binds to H3 HA but not H1 HA.
HB65 stains IAV virus-infected cells. HB65 localizes to the nucleus of infected cells upon virus entry by immunofluorescence.
Fi6 is broadly neutralizing of IAV, epitope validated in Corti et al., 2011. Fi6 inhibits binding of HA by MEDI8852.
CR9114 is broadly neutralizing of IAV, epitope validated in Dreyfus et al., 2013.
O19 epitope is validated in Fu et al., 2009.
W6/32 is validated by abcam.
Y8-1A6-6 inhibits the binding of IAV virions by H36-26 by flow virometry, antigen mapped in Yewdell and Gerhard, 1981. This antibody elicits a VSSC shift on PR8 virions by flow virometry.
H2-4B1-14 inhibits IAV H1N1 infection, antigen mapped in Yewdell and Gerhard, 1981.
H9-D3-4R2 aggregates virus particles, antigen mapped in Yewdell and Gerhard, 1981.
NA2-1c1 antibody binds to an IAV NA-expressing cell line.
NA2-10e10 antibody binds to an IAV NA-expressing cell line.
monoclonalAB65 binds infected cells, does not neutralize infection.
MEDI8852 neutralizes IAV H1 and H3 infection, does not inhibit binding of IAV particles to cells, inhibits IAV fusion. This antibody elicits a VSSC shift on PR8 and XUdorn virions by flow virometry.
F045 inhibits the binding of IAV virions by HC19 by flow virometry. This antibody elicits a VSSC shift and aggregates XUdorn virions by flow virometry.
14C2 plasmids were sequence verified, the purified antibody binds to IAV infected cells and IAV virions by flow cytometry and flow virometry.
T2-5D is known to bind the head region of A/California/07/2009 HA from Huang et al., 2015. This antibody elicits a VSSC shift on Cal0709 virions by flow virometry.
T2-7D is known to bind the head region of A/California/07/2009 HA from Huang et al., 2015. This antibody elicits a VSSC shift on Cal0709 virions by flow virometry.

Antibodies used in the antibody sweep experiments exhibit effects that group by antigenic region, further validating their specificities.

# Eukaryotic cell lines

Policy information about cell lines and Sex and Gender in Research

| | |
|---|---|
| Cell line source(s) | MDCK.2 cell line was obtained from ATCC (CCL-34). MDCK-Siat1 cells were from Sigma (Cat#05071502). 293F cells were from Thermo Fisher CAT# R79007 (obtained from Stephen Harrison, Harvard Medical School). Calu3 cell line was obtained from ATCC (HTB-55). Caco2 cell line was obtained from ATCC (HTB-37). A549 cell line was obtained from ATCC (CCL-185). |
| Authentication | None of the MDCK cell lines were authenticated due to immature status of non-human cell line authentication. 293 cells were not authenticated because they were only used to express antibodies that we went on to purify and independently authenticate. Calu3, Caco2, and A549 cells were authenticated by ATCC prior to purchase. |
| Mycoplasma contamination | Caco2 and Calu3 cell lines were tested for mycoplamsa and confirmed negative in 2024. All other cell lines were tested for mycoplasma contamination in 2023 and were confirmed negative. |
| Commonly misidentified lines (See ICLAC register) | No cell lines used are listed in the database of commonly misidentified cell lines. |

# Plants

| | |
|---|---|
| Seed stocks | *Report on the source of all seed stocks or other plant material used. If applicable, state the seed stock centre and catalogue number. If plant specimens were collected from the field, describe the collection location, date and sampling procedures.* |
| Novel plant genotypes | *Describe the methods by which all novel plant genotypes were produced. This includes those generated by transgenic approaches, gene editing, chemical/radiation-based mutagenesis and hybridization. For transgenic lines, describe the transformation method, the number of independent lines analyzed and the generation upon which experiments were performed. For gene-edited lines, describe the editor used, the endogenous sequence targeted for editing, the targeting guide RNA sequence (if applicable) and how the editor was applied.* |
| Authentication | *Describe any authentication procedures for each seed stock used or novel genotype generated. Describe any experiments used to assess the effect of a mutation and, where applicable, how potential secondary effects (e.g. second site T-DNA insertions, mosiacism, off-target gene editing) were examined.* |

# Flow Cytometry

## Plots

Confirm that:

☒ The axis labels state the marker and fluorochrome used (e.g. CD4-FITC).

☒ The axis scales are clearly visible. Include numbers along axes only for bottom left plot of group (a 'group' is an analysis of identical markers).

☒ All plots are contour plots with outliers or pseudocolor plots.

☒ A numerical value for number of cells or percentage (with statistics) is provided.

## Methodology

| | |
|---|---|
| Sample preparation | For flow virometry, DyLight550-labeled Sb H36-26 IgG, HC19 IgG, and T2-5D IgG, and T2-7D IgG stock solutions were diluted to 11.85 nM, 50 nM, 25nM, and 25nM, respectively, in 0.2% BSA and HNE20 (20 mM HEPES NaOH pH 7.4, 150 mM NaCl, and 0.2 mM EDTA). Infected-cell supernatants were undiluted or diluted up to 1:30 in HNE20 and combined 1:1 with antibody dilution in BSA. Sb H36-26 IgG was used to label PR8, HC19 IgG was used to label XUdorn or HK68, and T2-5D IgG or T2-7D IgG were used to label Cal0709. Binding reactions were incubated at room temperature for 30 min to 1 hour, then diluted 1:250 in HNE20. For flow cytometry, cells were fixed and permeabilized as described previously(9) and then stained with AF488- or JF646-labeled HB65, for infectivity assays which recognizes IAV NP. |
| Instrument | Flow virometry and cytometry were performed using the CytoFLEX S platform (Beckman Coulter). Laser powers were 70mW for violet and 50mW for yellow. For flow virometry, gain values were set to 300 for VSSC and 1000 for RFP. For infectivity assays, gain values were set to 85-103 for FSC, 102-333 for SSC, and 100 for FITC, when using AF488-HB65,and 565 for APC, when using JF646-HB65. For antibody internalization experiments, gain values were set to 92 for SSC, 85 for FSC, 3000 for APC and 50 for RFP. |
| Software | Cytexpert 2.5 |
| Cell population abundance | No sorting was performed, the entire population was analyzed. |
| Gating strategy | For infectivity assays and antibody internalization experiments, gating was applied around the major peak defining cells on the SSC-A vs FSC-A contour plots, then around the major peak defining the singlet-cell population as revealed by events |

distributed along a diagonal on the FSC-A vs. FSC-H contour plots. For flow virometry, gating was based on size and fluorescent intensity of an anti-HA antibody as depicted in Figure 1.

☒ Tick this box to confirm that a figure exemplifying the gating strategy is provided in the Supplementary Information.

nature portfolio | reporting summary

