## [Peer Review File · Nature Microbiology]

Influenza A Virus rapidly adapts particle shape to environmental pressures

Corresponding Author: Dr Tijana Ivanovic

Version 0:

Reviewer comments:

Reviewer #1

(Remarks to the Author)

The study of Partlow et al. addresses determinants of influenza virus pleomorphy. The authors developed a flow virometry approach that allows them to conveniently quantify the fraction of spherical and filamentous virions in unprocessed supernatants. Applying this method they observed that at low MOIs the released particles show a higher proportion of filamentous structures. While this effect is only seen specifically for PR8 in MDCK cells, similar increase in filamentous virions were detected upon anti-influenza antibody treatment during viral infection and assembly. The authors then show that similar results can be obtained by keeping cells in hypotonic solutions or by inhibiting viral attachment, fusion, or transcription. They conclude that all these treatments, at low infection, lead to an increase in membrane tension which favors filamentous budding of the virions. This would have the implication for virions budding under pressure to release filamentous structures, previously shown to better evade immunity.

The flow virometry methods is a great tool for determining virion shape. The immediate effect of variations of MOI, osmotic or antibody pressure on the released virion shape are intriguing and offer a nice insight into so far unknown determinants of virion pleomorphy. Whether this is an effect of membrane tension, or an adaptation of the virus to low virus yield is not entirely answered in this study. While raising a very intriguing concept in influenza virus shaping and spread more data are necessary to convincingly support this model. If proven, the implications of this finding will be of general interest for scientists working on infection and immunity.

General points:

The presented concepts are highly intriguing. Is it possible to validate these findings using a broader array of virus strains and cell types? While the manuscript effectively demonstrates the antibody effect across 2 strains and 2 cell lines, it would be beneficial to also confirm the roles of membrane tension and viral replication inhibitors in this context.

I am wondering about virion numbers. Why do spherical and filamentous particles not differ in their hemagglutination efficiency? The measurements in Fig S1D suggest that 1 HAU refers to the same number of spherical or filamentous particles, which seems counter-intuitive. I am just wondering whether this is a fair normalization method, as variations in infectious virion numbers can affect the outcome of antibody assays.

Did the authors consider in their flow virometry methods and in their interpretations that extracellular vesicles released from infected cells also have spherical shapes and might incorporate and display HA while not being infectious?

Is there a possibility to measure whether antibody binding to the cells affects membrane tension?

Antibodies directed against virus proteins affect the shape of the released virions. What about antibodies targeting cellular host transmembrane proteins? Does this also affect the shape? Or the membrane tension? If not, what is the proposed mechanism of antibody sensing and shape shifting by the virions?

It appears shape correlates with yield. Correlation analysis of shape vs yield throughout the data would be important. Can the authors exclude that any setting resulting in low yield determines shape? This seems to be crucial factors since also transcription inhibition affects shape. Thus far, attributing these effects to membrane tension is a too strong conclusion.

Please always indicate number of replicates in the experiments. Dependent on the number of replicates, consider to use SD instead of SEM. ANOVA testing is more suited than t tests.

Specific points:

The validation of virometry should include supernatants of uninfected cultured cells. Extracellular vesicle release might interfere with the assays. Optimally, cells would be additionally transfected with HA-encoding plasmids to see whether HA-carrying extracellular vesicles are detected by Sb H36-26 labeling.

Fig 1 elegantly shows how yield increases stronger with spread as compared to the single cycle setting. To me it is surprising that already after 0 hours post infection (1 h post exposure) virus particles are released. Are these infectious particles?

"In contrast, XUdom was no more filamentous than PR8 in Calu3 cells (Fig. 1, F and G), an unexpected finding that highlights the extent to which the cellular environment of infection can direct viral assembly." This also highlights that this observation is only based on one virus isolate in one specific cell type. Is this effect for PR8 also observed in other cells for example A549?

Fig 1 At low MOIs the virus yield in Calu cells is higher ($>\log_4$) than in MDCK cells ($<\log_4$). Might this be the reason that there is no increase in filaments?

Fig 2 Can you show a correlation between burst and shape? These observations are interesting, however limited by the fact that it is only seen for PR8 in MDCK.

"We infected MDCK cells with PR8 in the presence of varying doses of a panel of antibodies specific to viral surface antigens and measured virion yield and shape after 24 hours." These results indicate that the lower the yields are, the higher the filamentous fraction is (in agreement with Fig 2). In addition, black and blue bars also correspond to lower "MOIs" by decreasing infection rate. Could this be simply due to decreasing infection rates in the presence of the antibodies? Please also comment on how antibody addition after infection results in decreased yield.

Fig 3B. Aside from shape, do the bound antibodies interfere with Sb H36-26 labeling in virometry? More validation is necessary. Fig 3D excludes this for one antibody, but possibly not for all.

Fig S6 please explain in detail how it was assured that the antibody present in the medium does not inhibit reinfection. Determining infectious viral titers would help to assess how residual antibody might affect the readout.

Fig 3D. How long was waited after mixing cells into media? The figure suggests that immediately after mixing there is already virions with ~8% filaments? But in C there is no virions at 0 h.

Figure 3. How do the observed effects change with lower or higher MOIs?

Fig 4. Again yield correlates with shape. Also needs to be confirmed by other virus/ other cells.

Fig 4D Why is there always virus yield when applying the inhibitors? Can inhibitors completely prevent yield? Determining the infectious titer of resulting particles would give more information about the efficiency of the inhibitors.

How do the endocytosis inhibitors affect secretion and shape of any non-viral vesicles from the cells? What does the hyper and hypotonic treatment do to any vesicles secreted by a cell?

Fig S9 should also show initial infectivity of the particles, not only progeny virus after infection with the particles.

File S1. These are good calculations, however, the release and uptake of extracellular vesicles is not considered and might change the whole system. There might be a homeostasis in the cells where endocytosis is accompanied with exocytosis. It is very hypothetical and to attribute these effects to membrane tension more convincing experimental data are necessary.

Minor points:

"We also reproduced previous findings that XUdom produces a high proportion of filaments in MDCK cells for all conditions (Fig. 1E)." Provide reference.

Figure S2: In the legend "D" is described but there is no panel "D"

Fig S4: It seems the labelling is wrong in 3 of the 4 panels. Otherwise low MOI results in higher yields than high MOIs.

"An exception was XUdom infection of MDCK". From four settings analysed, the increase in filaments is only seen in one, therefore it seem PR8 in MDCK is the exception.

Reviewer #2

(Remarks to the Author)

In this manuscript Partlow et al wish to explore an area of influenza virus biology that is fairly poorly studied (or at least often in large disagreement with itself). To do so they develop a flow virometry assay to sensitivity and rapidly determine the morphology of virions without a need for expensive, time-consuming and often highly artefact-filled electron microscopy (which most previous studies have used). With this assay they are able to perform time-courses of virion morphology at different times during infection and use a series of elegant experiments to show that differential membrane tension and antibodies can have a large effect on virion morphology, partially decoupling this phenotype from the genotype of the virus (which most literature has been focussed

on).

This is a great study, utilising a new technology to answer some quite basic biological questions in a way that I'm not aware has been done before. I would therefore recommend it is published and believe this would be a good fit for nature microbiology. I have a couple of small points below that I think could be addressed to improve the paper further but other than that I greatly enjoyed reading it.

1st paragraph, page 3 (and Figure S3) – “To limit infections to the initially infected cells, we omitted trypsin protease, which is required for the activation of HA on produced virions to enable further rounds of infection (fig. S3)”. While this is true for MDCKs I'm not sure either from previous literature or even the authors own data (Figure S3) this is true for Calu-3. Calu-3 express a range of serine proteases (such as TMPRSS2), many of which have been described to activate influenza haemagglutinin in the absence of exogenous trypsin. I don't think this really changes the conclusions much but the authors should alter their wording here.

One issue that the authors appear to have slightly overlooked – filamentous virions are unlikely to be able to enter cells via typical endocytosis pathways that spherical virions use, instead being more restricted to macropinocytosis – this will also have a large impact on membrane tension in cells infected by these filamentous virions as, I would assume macropinocytosis is much less efficient per virion for entry per the amount of membrane needed? It would be interesting for the authors to discuss this as it seems, by this route, infection filamentous virions could bias downstream progeny to be more likely to be filamentous?

Decision Letter:

15th January 2024

Dear Tijana,

Thank you for your patience while your manuscript "Influenza A Virus Infections Sense Host Membrane Tension to Dynamically Tune Assembly" was under peer-review at Nature Microbiology. And I must apologize again for the delay in getting back to you due to the high workload of the editorial team at the moment. Nonetheless, it has now been seen by 2 referees, whose expertise and comments you will find at the end of this email. Although they find your work of some potential interest, they have raised a number of concerns that will need to be addressed before we can consider publication of the work in Nature Microbiology.

In particular, reviewer #1 (whose report was send to you before) would like more supporting data for the proposed mechanisms here, additional confirmation of your findings using different virus strains and cells, as well as to take extracellular vesicles into account in your analysis. Reviewer #2 raised how different entry mechanisms between spherical and filamentous virus particles would play a role, as endocytosis and macropinocytosis pathways differ in membrane usage.

Should further experimental data allow you to address these criticisms, we would be happy to look at a revised manuscript.

Please include a data availability statement as a separate section after Methods but before references, under the heading "Data Availability". This section should inform readers about the availability of the data used to support the conclusions of your study. This information includes accession codes to public repositories (data banks for protein, DNA or RNA sequences, microarray, proteomics data etc...), references to source data published alongside the paper, unique identifiers such as URLs to data repository entries, or data set DOIs, and any other statement about data availability. At a minimum, you should include the following statement: "The data that support the findings of this study are available from the corresponding author upon request", mentioning any restrictions on availability. If DOIs are provided, we also strongly encourage including these in the Reference list (authors, title, publisher (repository name), identifier, year). For more guidance on how to write this section please see: <http://www.nature.com/authors/policies/data/data-availability-statements-data-citations.pdf>

* If you have not done so already we suggest that you begin to revise your manuscript so that it conforms to our Article format instructions at <http://www.nature.com/nmicrobiol/info/final-submission>. Refer also to any guidelines provided in this letter.

* Include a revised version of any required reporting checklist. It will be available to referees (and, potentially, statisticians) to aid

in their evaluation if the manuscript goes back for peer review. A revised checklist is essential for re-review of the paper.

Link Redacted

Note: This url links to your confidential homepage and associated information about manuscripts you may have submitted or be reviewing for us. If you wish to forward this e-mail to co-authors, please delete this link to your homepage first.

Nature Microbiology is committed to improving transparency in authorship. As part of our efforts in this direction, we are now requesting that all authors identified as 'corresponding author' on published papers create and link their Open Researcher and Contributor Identifier (ORCID) with their account on the Manuscript Tracking System (MTS), prior to acceptance. This applies to primary research papers only. ORCID helps the scientific community achieve unambiguous attribution of all scholarly contributions. You can create and link your ORCID from the home page of the MTS by clicking on 'Modify my Springer Nature account'. For more information please visit www.springernature.com/orcid.

If you wish to submit a suitably revised manuscript we would hope to receive it within 6 months. If you cannot send it within this time, please let us know. We will be happy to consider your revision, even if a similar study has been accepted for publication at Nature Microbiology or published elsewhere (up to a maximum of 6 months).

Yours sincerely,

Reviewer Expertise:

Referee #1: Virology, biochemistry

Referee #2: Respiratory viruses

Reviewer Comments:

Reviewer #1 (Remarks to the Author):

The study of Partlow et al. addresses determinants of influenza virus pleomorphy. The authors developed a flow virometry approach that allows them to conveniently quantify the fraction of spherical and filamentous virions in unprocessed supernatants. Applying this method they observed that at low MOIs the released particles show a higher proportion of filamentous structures. While this effect is only seen specifically for PR8 in MDCK cells, similar increase in filamentous virions were detected upon anti-influenza antibody treatment during viral infection and assembly. The authors then show that similar results can be obtained by keeping cells in hypotonic solutions or by inhibiting viral attachment, fusion, or transcription. They conclude that all these treatments, at low infection, lead to an increase in membrane tension which favors filamentous budding of the virions. This would have the implication for virions budding under pressure to release filamentous structures, previously shown to better evade immunity.

The flow virometry methods is a great tool for determining virion shape. The immediate effect of variations of MOI, osmotic or antibody pressure on the released virion shape are intriguing and offer a nice insight into so far unknown determinants of virion pleomorphy. Whether this is an effect of membrane tension, or an adaptation of the virus to low virus yield is not entirely answered in this study. While raising a very intriguing concept in influenza virus shaping and spread more data are necessary to convincingly support this model. If proven, the implications of this finding will be of general interest for scientists working on infection and immunity.

General points:

The presented concepts are highly intriguing. Is it possible to validate these findings using a broader array of virus strains and cell types? While the manuscript effectively demonstrates the antibody effect across 2 strains and 2 cell lines, it would be

beneficial to also confirm the roles of membrane tension and viral replication inhibitors in this context.

I am wondering about virion numbers. Why do spherical and filamentous particles not differ in their hemagglutination efficiency? The measurements in Fig S1D suggest that 1 HAU refers to the same number of spherical or filamentous particles, which seems counter-intuitive. I am just wondering whether this is a fair normalization method, as variations in infectious virion numbers can affect the outcome of antibody assays.

Did the authors consider in their flow virometry methods and in their interpretations that extracellular vesicles released from infected cells also have spherical shapes and might incorporate and display HA while not being infectious?

Is there a possibility to measure whether antibody binding to the cells affects membrane tension?

Antibodies directed against virus proteins affect the shape of the released virions. What about antibodies targeting cellular host transmembrane proteins? Does this also affect the shape? Or the membrane tension? If not, what is the proposed mechanism of antibody sensing and shape shifting by the virions?

It appears shape correlates with yield. Correlation analysis of shape vs yield throughout the data would be important. Can the authors exclude that any setting resulting in low yield determines shape? This seems to be crucial factors since also transcription inhibition affects shape. Thus far, attributing these effects to membrane tension is a too strong conclusion.

Please always indicate number of replicates in the experiments. Dependent on the number of replicates, consider to use SD instead of SEM. ANOVA testing is more suited than t tests.

Specific points:

The validation of virometry should include supernatants of uninfected cultured cells. Extracellular vesicle release might interfere with the assays. Optimally, cells would be additionally transfected with HA-encoding plasmids to see whether HA-carrying extracellular vesicles are detected by Sb H36-26 labeling.

Fig 1 elegantly shows how yield increases stronger with spread as compared to the single cycle setting. To me it is surprising that already after 0 hours post infection (1 h post exposure) virus particles are released. Are these infectious particles?

"In contrast, XUdorn was no more filamentous than PR8 in Calu3 cells (Fig. 1, F and G), an unexpected finding that highlights the extent to which the cellular environment of infection can direct viral assembly." This also highlights that this observation is only based on one virus isolate in one specific cell type. Is this effect for PR8 also observed in other cells for example A549?

Fig 1 At low MOIs the virus yield in Calu cells is higher ($>\log_4$) than in MDCK cells ($<\log_4$). Might this be the reason that there is no increase in filaments?

Fig 2 Can you show a correlation between burst and shape? These observations are interesting, however limited by the fact that it is only seen for PR8 in MDCK.

"We infected MDCK cells with PR8 in the presence of varying doses of a panel of antibodies specific to viral surface antigens and measured virion yield and shape after 24 hours." These results indicate that the lower the yields are, the higher the filamentous fraction is (in agreement with Fig 2). In addition, black and blue bars also correspond to lower "MOIs" by decreasing infection rate. Could this be simply due to decreasing infection rates in the presence of the antibodies? Please also comment on how antibody addition after infection results in decreased yield.

Fig 3B. Aside from shape, do the bound antibodies interfere with Sb H36-26 labeling in virometry? More validation is necessary. Fig 3D excludes this for one antibody, but possibly not for all.

Fig S6 please explain in detail how it was assured that the antibody present in the medium does not inhibit reinfection. Determining infectious viral titers would help to assess how residual antibody might affect the readout.

Fig 3D. How long was waited after mixing cells into media? The figure suggests that immediately after mixing there is already virions with ~8% filaments? But in C there is no virions at 0 h.

Figure 3. How do the observed effects change with lower or higher MOIs?

Fig 4. Again yield correlates with shape. Also needs to be confirmed by other virus/ other cells.

Fig 4D Why is there always virus yield when applying the inhibitors? Can inhibitors completely prevent yield? Determining the infectious titer of resulting particles would give more information about the efficiency of the inhibitors.

How do the endocytosis inhibitors affect secretion and shape of any non-viral vesicles from the cells? What does the hyper and hypotonic treatment do to any vesicles secreted by a cell?

Fig S9 should also show initial infectivity of the particles, not only progeny virus after infection with the particles.

File S1. These are good calculations, however, the release and uptake of extracellular vesicles is not considered and might

change the whole system. There might be a homeostasis in the cells where endocytosis is accompanied with exocytosis. It is very hypothetical and to attribute these effects to membrane tension more convincing experimental data are necessary.

Minor points:

"We also reproduced previous findings that XUDorn produces a high proportion of filaments in MDCK cells for all conditions (Fig. 1E)." Provide reference.

Figure S2: In the legend "D" is described but there is no panel "D"

Fig S4: It seems the labelling is wrong in 3 of the 4 panels. Otherwise low MOI results in higher yields than high MOIs.

"An exception was XUDorn infection of MDCK". From four settings analysed, the increase in filaments is only seen in one, therefore it seem PR8 in MDCK is the exception.

Reviewer #2 (Remarks to the Author):

In this manuscript Partlow et al wish to explore an area of influenza virus biology that is fairly poorly studied (or at least often in large disagreement with itself). To do so they develop a flow virometry assay to sensitivity and rapidly determine the morphology of virions without a need for expensive, time-consuming and often highly artefact-filled electron microscopy (which most previous studies have used). With this assay they are able to perform time-courses of virion morphology at different times during infection and use a series of elegant experiments to show that differential membrane tension and antibodies can have a large effect on virion morphology, partially decoupling this phenotype from the genotype of the virus (which most literature has been focussed on).

This is a great study, utilising a new technology to answer some quite basic biological questions in a way that I'm not aware has been done before. I would therefore recommend it is published and believe this would be a good fit for nature microbiology. I have a couple of small points below that I think could be addressed to improve the paper further but other than that I greatly enjoyed reading it.

1st paragraph, page 3 (and Figure S3) – "To limit infections to the initially infected cells, we omitted trypsin protease, which is required for the activation of HA on produced virions to enable further rounds of infection (fig. S3)". While this is true for MDCKs I'm not sure either from previous literature or even the authors own data (Figure S3) this is true for Calu-3. Calu-3 express a range of serine proteases (such as TMPRSS2), many of which have been described to activate influenza haemagglutinin in the absence of exogenous trypsin. I don't think this really changes the conclusions much but the authors should alter their wording here.

One issue that the authors appear to have slightly overlooked – filamentous virions are unlikely to be able to enter cells via typical endocytosis pathways that spherical virions use, instead being more restricted to macropinocytosis – this will also have a large impact on membrane tension in cells infected by these filamentous virions as, I would assume macropinocytosis is much less efficient per virion for entry per the amount of membrane needed? It would be interesting for the authors to discuss this as it seems, by this route, infection filamentous virions could bias downstream progeny to be more likely to be filamentous?

Version 1:

Reviewer comments:

Reviewer #1

(Remarks to the Author)

Partlow and colleagues tackle the really challenging problem of finding a mechanism for how and why influenza virus changes shape during replication and varies between variants, cells and under external pressure. They use the very innovative virometry method to determine percentages of spherical or filamentous virions released by cells. They conclude that under environmental pressure especially antibodies lead to influenza adapting by producing more filamentous virions. They interpret that this effect might be attributed to changed membrane tension potentially due to antibody binding potentially followed by endocytosis and similar effects seen under osmotic changes. While this manuscript is highly innovative and offers intriguing insights into the pleomorphic nature of influenza virus, also alternative explanations could lead to the observed effects. To make the authors' conclusions clearer, the manuscript would benefit from more streamlined and clearly presented experiments. The current setup and reasoning are somewhat hard to follow due to the topic's complexity. This paper is highly interesting and important, but in its current state, it might not capture the attention of a broader audience.

General points:

The manuscript would strongly benefit from improving readability and making the experimental results and their meanings clearer. Currently, understanding the text and figures requires extensive knowledge of virology and influenza, and even then, one must think very hard to follow.

Can the following alternative explanations be disproven?

- Speed: A high replication speed can explain lower number of filamentous virions. A fast virion assembly and budding might increase yield depending on high MOI, or viral determinants (fast polymerase) or cellular determinants (higher metabolism, less restriction proteins, not perfectly adapted to the species or osmotic stress).
- Release: A higher viral release leads to more spherical ones. Or a low release to more filamentous ones. This is closely tied to fast replication but focuses more on release. Maybe some cells/virus strains give the virus more time to assemble before release thus resulting in more filaments.
- Spread: More spread leads to more spherical ones (Fig 1). Antibodies prevent spread thus we have more filamentous ones in the presence of antibodies (Fig 5).
- Reuptake: Antibodies could lead to reuptake of the small virions which increases the relative number of filamentous virions in the supernatant.

Do you have an explanation for why, in general, first more filamentous, then spherical and then an increase in filament production is seen? How does this fit to membrane tension? Or to replication efficiency/speed?

It should become clear whether it is generally talked about the 24 h time point. Or whether the dynamic is analyzed. It gets a little confusing because of this complexity. Is the effect of attenuation on shape observable early or late or both?

In all experiments it is very important to mention whether spread is excluded or not.

The discussion can be longer and address a lot of the uncertainties.

Specific points:

The introduction says that in animals filamentous virions are favored. How is this with the new cow-infecting viruses (just a question out of curiosity for potential follow-ups)? Instead of external pressure, this might also be due to slow replication/low yield/low release in different species?

Fig 1C with and without spread is not defined at this time in the results text or the figure legend.

Line 91 the term "spread" should explicitly stated here in the context of trypsin as it is used throughout the text when trypsin is not added.

Fig 1: On Calus we see the typical filamentous – spherical – filamentous phenotype. On MDCK cells we see more filamentous particles for PR8 at low MOI without spread, and more filamentous particles for XUdom in general. Could the interpretation also be: it is not extrinsic factors but depends on cells or viral strain?

Fig 2A-D: We mostly see the typical filament-sphere-filament phenotype. On cacos there is always filament increase without the typical drop. H1N1 starts with a high filament levels. Again a conclusion might be that it depends on strain or cell or a combination of both.

Fig 2E: If understood correctly, the efficiency is defined as $\text{yield/cell} = \text{yield/MOI}$. This would also correspond to a fast replication/fast release of particles. This would also make sense as fast assembly and release of virions with only limited available genome/protein material would result in small virions (=spheres).

Extended Fig 3B: Isn't there a (weak) positive correlation between infectivity and shape?

Fig 3: this nicely indicates that the "proportion of filaments either decreased or remained constant with increasing MOI". Fig E is a little confusing as it shows exactly this, but in the text describes the opposite "that increasing filaments with lower MOI". It is both correct, but could be phrased easier. Again, could it be explained by: high MOI = fast replication/fast release = more spheres/less filaments?

Fig 3F: This graph is difficult to interpret or compare as the shape on Y and yield on X axis has never been shown before. Does yield correspond to inhibitor concentration? How does absence of inhibitor look like? Why is there always a yield in the presence of the inhibitors? This is unclear to me. But again results would fit to: low MOI/attenuated and slow replication = more filaments. (more time for a virus to assemble more and more proteins before release).

Fig 3F: the difference between the inhibitors and what conclusion can be drawn from it is not entirely clear to me. I agree that attachment inhibition best resembles low MOI, but would this mean less spheres? Another illustration of the graph would be helpful e.g. showing shape vs inhibitor concentrations (and also absence of inhibitor). Also here: high yield/high speed = more spheres.

Fig 3G: This experiment would also be easier to understand when the inhibitor concentrations would be plotted.

Could speed be measured by genome numbers per cell?

Fig 4: Is this done with or without spread? If with spread: do the antibodies inhibit spread? This would again recapitulate a "lower MOI" and explain higher filaments.

Fig 5AB This is a very nice and important experiment. It might show that spread is not responsible for the observed effects?

Fig 5CD. This again could be explained by: slower or higher release of virus/slower higher replication / meaning faster or slower assembly.

Maybe antibodies lead to re-uptake of small virions, but not the larger ones? This is why the fraction of larger virions is found in the supernatant. This would also fit to Extended Fig 9A.

Maybe doing the antibody assay with or without trypsin or ammonium chloride could help solving this?

Replies to rebuttal:

Thank you for the suggestion. Understanding whether host-targeted antibodies alter the shape of virions would help unravel the mechanism behind antibody treatments. We tested whether an α MHC antibody affects the shape of virions, seen in Extended Data Figure 5C, and saw no effects, but further experiments would be needed to answer this conclusively.

Now included in Extended Data Figure 9 we show evidence that α HA antibodies are endocytosed by infected cells. If the antibodies are inducing endocytosis, this could attribute to increased membrane tension. There are examples of α Host antibodies inducing endocytosis in certain autoimmune diseases (such as Graves' disease) or in monoclonal antibody treatments (such as Trastuzumab). However, α Host antibodies are unlikely to be physiologically relevant in influenza infections and were thus not explored further in this manuscript.

This could also indicate that only antibodies that target viral structures interfere with spheres. Meaning that it is not membrane tension but targeting the virion and reducing spread or increasing uptake or similar effects.

Reviewer #2

(Remarks to the Author)

In general my points have been thoroughly addressed, and, to my interpretation, I believe the concerns of the other reviewer have also been addressed (although I will defer to their own interpretation on this).

I enjoyed re-reading through the paper, believe the additions have improved it from the first submission and still believe it has a lot of merit and would recommend its publication with the following minor textual addition:

My one additional point after re-review is that the authors are now using a pair of virus isolates (rather than the laboratory strains/attenuated reassortants they previously used). This includes the use of the pandemic strain A/Hong Kong/1/1968 (HK68). My understanding is there is variability in different regions about whether this qualifies as a virus that should be used at CL2 or CL3 (with most places moving towards restricting the virus to CL3-only use due to a lack of cross-reactive immunity in the population). It would be helpful, with the addition of these new viruses, if the authors included a biosafety statement at the start of the materials and methods to describe what containment level this work was done at, and what permissions were sought, and by which authorities to use these virus isolates (at either CL2 or CL3).

Decision Letter:

16th July 2024

Dear Tijana,

Thank you for your patience while your manuscript "Influenza A Virus Infections Sense Attenuated Infection to Dynamically Tune Assembly" was under peer-review at Nature Microbiology. It has now been seen again by the original 2 referees, whose expertise and comments you will find at the end of this email. Although they find your work of some potential interest, they have raised a number of concerns that will need to be addressed before we can consider publication of the work in Nature Microbiology.

In particular, reviewer #1 remains unconvinced that the conclusions are fully supported and suggests additional experimentation and discussions to put alternative theories into context. Please also add a biosafety section to the methods, as requested by reviewer #2.

Should further experimental data allow you to address these criticisms, we would be happy to look at a revised manuscript.

Please include a data availability statement as a separate section after Methods but before references, under the heading "Data Availability". This section should inform readers about the availability of the data used to support the conclusions of your study. This information includes accession codes to public repositories (data banks for protein, DNA or RNA sequences, microarray, proteomics data etc...), references to source data published alongside the paper, unique identifiers such as URLs to data repository entries, or data set DOIs, and any other statement about data availability. At a minimum, you should include the following statement: "The data that support the findings of this study are available from the corresponding author upon request", mentioning any restrictions on availability. If DOIs are provided, we also strongly encourage including these in the Reference list (authors, title, publisher (repository name), identifier, year). For more guidance on how to write this section please see: <http://www.nature.com/authors/policies/data/data-availability-statements-data-citations.pdf>

* If you have not done so already we suggest that you begin to revise your manuscript so that it conforms to our Article format instructions at <http://www.nature.com/nmicrobiol/info/final-submission>. Refer also to any guidelines provided in this letter.

When submitting the revised version of your manuscript, please pay close attention to our [href="https://www.nature.com/nature-portfolio/editorial-policies/image-integrity">Digital Image Integrity Guidelines](https://www.nature.com/nature-portfolio/editorial-policies/image-integrity) and to the following points below:

Link Redacted

Note: This url links to your confidential homepage and associated information about manuscripts you may have submitted or be reviewing for us. If you wish to forward this e-mail to co-authors, please delete this link to your homepage first.

Nature Microbiology is committed to improving transparency in authorship. As part of our efforts in this direction, we are now requesting that all authors identified as 'corresponding author' on published papers create and link their Open Researcher and Contributor Identifier (ORCID) with their account on the Manuscript Tracking System (MTS), prior to acceptance. This applies to primary research papers only. ORCID helps the scientific community achieve unambiguous attribution of all scholarly contributions. You can create and link your ORCID from the home page of the MTS by clicking on 'Modify my Springer Nature account'. For more information please visit www.springernature.com/orcid.

If you wish to submit a suitably revised manuscript we would hope to receive it within 3 months. If you cannot send it within this time, please let us know. We will be happy to consider your revision, even if a similar study has been accepted for publication at Nature Microbiology or published elsewhere (up to a maximum of 3 months).

Yours sincerely,

Reviewer Expertise:

Referee #1: molecular virology

Referee #2: influenza, virus entry

Reviewer Comments:

Reviewer #1 (Remarks to the Author):

Partlow and colleagues tackle the really challenging problem of finding a mechanism for how and why influenza virus changes shape during replication and varies between variants, cells and under external pressure. They use the very innovative virometry method to determine percentages of spherical or filamentous virions released by cells. They conclude that under environmental pressure especially antibodies lead to influenza adapting by producing more filamentous virions. They interpret that this effect might be attributed to changed membrane tension potentially due to antibody binding potentially followed by endocytosis and similar effects seen under osmotic changes. While this manuscript is highly innovative and offers intriguing insights into the pleomorphic nature of influenza virus, also alternative explanations could lead to the observed effects. To make the authors' conclusions clearer, the manuscript would benefit from more streamlined and clearly presented experiments. The current setup and reasoning are somewhat hard to follow due to the topic's complexity. This paper is highly interesting and important, but in its current state, it might not capture the attention of a broader audience.

General points:

The manuscript would strongly benefit from improving readability and making the experimental results and their meanings clearer. Currently, understanding the text and figures requires extensive knowledge of virology and influenza, and even then, one must think very hard to follow.

Can the following alternative explanations be disproven?

- Speed: A high replication speed can explain lower number of filamentous virions. A fast virion assembly and budding might increase yield depending on high MOI, or viral determinants (fast polymerase) or cellular determinants (higher metabolism, less restriction proteins, not perfectly adapted to the species or osmotic stress).
- Release: A higher viral release leads to more spherical ones. Or a low release to more filamentous ones. This is closely tied to fast replication but focuses more on release. Maybe some cells/virus strains give the virus more time to assemble before release thus resulting in more filaments.
- Spread: More spread leads to more spherical ones (Fig 1). Antibodies prevent spread thus we have more filamentous ones in the presence of antibodies (Fig 5).
- Reuptake: Antibodies could lead to reuptake of the small virions which increases the relative number of filamentous virions in the supernatant.

Do you have an explanation for why, in general, first more filamentous, then spherical and then an increase in filament production is seen? How does this fit to membrane tension? Or to replication efficiency/speed?

It should become clear whether it is generally talked about the 24 h time point. Or whether the dynamic is analyzed. It gets a little confusing because of this complexity. Is the effect of attenuation on shape observable early or late or both?

In all experiments it is very important to mention whether spread is excluded or not.

The discussion can be longer and address a lot of the uncertainties.

Specific points:

The introduction says that in animals filamentous virions are favored. How is this with the new cow-infecting viruses (just a question out of curiosity for potential follow-ups)? Instead of external pressure, this might also be due to slow replication/low yield/low release in different species?

Fig 1C with and without spread is not defined at this time in the results text or the figure legend.

Line 91 the term "spread" should explicitly stated here in the context of trypsin as it is used throughout the text when trypsin is not added.

Fig 1: On Calus we see the typical filamentous – spherical – filamentous phenotype. On MDCK cells we see more filamentous particles for PR8 at low MOI without spread, and more filamentous particles for XUDom in general. Could the interpretation also be: it is not extrinsic factors but depends on cells or viral strain?

Fig 2A-D: We mostly see the typical filament-sphere-filament phenotype. On cacos there is always filament increase without the typical drop. H1N1 starts with a high filament levels. Again a conclusion might be that it depends on strain or cell or a combination of both.

Fig 2E: If understood correctly, the efficiency is defined as $\text{yield/cell} = \text{yield/MOI}$. This would also correspond to a fast replication/fast release of particles. This would also make sense as fast assembly and release of virions with only limited available genome/protein material would result in small virions (=spheres).

Extended Fig 3B: Isn't there a (weak) positive correlation between infectivity and shape?

Fig 3: this nicely indicates that the "proportion of filaments either decreased or remained constant with increasing MOI". Fig E is a little confusing as it shows exactly this, but in the text describes the opposite "that increasing filaments with lower MOI". It is both correct, but could be phrased easier. Again, could it be explained by: high MOI = fast replication/fast release = more spheres/less filaments?

Fig 3F: This graph is difficult to interpret or compare as the shape on Y and yield on X axis has never been shown before. Does yield correspond to inhibitor concentration? How does absence of inhibitor look like? Why is there always a yield in the presence of the inhibitors? This is unclear to me. But again results would fit to: low MOI/attenuated and slow replication = more filaments. (more time for a virus to assemble more and more proteins before release).

Fig 3F: the difference between the inhibitors and what conclusion can be drawn from it is not entirely clear to me. I agree that attachment inhibition best resembles low MOI, but would this mean less spheres? Another illustration of the graph would be helpful e.g. showing shape vs inhibitor concentrations (and also absence of inhibitor). Also here: high yield/high speed = more spheres.

Fig 3G: This experiment would also be easier to understand when the inhibitor concentrations would be plotted.

Could speed be measured by genome numbers per cell?

Fig 4: Is this done with or without spread? If with spread: do the antibodies inhibit spread? This would again recapitulate a "lower MOI" and explain higher filaments.

Fig 5AB This is a very nice and important experiment. It might show that spread is not responsible for the observed effects?

Fig 5CD. This again could be explained by: slower or higher release of virus/slower higher replication / meaning faster or slower assembly.

Maybe antibodies lead to re-uptake of small virions, but not the larger ones? This is why the fraction of larger virions is found in the supernatant. This would also fit to Extended Fig 9A.

Maybe doing the antibody assay with or without trypsin or ammonium chloride could help solving this?

Replies to rebuttal:

Thank you for the suggestion. Understanding whether host-targeted antibodies alter the shape of virions would help unravel the mechanism behind antibody treatments. We tested whether an α MHC antibody affects the shape of virions, seen in Extended Data Figure 5C, and saw no effects, but further experiments would be needed to answer this conclusively.

Now included in Extended Data Figure 9 we show evidence that α HA antibodies are endocytosed by infected cells. If the antibodies are inducing endocytosis, this could attribute to increased membrane tension. There are examples of α Host antibodies inducing endocytosis in certain autoimmune diseases (such as Graves' disease) or in monoclonal antibody treatments (such as Trastuzumab). However, α Host antibodies are unlikely to be physiologically relevant in influenza infections and were thus not explored further in this manuscript.

This could also indicate that only antibodies that target viral structures interfere with spheres. Meaning that it is not membrane tension but targeting the virion and reducing spread or increasing uptake or similar effects.

Reviewer #2 (Remarks to the Author):

In general my points have been thoroughly addressed, and, to my interpretation, I believe the concerns of the other reviewer have also been addressed (although I will defer to their own interpretation on this).

I enjoyed re-reading through the paper, believe the additions have improved it from the first submission and still believe it has a lot of merit and would recommend its publication with the following minor textual addition:

My one additional point after re-review is that the authors are now using a pair of virus isolates (rather than the laboratory strains/attenuated reassortants they previously used). This includes the use of the pandemic strain A/Hong Kong/1/1968 (HK68). My understanding is there is variability in different regions about whether this qualifies as a virus that should be used at CL2 or CL3 (with most places moving towards restricting the virus to CL3-only use due to a lack of cross-reactive immunity in the population). It would be helpful, with the addition of these new viruses, if the authors included a biosafety statement at the start of the materials and methods to describe what containment level this work was done at, and what permissions were sought, and by which authorities to use these virus isolates (at either CL2 or CL3).

Version 2:

Reviewer comments:

Reviewer #1

(Remarks to the Author)

The revised manuscript by Partlow et al. is greatly improved in terms of clarity, presentation, and supporting experiments. The studied problem is highly complex, making the evaluation and interpretation as well as the presentation of the data challenging.

First, I believe this manuscript advances the field both methodologically and scientifically. However, I still have some doubts regarding certain conclusions and interpretations. What the paper elegantly shows is that shape shifts during replication and varies depending on the cell line, viral strain, MOI, and environmental pressure. Thus, shape-shifting appears to be an adaptation of the virus to its environment (host cell and external pressures), which is not genetically encoded but variable. However, my conclusion is that it all depends on viral "efficiency," as the authors call it, and they demonstrate a nice correlation. My hypothesis would be that anything affecting efficiency also affects viral shape. This could include the cell in which the virus replicates, antibody pressure, antiviral pressure, osmotic pressure...

What I do not see is that antibody pressure, which does not affect efficiency, results in a shape shift. Thus, shape appears to be simply an effect of viral efficiency. In the end, this may help the virus remain in the host, continue spreading, and escape neutralizing antibodies. If the authors address this concern and tone down the narrative that antibody pressure directly induces filaments that evade antibodies (and put it in the discussion), it will strengthen the paper.

General points:

It seems to me that "efficiency" is the decisive factor, which varies between cells, viral strains, and environmental factors such as pressure from antibodies, replication inhibitors, or osmotic stress. This means that something binding to the virus but not changing efficiency would not result in shape-shifting. In the case of M2 antibodies, this appears to be true: only when M2 antibodies have antiviral activity does a shape change occur. For those where they don't affect replication, the shape remains unchanged (e.g., EFig 8 O19; EFig 7 A Ab65; O19). In EFig 6, there are minor effects without inhibition, but these appear negligible compared to other settings.

To me it remains inconclusive if shape shift is a response to antibody pressure. While in the end it might help the virus spread further, it seems to be a consequence of replication and release rather than directly related to antibody pressure e.g. by membrane tension or similar effects. This should be discussed more clearly, and maybe only late in the discussion. The text should emphasize the main findings and shift away from focusing on antibody pressure.

Specific points:

The terms "band" and "smear" appear suddenly in the figures without further introduction. A brief explanation in the results section would help clarify this.

Fig 1 Consider using the same Y-axis scale across the panels to avoid making small shifts appear disproportionate.

Lines 111-115 I am not entirely convinced by these conclusions. It seems that the cell line is the most crucial factor in determining shape, with variation between viruses within each cell line. For example, MDCK cells are always more filamentous and Calu3 cells more spherical. Similarly, PR8 is typically more spherical, and Xudorn is more filamentous, as described in the literature. Perhaps normalizing within each cell line, setting one virus to 100%, and then showing how the others deviate would clarify this. While the final conclusion that no single factor is decisive is correct, cells and strains do predetermine the extent and range of % shape.

Fig 2C: Including Spearman's r and p -values would strengthen the statement that there is no correlation. Visually, it seems there might be one (higher infectivity = more filaments). Again, the clustering of cells highlights that they are the most determining factor.

Fig 2D How was the shape of Cal on A549 determined here (it wasn't in Figures 2A/B)? Also, why are only two Caco results shown?

Fig 2F Performing these correlations is challenging because, as seen in Figure 2A, shape percentage already depends on the cell type. Comparing MDCK and Calu3 % shapes, for example, will not correlate. Possibly, a variable like % "shapeshift" would be more comparable.

Fig 2E Here % shapeshift is nicely used as a variable. It might be helpful to organize this by cell type, as in Figure 2A. Consider

using % shapeshift in more of the analyses.

Fig. 2G This also suggests that low vs. high virion output is cell-type dependent and could influence shape (a cell-intrinsic factor).

What is the explanation for the lower increase in filaments when using entry inhibitors compared to other treatments?

174-177 Why is the post effect stronger than the pre effect?

179- Is this really accurate? When there's no inhibition, there's hardly any change, only significant minor shifts. It might help to align all Y axes for easier comparison and acknowledge the actual ranges of change.

227-228 The same (even stronger) effects were seen with Baloxavir. This supports the idea that the main effect is related to efficiency. It might be interesting to study Baloxavir kinetics.

252 I would weaken this statement. As described, shape depends on cell type and strain but is also variable under certain conditions.

287-288 his again shows that anything enhancing or weakening efficiency results in more spheres or filaments, respectively.

Do the resulting filaments escape antibody neutralization more effectively? Can this be tested using antibody titration on the resulting virions?

Have you tried enhancing efficiency with something like nucleotides, growth factors, or FCS? If so, does this shift the virus toward more spherical shapes? Temperature differences might also affect shape outcomes.

Have you confirmed how the filaments resulting from antibody treatment look via TEM?

Decision Letter:

5th November 2024

Dear Tijana,

Thank you for your patience while your manuscript "Influenza A Virus Rapidly Assembles Filaments in Response to Attenuation and Antibodies" was under peer-review at Nature Microbiology. I must apologize for the delay in reaching a decision since I was attending a conference. Nonetheless, it has now been seen by the original referee #1, whose expertise and comments you will find at the of this email. You will see from their comments below that while they find your work of interest, some important points are raised. We are very interested in the possibility of publishing your study in Nature Microbiology, but would like to consider your response to these concerns in the form of a revised manuscript before we make a final decision on publication.

In particular, you will see that this referee remains unconvinced that antibody pressure leads to changes in shape, suggests to tone down the conclusions and to refocus the manuscript on virus efficiency by changing the text. We would recommend to revise your manuscript according to the suggestions of this referee and would not demand additional experiments. If the editorial team is satisfied with the textual changes, we will not send this back to peer review.

If you have not done so already please begin to revise your manuscript so that it conforms to our Article format instructions at <http://www.nature.com/nmicrobiol/info/final-submission/>

The usual length limit for a Nature Microbiology Article is six display items (figures or tables) and 3,000 words. We have some flexibility, and can allow a revised manuscript at 3,500 words, but please consider this a firm upper limit. There is a trade-off of ~250 words per display item, so if you need more space, you could move a Figure or Table to Supplementary Information.

Some reduction could be achieved by focusing any introductory material and moving it to the start of your opening 'bold' paragraph, whose function is to outline the background to your work, describe in a sentence your new observations, and explain your main conclusions. The discussion should also be limited. Methods should be described in a separate section following the discussion, we do not place a word limit on Methods.

Nature Microbiology titles should give a sense of the main new findings of a manuscript, and should not contain punctuation. Please keep in mind that we strongly discourage active verbs in titles, and that they should ideally fit within 90 characters each (including spaces).

We strongly support public availability of data. Please place the data used in your paper into a public data repository, if one exists, or alternatively, present the data as Source Data or Supplementary Information. If data can only be shared on request, please explain why in your Data Availability Statement, and also in the correspondence with your editor. For some data types,

deposition in a public repository is mandatory - more information on our data deposition policies and available repositories can be found at <https://www.nature.com/nature-research/editorial-policies/reporting-standards#availability-of-data>.

Please include a data availability statement as a separate section after Methods but before references, under the heading "Data Availability". This section should inform readers about the availability of the data used to support the conclusions of your study. This information includes accession codes to public repositories (data banks for protein, DNA or RNA sequences, microarray, proteomics data etc...), references to source data published alongside the paper, unique identifiers such as URLs to data repository entries, or data set DOIs, and any other statement about data availability. At a minimum, you should include the following statement: "The data that support the findings of this study are available from the corresponding author upon request", mentioning any restrictions on availability. If DOIs are provided, we also strongly encourage including these in the Reference list (authors, title, publisher (repository name), identifier, year). For more guidance on how to write this section please see: <http://www.nature.com/authors/policies/data/data-availability-statements-data-citations.pdf>

To improve the accessibility of your paper to readers from other research areas, please pay particular attention to the wording of the paper's opening bold paragraph, which serves both as an introduction and as a brief, non-technical summary in about 150 words. If, however, you require one or two extra sentences to explain your work clearly, please include them even if the paragraph is over-length as a result. The opening paragraph should not contain references. Because scientists from other sub-disciplines will be interested in your results and their implications, it is important to explain essential but specialised terms concisely. We suggest you show your summary paragraph to colleagues in other fields to uncover any problematic concepts.

If your paper is accepted for publication, we will edit your display items electronically so they conform to our house style and will reproduce clearly in print. If necessary, we will re-size figures to fit single or double column width. If your figures contain several parts, the parts should form a neat rectangle when assembled. Choosing the right electronic format at this stage will speed up the processing of your paper and give the best possible results in print. We would like the figures to be supplied as vector files - EPS, PDF, AI or postscript (PS) file formats (not raster or bitmap files), preferably generated with vector-graphics software (Adobe Illustrator for example). Please try to ensure that all figures are non-flattened and fully editable. All images should be at least 300 dpi resolution (when figures are scaled to approximately the size that they are to be printed at) and in RGB colour format. Please do not submit Jpeg or flattened TIFF files. Please see also 'Guidelines for Electronic Submission of Figures' at the end of this letter for further detail.

Figure legends must provide a brief description of the figure and the symbols used, within 350 words, including definitions of any error bars employed in the figures.

When submitting the revised version of your manuscript, please pay close attention to our [href="https://www.nature.com/nature-research/editorial-policies/image-integrity">Digital Image Integrity Guidelines. and to the following points below:](https://www.nature.com/nature-research/editorial-policies/image-integrity)

Please include a statement before the acknowledgements naming the author to whom correspondence and requests for materials should be addressed.

Finally, we require authors to include a statement of their individual contributions to the paper -- such as experimental work, project planning, data analysis, etc. -- immediately after the acknowledgements. The statement should be short, and refer to authors by their initials. For details please see the Authorship section of our joint Editorial policies at http://www.nature.com/authors/editorial_policies/authorship.html

* include a point-by-point response to any editorial suggestions and to our referees. Please include your response to the editorial suggestions in your cover letter, and please upload your response to the referees as a separate document.

* ensure it complies with our format requirements for Letters as set out in our guide to authors at www.nature.com/nmicrobiol/info/gta/

* state in a cover note the length of the text, methods and legends; the number of references; number and estimated final size of figures and tables

* resubmit electronically if possible using the link below to access your home page:

Link Redacted

*This url links to your confidential homepage and associated information about manuscripts you may have submitted or be reviewing for us. If you wish to forward this e-mail to co-authors, please delete this link to your homepage first.

Please ensure that all correspondence is marked with your Nature Microbiology reference number in the subject line.

Nature Microbiology is committed to improving transparency in authorship. As part of our efforts in this direction, we are now requesting that all authors identified as 'corresponding author' on published papers create and link their Open Researcher and Contributor Identifier (ORCID) with their account on the Manuscript Tracking System (MTS), prior to acceptance. This applies to primary research papers only. ORCID helps the scientific community achieve unambiguous attribution of all scholarly contributions. You can create and link your ORCID from the home page of the MTS by clicking on 'Modify my Springer Nature account'. For more information please visit www.springernature.com/orcid.

We hope to receive your revised paper within three weeks. If you cannot send it within this time, please let us know.

Yours sincerely,

Reviewer Expertise:

Referee #1: virus entry, virometry

Reviewers Comments:

Reviewer #1 (Remarks to the Author):

The revised manuscript by Partlow et al. is greatly improved in terms of clarity, presentation, and supporting experiments. The studied problem is highly complex, making the evaluation and interpretation as well as the presentation of the data challenging.

First, I believe this manuscript advances the field both methodologically and scientifically. However, I still have some doubts regarding certain conclusions and interpretations. What the paper elegantly shows is that shape shifts during replication and varies depending on the cell line, viral strain, MOI, and environmental pressure. Thus, shape-shifting appears to be an adaptation of the virus to its environment (host cell and external pressures), which is not genetically encoded but variable. However, my conclusion is that it all depends on viral "efficiency," as the authors call it, and they demonstrate a nice correlation. My hypothesis would be that anything affecting efficiency also affects viral shape. This could include the cell in which the virus replicates, antibody pressure, antiviral pressure, osmotic pressure...

What I do not see is that antibody pressure, which does not affect efficiency, results in a shape shift. Thus, shape appears to be simply an effect of viral efficiency. In the end, this may help the virus remain in the host, continue spreading, and escape neutralizing antibodies. If the authors address this concern and tone down the narrative that antibody pressure directly induces filaments that evade antibodies (and put it in the discussion), it will strengthen the paper.

General points:

It seems to me that "efficiency" is the decisive factor, which varies between cells, viral strains, and environmental factors such as pressure from antibodies, replication inhibitors, or osmotic stress. This means that something binding to the virus but not changing efficiency would not result in shape-shifting. In the case of M2 antibodies, this appears to be true: only when M2 antibodies have antiviral activity does a shape change occur. For those where they don't affect replication, the shape remains unchanged (e.g., EFig 8 O19; EFig 7 A Ab65; O19). In EFig 6, there are minor effects without inhibition, but these appear negligible compared to other settings.

To me it remains inconclusive if shape shift is a response to antibody pressure. While in the end it might help the virus spread further, it seems to be a consequence of replication and release rather than directly related to antibody pressure e.g. by membrane tension or similar effects. This should be discussed more clearly, and maybe only late in the discussion. The text should emphasize the main findings and shift away from focusing on antibody pressure.

Specific points:

The terms "band" and "smear" appear suddenly in the figures without further introduction. A brief explanation in the results section would help clarify this.

Fig 1 Consider using the same Y-axis scale across the panels to avoid making small shifts appear disproportionate.

Lines 111-115 I am not entirely convinced by these conclusions. It seems that the cell line is the most crucial factor in determining shape, with variation between viruses within each cell line. For example, MDCK cells are always more filamentous

and Calu3 cells more spherical. Similarly, PR8 is typically more spherical, and Xudorn is more filamentous, as described in the literature. Perhaps normalizing within each cell line, setting one virus to 100%, and then showing how the others deviate would clarify this. While the final conclusion that no single factor is decisive is correct, cells and strains do predetermine the extent and range of % shape.

Fig 2C: Including Spearman's r and p -values would strengthen the statement that there is no correlation. Visually, it seems there might be one (higher infectivity = more filaments). Again, the clustering of cells highlights that they are the most determining factor.

Fig 2D How was the shape of Cal on A549 determined here (it wasn't in Figures 2A/B)? Also, why are only two Caco results shown?

Fig 2F Performing these correlations is challenging because, as seen in Figure 2A, shape percentage already depends on the cell type. Comparing MDCK and Calu3 % shapes, for example, will not correlate. Possibly, a variable like % "shapeshift" would be more comparable.

Fig 2E Here % shapeshift is nicely used as a variable. It might be helpful to organize this by cell type, as in Figure 2A. Consider using % shapeshift in more of the analyses.

Fig. 2G This also suggests that low vs. high virion output is cell-type dependent and could influence shape (a cell-intrinsic factor).

What is the explanation for the lower increase in filaments when using entry inhibitors compared to other treatments?

174-177 Why is the post effect stronger than the pre effect?

179- Is this really accurate? When there's no inhibition, there's hardly any change, only significant minor shifts. It might help to align all Y axes for easier comparison and acknowledge the actual ranges of change.

227-228 The same (even stronger) effects were seen with Baloxavir. This supports the idea that the main effect is related to efficiency. It might be interesting to study Baloxavir kinetics.

252 I would weaken this statement. As described, shape depends on cell type and strain but is also variable under certain conditions.

287-288 his again shows that anything enhancing or weakening efficiency results in more spheres or filaments, respectively.

Do the resulting filaments escape antibody neutralization more effectively? Can this be tested using antibody titration on the resulting virions?

Have you tried enhancing efficiency with something like nucleotides, growth factors, or FCS? If so, does this shift the virus toward more spherical shapes? Temperature differences might also affect shape outcomes.

Have you confirmed how the filaments resulting from antibody treatment look via TEM?

Version 3:

Decision Letter:

Our ref: NMICROBIOL-23102764C

2nd December 2024

Dear Tijana

Thank you for submitting your revised manuscript "Rapid Increase in Filament Production During Attenuated Influenza A Virus Infection" (NMICROBIOL-23102764C). It has now been seen by the original referee and the comments are below. The reviewers find that the paper has improved in revision, and therefore we'll be happy in principle to publish it in Nature Microbiology, pending minor revisions to satisfy the referees' final requests and to comply with our editorial and formatting guidelines.

We are now performing detailed checks on your paper and will send you a checklist detailing our editorial and formatting requirements in about a week. Please do not upload the final materials and make any revisions until you receive this additional

information from us.

Thank you again for your interest in Nature Microbiology Please do not hesitate to contact me if you have any questions.

Sincerely,

Version 4:

Decision Letter:

3rd January 2025

Dear Tijana,

Happy new year and I am delighted to accept your Article "Influenza A Virus rapidly adapts particle shape to environmental pressures" for publication in Nature Microbiology. Thank you for having chosen to submit your work to us and many congratulations.

Authors may need to take specific actions to achieve <https://www.springernature.com/gp/open-research/funding/policy-compliance-faqs> compliance with funder and institutional open access mandates. If your research is supported by a funder that requires immediate open access (e.g. according to <https://www.springernature.com/gp/open-research/plan-s-compliance>) Plan S principles) then you should select the gold OA route, and we will direct you to the compliant route where possible. For authors selecting the subscription publication route, the journal's standard licensing terms will need to be accepted, including <https://www.nature.com/nature-portfolio/editorial-policies/self-archiving-and-license-to-publish> self-archiving policies. Those licensing terms will supersede any other terms that the author or any third party may assert apply to any version of the manuscript.

We welcome the submission of potential cover material (including a short caption of around 40 words) related to your manuscript; suggestions should be sent to Nature Microbiology as electronic files (the image should be 300 dpi at 210 x 297 mm in either TIFF or JPEG format). Please note that such pictures should be selected more for their aesthetic appeal than for their scientific content, and that colour images work better than black and white or grayscale images. Please do not try to design a cover with the Nature Microbiology logo etc., and please do not submit composites of images related to your work. I am sure you

will understand that we cannot make any promise as to whether any of your suggestions might be selected for the cover of the journal.

As soon as your article is published, you will receive an automated email with your shareable link. Congrats again to you and your co-authors! We are looking forward to seeing your paper published.

With kind regards,

P.S. Click on the following link if you would like to recommend Nature Microbiology to your librarian
<http://www.nature.com/subscriptions/recommend.html#forms>

** Visit the Springer Nature Editorial and Publishing website at http://editorial-jobs.springernature.com?utm_source=ejP_NMicro_email&utm_medium=ejP_NMicro_email&utm_campaign=ejp_NMicro for more information about our career opportunities. If you have any questions please click [here](mailto:editorial.publishing.jobs@springernature.com). **

We would like to thank the editor and reviewers for their careful consideration of our manuscript. We believe our manuscript has been improved substantially after considerations of points made by the reviewers and the new data collected to address the comments. We extended our analysis to 16 virus-cell combinations, and one key takeaway is that shape response is a conserved feature across IAV strains tuned to replicative efficiency of a given virus in a given cell environment. We also added experiments in support of tension changes being an acute shape trigger for transient effects, antibody binding or osmotic pressure changes, but we softened the language around tension throughout. As a result of this refocus on attenuation, we reorganized the presentation of the data to emphasize our key new insights and modified the title accordingly. We list our point-by-point responses below.

Reviewer #1 (Remarks to the Author):

The study of Partlow et al. addresses determinants of influenza virus pleomorphy. The authors developed a flow virometry approach that allows them to conveniently quantify the fraction of spherical and filamentous virions in unprocessed supernatants. Applying this method they observed that at low MOIs the released particles show a higher proportion of filamentous structures. While this effect is only seen specifically for PR8 in MDCK cells, similar increase in filamentous virions were detected upon anti-influenza antibody treatment during viral infection and assembly. The authors then show that similar results can be obtained by keeping cells in hypotonic solutions or by inhibiting viral attachment, fusion, or transcription. They conclude that all these treatments, at low infection, lead to an increase in membrane tension which favors filamentous budding of the virions. This would have the implication for virions budding under pressure to release filamentous structures, previously shown to better evade immunity. The flow virometry methods is a great tool for determining virion shape. The immediate effect of variations of MOI, osmotic or antibody pressure on the released virion shape are intriguing and offer a nice insight into so far unknown determinants of virion pleomorphy. Whether this is an effect of membrane tension, or an adaptation of the virus to low virus yield is not entirely answered in this study. While raising a very intriguing concept in influenza virus shaping and spread more data are necessary to convincingly support this model. If proven, the implications of this finding will be of general interest for scientists working on infection and immunity.

General points:

The presented concepts are highly intriguing. Is it possible to validate these findings using a broader array of virus strains and cell types? While the manuscript effectively demonstrates the antibody effect across 2 strains and 2 cell lines, it would be beneficial to also confirm the roles of membrane tension and viral replication inhibitors in this context.

Thank you for this suggestion. In the resubmitted manuscript, we introduced a broader array of virus strains and cell lines to further explore our observed trends. This allowed us to draw solid conclusions about conserved and context-specific effects as well as to identify trends that were not apparent with only a couple of strains in a couple of cell lines. We summarize those key insights in subsequent responses below. Specifically, we added virus strains A/Hong Kong/1/1968 (HK68) (H3N2) and A/California/07/2009 (Cal0709) (H1N1) and cell lines A549 (human lung carcinoma) and Caco2 (human

epithelial colorectal adenocarcinoma). Using these strains we:

- Performed time courses of infection (16 virus-cell combinations at 2 MOIs), and observed conservation of shape trends over time of infection, now presented in Figure 2.
- Performed MOI ranges (16 virus-cell combinations at 8 MOIs), and measured how shape changes for each cell-virus combination at t24 h.p.i. as MOI changes, now presented in Figure 3. MOI effects were context specific.
- Performed antibody treatment and found that antibodies induced filaments in all cell lines by all strains that at baseline display moderate to low proportion of filaments, including both newly added strains, HK68 and Cal0709. This data is now presented in Extended Data Figure 8A-D. Cal0709, like the original XUdorn in MDCK cells, is filamentous at baseline in all four cell lines, and in those cases, antibodies do not induce additional effects. However, even Cal0709 showed antibody induced effects when produced in eggs, where its baseline shape was only moderately filamentous (~15-30%).
- Performed osmotic shock treatment, and similarly recapitulated the effects by PR8 in all four cell lines (previously performed only in MDCK cells), and with XUdorn and HK68 in Calu3 cells, now presented in Extended Data Figure 8. We further showed that osmotic treatment effects are readily reversible (Extended Data Figure 8K-L).

I am wondering about virion numbers. Why do spherical and filamentous particles not differ in their hemagglutination efficiency? The measurements in Fig S1D suggest that 1 HAU refers to the same number of spherical or filamentous particles, which seems counter-intuitive. I am just wondering whether this is a fair normalization method, as variations in infectious virion numbers can affect the outcome of antibody assays.

This is an excellent question. In our previous work, Li *et al.* 2021, we performed correlations of hemagglutination and particle counting by electron microscopy with reference beads and found that the number of virions needed to induce hemagglutination is the same for spherical and filamentous fractions. Despite the large size difference between spherical virions and longer filaments, the hemagglutination assay seems to report on the ability of virions to bind to red blood cells, which at baseline does not differ for virions of different size. The reason for this might be the comparatively larger size of red blood cells, whose crowding around virions might be limiting. Indeed, antibodies that block virion attachment inhibit hemagglutination of spherical virion fractions more readily than that of filamentous fractions. This behavior resembles simple binding of virions to adherent cells in culture where crosslinking of cells is not possible, and which reveals no difference between spherical and filamentous fractions at baseline and greater sensitivity of spheres to attachment pressure.

In the current study we do not rely on this normalization method to derive particle numbers, but instead count virions directly by flow virometry, and counts by flow virometry were internally validated using comparison to counts obtained by including reference beads.

Did the authors consider in their flow virometry methods and in their interpretations that extracellular vesicles released from infected cells also have spherical shapes and might incorporate and display HA while not being infectious?

Thank for bringing this to our attention. Extracellular vesicles produced by cells may contain HA when the cell is infected and could thus be counted in our flow virometry experiments. To address whether

detection of these vesicles might muddle the interpretations of our data, we measured the presence of RNase-protected IAV gene segments in the supernatants of infected cells by qRT-PCR and correlated these values to particle counts from flow virometry. We found no significant differences in genome content between antibody-treated and untreated samples of PR8 and XUD; each in MDCK and Calu3 cells (Extended Data Figure 6B). This shows that the relative presence of extracellular vesicles is similar for conditions that produce different proportion of filaments, and thus the contribution of HA⁺ extracellular vesicles to our conclusions is absent or negligible. See also our answer to your specific point 1 below.

Is there a possibility to measure whether antibody binding to the cells affects membrane tension?

Thank you for this question, as it is also something we are actively thinking about. Measuring membrane tension and its correlation to shape is of great interest to us and has posed significant technical challenges. We believe that while this work proposes changes in tension as a trigger of filament production, demonstrating a direct effect is beyond the scope of this manuscript. As such, we have softened the language implicating membrane tension, and propose it as a hypothesis that needs further exploration. Our results and new analyses inspired by the reviews now focus more on viral attenuation as a signal for virion shape changes.

Antibodies directed against virus proteins affect the shape of the released virions. What about antibodies targeting cellular host transmembrane proteins? Does this also affect the shape? Or the membrane tension? If not, what is the proposed mechanism of antibody sensing and shape shifting by the virions?

Thank you for the suggestion. Understanding whether host-targeted antibodies alter the shape of virions would help unravel the mechanism behind antibody treatments. We tested whether an α MHC antibody affects the shape of virions, seen in Extended Data Figure 5C, and saw no effects, but further experiments would be needed to answer this conclusively.

Now included in Extended Data Figure 9 we show evidence that α HA antibodies are endocytosed by infected cells. If the antibodies are inducing endocytosis, this could attribute to increased membrane tension. There are examples of α Host antibodies inducing endocytosis in certain autoimmune diseases (such as Graves' disease) or in monoclonal antibody treatments (such as Trastuzumab). However, α Host antibodies are unlikely to be physiologically relevant in influenza infections and were thus not explored further in this manuscript.

It appears shape correlates with yield. Correlation analysis of shape vs yield throughout the data would be important. Can the authors exclude that any setting resulting in low yield determines shape? This seems to be crucial factors since also transcription inhibition affects shape. Thus far, attributing these effects to membrane tension is a too strong conclusion.

Thank you for this comment, we have given this point great consideration, particularly as we included additional viral strains and cell lines in our experiments. When we analyzed shape vs. yield for all 16 combinations of virus strain and cell line, we did not see a clear correlation (shown in Extended Data Figure 3C). However, when we instead calculated infection efficiency (as yield/input), a strong correlation appeared.

This observation inspired us to reanalyze much of our data and significantly affected the interpretation of our findings and extended the reach of our conclusions. This efficiency vs. shape trend is present throughout our data. In treatment experiments, where input is constant and output changes, this manifests as the clear yield vs. shape correlation you describe, however analyzing efficiency instead of just yield unifies observations across our strains and cell lines and for any scenario where we varied virion input.

Nonetheless, there are examples of cases where yield and shape are not correlated:

- MOI curves in Figure 3A-D. In many cases, there is no shape change with changing yield.
- In antibody treatments, some antibodies that have little to no effect on yield induce shape effects.
- In time courses, almost universally at late times, the fraction of filaments increases despite yield also increasing.
- While shape varied with efficiency almost universally, the magnitude of the response was different in different settings (e.g. the efficiency-shape curve for post treatments was generally steeper than for pre treatments).

The unifying trends and exceptions together show that shape effects are robust and specific, and strongly argue against the presence of artifacts. In general, treatments that reduce yield during infection have stronger effects later in infection (e.g. inhibition of endosomal fusion or transcription vs. attachment, or antibody post vs. pre treatment), but there are shape changes that arise independently of infection attenuation.

Please always indicate number of replicates in the experiments. Dependent on the number of replicates, consider to use SD instead of SEM. ANOVA testing is more suited than t tests.

Thank you for your suggestion to consider this more carefully. In general, we choose SEM for several reasons. 1. We are interested in the mean value of our measurement, not necessarily the spread or variability in the data. 2. SEM considers an increased accuracy with higher number of replicates, as our number of measurements ranges from three up to nine replicates.

As for statistics, we chose t-test as we are comparing treated samples to untreated controls, but not also comparing treatments to each other (which would require ANOVA).

Specific points:

The validation of virometry should include supernatants of uninfected cultured cells. Extracellular vesicle release might interfere with the assays. Optimally, cells would be additionally transfected with HA-encoding plasmids to see whether HA-carrying extracellular vesicles are detected by Sb H36-26 labeling.

Thank you for this suggestion. Uninfected cells do not produce particles positive for Sb H36-26, a control that we routinely include. To address whether EVs detected by Sb H36-26 from infected cells might be skewing our interpretations for the effects of treatments on authentic virions, we included qRT-PCT data demonstrating that their relative proportion in infected cell supernatants is not changing with treatment that changes virion shape. This argues strongly that the contribution of HA+ extracellular vesicles to our conclusions is absent or negligible.

Finally, for your own information (not included in the manuscript), we found that HA⁺ EVs produced by PR8 HA expression from a stable cell line or in PR8 infections are lower in HA content and abundance than authentic virions and are clearly separable from the virion peak based on Sb H36-26 fluorescence intensity. Importantly, they are a minor population in treated or untreated cell supernatants, and their relative prevalence is not changing strongly for different treatments. See examples of these analyses below. We chose not to include this data because it involved development of new assays and analyses that get at a question of genome packaging, which in the context of the newly added qRT-PCR controls was redundant and is a major subject of its own that will be explored in future work. Additionally, although we were not looking for these vesicles at the time, we now see that the virus gating strategy used in the manuscript largely excluded them from our analysis.

Fig 1 elegantly shows how yield increases stronger with spread as compared to the single cycle setting. To me it is surprising that already after 0 hours post infection (1 h post exposure) virus particles are released. Are these infectious particles?

We believe that the particles seen at t₀ were bound to cells but released when the culture medium is added. This could be stochastic or due to activity of NA. This is particularly noticeable in controls with ammonium chloride added at t₀, where the “unbinding” population is clearly observable at t₄ and onward, when no virus is produced by cells. We have added a sentence in the results to make this clear. Importantly, these particles were only detectable/quantifiable in high MOI cases, and the earliest measured progeny particle counts exceeded those values by at least one, and up to a few orders of magnitude (see Extended Data Figure 2).

“In contrast, XUDorn was no more filamentous than PR8 in Calu3 cells (Fig. 1, F and G), an unexpected

finding that highlights the extent to which the cellular environment of infection can direct viral assembly.” This also highlights that this observation is only based on one virus isolate in one specific cell type. Is this effect for PR8 also observed in other cells for example A549?

Thank you for this comment, which also played a large role in our extensive reanalysis. With our expanded repertoire of virus strains and cell lines (including A549), we clearly see that the combination of cell line and virus can have dramatic effects on the shape of the virus produced, and that these effects seem to depend on the efficiency of a given virus strain to replicate in the cell line.

Fig 1 At low MOIs the virus yield in Calu cells is higher ($>\log 4$) than in MDCK cells ($<\log 4$). Might this be the reason that there is no increase in filaments?

Thank you for this question as we also wondered about that. By including 16 virus-cell combinations in our time courses (new Figure 2), we were able to probe this question further and found no correlation between the low MOI yield and shape (new Extended Data Figure 3C). We instead found a striking correlation between shape and viral efficiency (yield/input) at low MOI in a given cell line (new Figure 2E). Shape of progeny virions thus seems finely tuned to the intrinsic ability of the virus to replicate in a low-yield regime. This relationship breaks down at high MOI likely due to genome interactions in co-entry events previously shown by others to be neutral, favorable, or antagonistic depending on the virus-cell combination (new Figure 2F, and we added a reference to this prior work). The effect of MOI on shape thus follows different trends for different virus-cell combinations reflecting this complex biology that is outside the scope of this manuscript (new Figure 3). Interestingly, however, low MOI either has no effect on shape or favors filaments (new Figure 3E), so it is either neutral or favorable for further cycles of infection under pressure.

Fig 2 Can you show a correlation between burst and shape? These observations are interesting, however limited by the fact that it is only seen for PR8 in MDCK.

Thank you for this comment, which inspired us to perform MOI experiments with four of our virus strains in four cell lines (Figure 3). Our results led us to conclude that burst does not necessarily predict shape across cell lines. Additionally, now with 16 virus-cell combinations instead of the original 4, it has become clear that as MOI changes, burst may increase or decrease, and this also does not predict shape. Finally, in hindsight, we realized that the precise analysis of burst critically depended on knowing the exact fraction of infected cells, which may vary from the intended value based on virion input. We didn't want to introduce needless error by using estimates from Poisson instead of measurements in each experiment, and as such, we have removed burst analysis from the paper. Importantly, we added the correlation between efficiency and shape that proved quite informative and was instead emphasized in the revised manuscript.

“We infected MDCK cells with PR8 in the presence of varying doses of a panel of antibodies specific to viral surface antigens and measured virion yield and shape after 24 hours.” These results indicate that the lower the yields are, the higher the filamentous fraction is (in agreement with Fig 2). In addition, black and blue bars also correspond to lower “MOIs” by decreasing infection rate. Could this be simply due to decreasing infection rates in the presence of the antibodies? Please also comment on how antibody addition after infection results in decreased yield.

In the blue and black bar samples ('pre' and 'all' treatments) effects are similar to effects of lower MOI for antibodies that inhibit virion attachment to cells. In fact, we agree that in general, the decrease in the infection extent by the attachment antibody is solely responsible for the shape effects. This is supported by the fact that in Calu3 cells, where decreased MOI does not change shape, there is almost no effect on shape from pre-treatment by an attachment antibody (blue bars). This supports how post-treatment (red bars) may be a distinct mechanism of shape change. We would also note that inhibition by antibodies is not always equivalent to decrease in MOI. Other antibodies inhibit fusion (which occurs after viral endocytosis) and is thus distinct from decreasing MOI. These antibodies reduce the efficiency of infection by viruses that engage cells, and their shape trends are equivalent to those derived from low MOI infections by our panel of virus strains and cell lines.

Fig 3B. Aside from shape, do the bound antibodies interfere with Sb H36-26 labeling in virometry? More validation is necessary. Fig 3D excludes this for one antibody, but possibly not for all.

We tested all antibodies for interference with Sb H36-26 labeling, and only Sb H36-26 and Sa Y8-1A6 interfered. This is why they are excluded from what is now Figure 6C, D. However, even for these antibodies, no interference was observed in supernatants at 24 hours. We believe that this is because antibodies may be internalized by endocytosis, as shown in new Extended Data Fig 9.

Fig S6 please explain in detail how it was assured that the antibody present in the medium does not inhibit reinfection. Determining infectious viral titers would help to assess how residual antibody might affect the readout.

For reinfection experiments, we infected at 3 virions per cell. We reviewed our pre-treatment data and determined the concentration of antibody attenuated infection no more than 10%. We then chose the most filamentous sample where dilution to the 3 virion/cell input reduced the antibody concentration below the target concentration.

Fig 3D. How long was waited after mixing cells into media? The figure suggests that immediately after mixing there is already virions with ~8% filaments? But in C there is no virions at 0 h.

Thank you for pointing this out to us. We recognize that this data presentation was unclear. What was previously shown as "t0" was actually t20 h.p.i., before the cells were washed into treatment. We have adjusted this figure for clarity (now Figure 5A-B).

Figure 3. How do the observed effects change with lower or higher MOIs?

Thank you for the question. Our initial experiments before the preparation of the data for this manuscript showed that antibody effects are present at a range of MOIs. However, the virus must be relatively spherical at the MOI to observe an increase in filaments. Infections that change shape with changing MOI, such as HK68 in Calu3 cells show antibody effects at MOIs that produce more spheres (see also our response to the General Point 1 above and Extended Data Figure 8). In summary, qualitatively there is no change to antibody effects with MOI, as long as the virus produced in untreated conditions is fairly spherical. For the experiments presented, we chose an MOI with high

dynamic range where attenuation still allows measurable particle counts. To illustrate this point further, experiments presented in Figure 3B used antibodies Sb H36-26 and MEDI8852 in infection of MDCK cells by PR8 at MOI 3, while PR8 infections of MDCK cells were performed at MOI 0.6 in the large antibody sweep experiment in Figure 4.

Fig 4. Again yield correlates with shape. Also needs to be confirmed by other virus/ other cells.

Thank you for this comment, and we also find it very interesting that even this transient treatment exhibits a clear yield/shape correlation. We performed osmolarity experiments with other virus strains and cell lines and found the effects to be a robust response. Now shown in Extended Data Figure 8E-J.

Fig 4D Why is there always virus yield when applying the inhibitors? Can inhibitors completely prevent yield? Determining the infectious titer of resulting particles would give more information about the efficiency of the inhibitors.

Thank you for the question. Increasing concentrations of MEDI8852 IgG or Sb H36-26 certainly completely prevents yield. We do observe a plateau when inhibiting with baloxavir, but this may be due to solubility of the chemical. For this figure, now Figure 3F, we carefully determined inhibitor treatments that reduced but did not eliminate viral yield.

How do the endocytosis inhibitors affect secretion and shape of any non-viral vesicles from the cells?

Thank you for this interesting question. It is intriguing to wonder how the treatments that effect virion shape may affect non-viral vesicles. In our revision experiments, we performed several experiments that suggest our interpretations are not skewed by presence of HA-containing EVs. Further analysis into the shape and properties of non-viral (HA⁻) EVs would require us to develop another small-particle flow cytometry assay specific for HA-negative EVs, which we feel is outside the scope of this manuscript.

What does the hyper and hypotonic treatment do to any vesicles secreted by a cell?

Another interesting question. As mentioned above, we aren't measuring HA⁻ vesicles, and can't comment on how they are affected by tonicity. We have performed additional experiments and do see the effects are reversible (Extended Data Figure 8K-L), therefore if there are cellular changes, they are not drastically and permanently changing the cell biology of virion budding.

Fig S9 should also show initial infectivity of the particles, not only progeny virus after infection with the particles.

Thank you for the comment. In this figure, now Extended Data Figure 6A, the sample labeled 'input' shows the initial infectivity of the particles.

File S1. These are good calculations, however, the release and uptake of extracellular vesicles is not considered and might change the whole system. There might be a homeostasis in the cells where endocytosis is accompanied with exocytosis. It is very hypothetical and to attribute these effects to membrane tension more convincing experimental data are necessary.

Thank you for this comment. There is probably cell homeostasis even in the context of transient treatments, and we toned this down. We have removed our calculations and file S1 as our main conclusions no longer focus on tension changes but rather on the efficiency of viral replication as the main driver of virion shape.

Minor points:

“We also reproduced previous findings that XUDorn produces a high proportion of filaments in MDCK cells for all conditions (Fig. 1E).” Provide reference.

We have added two references:

Ivanovic T, Rozendaal R, Floyd DL, Popovic M, van Oijen AM, Harrison SC. Kinetics of proton transport into influenza virions by the viral M2 channel. *PLoS One* 7(3):e31566.

Li T, Li Z, Deans EE, Mittler E, Liu M, Chandran K, Ivanovic T. The shape of pleomorphic virions determines resistance to cell-entry pressure. *Nat. Microbiol.* 6(5):617-629.

Figure S2: In the legend “D” is described but there is no panel “D”

As we have removed our burst analysis, the Poisson calculations have been removed.

Fig S4: It seems the labelling is wrong in 3 of the 4 panels. Otherwise low MOI results in higher yields than high MOIs.

This version of the manuscript no longer includes this figure.

“An exception was XUDorn infection of MDCK”. From four settings analysed, the increase in filaments is only seen in one, therefore it seem PR8 in MDCK is the exception.

We have removed our burst analysis and no longer make this point. Our expanded set of virus strains and cell lines has given us a more in depth and broad analysis.

Reviewer #2 (Remarks to the Author):

In this manuscript Partlow et al wish to explore an area of influenza virus biology that is fairly poorly studied (or at least often in large disagreement with itself). To do so they develop a flow virometry assay to sensitivity and rapidly determine the morphology of virions without a need for expensive, time-consuming and often highly artefact-filled electron microscopy (which most previous studies have used). With this assay they are able to perform time-courses of virion morphology at different times during infection and use a series of elegant experiments to show that differential membrane tension and antibodies can have a large effect on virion morphology, partially decoupling this phenotype from the genotype of the virus (which most literature has been focussed on).

This is a great study, utilising a new technology to answer some quite basic biological questions in a way that I’m not aware has been done before. I would therefore recommend it is published and believe this would be a good fit for nature microbiology. I have a couple of small points below that I think could be

addressed to improve the paper further but other than that I greatly enjoyed reading it.

1st paragraph, page 3 (and Figure S3) – “To limit infections to the initially infected cells, we omitted trypsin protease, which is required for the activation of HA on produced virions to enable further rounds of infection (fig. S3)”. While this is true for MDCKs I’m not sure either from previous literature or even the authors own data (Figure S3) this is true for Calu-3. Calu-3 express a range of serine proteases (such as TMPRSS2), many of which have been described to activate influenza haemagglutinin in the absence of exogenous trypsin. I don’t think this really changes the conclusions much but the authors should alter their wording here.

Thank you for this comment catching an oversight on our part. It is true, even in our data, that there is partial activation of HA in the Calu3 cell line. For this reason, no-spread infections plotted in Figure 1F and G now plot infections where spread was limited by addition of ammonium chloride at t4, ensuring we are preventing spread. Additionally, in all our new experiments with additional virus strains and cell lines, we include ammonium chloride at t4 h.p.i. to prevent any cell line/virus specific HA activation from clouding our results.

One issue that the authors appear to have slightly overlooked – filamentous virions are unlikely to be able to enter cells via typical endocytosis pathways that spherical virions use, instead being more restricted to macropinocytosis – this will also have a large impact on membrane tension in cells infected by these filamentous virions as, I would assume macropinocytosis is much less efficient per virion for entry per the amount of membrane needed? It would be interesting for the authors to discuss this as it seems, by this route, infection filamentous virions could bias downstream progeny to be more likely to be filamentous?

We thank the reviewer for this insightful comment. The reviewer is correct that filaments enter cells by obligatory micropinocytosis, and, indeed, in our previous work we demonstrated that the infectivity advantage of filaments over spheres when fusion is inhibited by antibodies can be abrogated by inhibiting micropinocytosis (Li et al., 2021). To address this, we infected MDCK cells with the same number of spheres or filaments and measured shape of progeny virions after yields were reduced with baloxavir but didn’t see strong effects. We do not know if the details of our preliminary experiment design precluded us from seeing strong effects, but our new experiments and analyses changed how we now interpret the effect of early infection attenuation on shape. Our new analyses unify shape properties of 4 virus isolates across 4 cell types and show striking correlation between efficiency (virion yield divided by the input virion number) and shape (new Figure 2). We further observed that early attenuation of infection past virion attachment (original data, now presented as Figure 3F) results in same dependence of shape on yield as that of shape on efficiency at low MOI, where large changes in membrane tension are not induced by entering virions (compare Figures 2E and 3F). We thus think that early attenuation of infection might trigger shape changes by a different mechanism from the late, acute events, where we think changes in membrane tension might play a bigger role. Shape differences and the role of membrane depletion by the initiating particles may or may not contribute to this early effect, but defining this unambiguously is outside of the current scope and will be explored in future work. We rearranged our results to better describe new insights that correlate efficiency (yield/input) and shape (new Figure 2) and toned down our conclusion for the role of membrane tension changes as the early trigger.

Partlow *et.al*, Influenza A Virus Rapidly Assembles Filaments in Response to Attenuation and Antibodies

NMICROBIOL-23102764B Response to reviewers.

Nature Microbiology

Reviewer Expertise:

Referee #1: molecular virology

Referee #2: influenza, virus entry

Reviewer Comments:

Reviewer #1 (Remarks to the Author):

Partlow and colleagues tackle the really challenging problem of finding a mechanism for how and why influenza virus changes shape during replication and varies between variants, cells and under external pressure. They use the very innovative virometry method to determine percentages of spherical or filamentous virions released by cells. They conclude that under environmental pressure especially antibodies lead to influenza adapting by producing more filamentous virions. They interpret that this effect might be attributed to changed membrane tension potentially due to antibody binding potentially followed by endocytosis and similar effects seen under osmotic changes. While this manuscript is highly innovative and offers intriguing insights into the pleomorphic nature of influenza virus, also alternative explanations could lead to the observed effects. To make the authors' conclusions clearer, the manuscript would benefit from more streamlined and clearly presented experiments. The current setup and reasoning are somewhat hard to follow due to the topic's complexity. This paper is highly interesting and important, but in its current state, it might not capture the attention of a broader audience.

Thank you for your detailed and thorough evaluation of our work. You took great care in understanding and exploring our data, and this has helped us improve the quality of our manuscript. Specifically in these revisions, your comments on the clarity, complexity, and readability of our manuscript have helped to refine our presentation and make our story easier to follow. In addition, we have included new experiments to address uncertainties in our previous version. One addresses the speed of viral genome replication and how that may affect

shape, and another addresses whether antibodies may cause the virions produced by an infection to be selectively endocytosed by cells in a size-specific manner. These experiments are described in detail, along with our point-by-point responses, below.

General points:

The manuscript would strongly benefit from improving readability and making the experimental results and their meanings clearer. Currently, understanding the text and figures requires extensive knowledge of virology and influenza, and even then, one must think very hard to follow.

Thank you for pointing this out. We have now made major changes in 4 out of 6 main display items (Figs. 2-4, and the Model Fig. 6) mostly aiming to improve clarity of presentation. Aided by the refocus provided by the figure rearrangements, we have greatly simplified the flow in the manuscript text. One key point that we worked to clarify in this revision is that our results reveal two distinct triggers of shape changes, one which is attenuation-dependent, efficiency vs. shape relationship, now the focus of the first part of the manuscript (Figs. 1-4, with PRE effects in Fig. 4), and one which is attenuation-independent, and the focus of the second part of the manuscript (Fig. 4-5, with POST effects in Fig. 4). Part I - We dissected the efficiency-shape relationship with targeted inhibitors of defined early steps in infection to show that MOI is not a universal predictor of shape, but the efficiency of any step after attachment and before genome expression influences shape (Figs. 2-3). Part II - We dissected the late effects to show that they are fast, potent (even for an established infection after robust genome amplification) and decoupled from the extent of attenuation or are attenuation-independent. Finally, we simplified the title while making sure it emphasizes both key sets of our findings – shape tuning in response to attenuation, and shape tuning in response to antibody binding.

Display item changes:

In Fig. 2, we now digested the results of 32 infection time-courses (4 virus strains, in 4 cell lines, at 2 MOIs), 16 MOI-range curves (4 strains in 4 cell types), and the new RT-qPCR experiments, in a way that focuses on the relationship between individual infection parameters and shape. All the detailed data from which these key trends derive have been moved to Extended Data Figs. 3 and 4.

Fig. 3 was rearranged to make key results easier to follow and interpret. We included a diagram outlining the steps in infection that we inhibited systematically in this experiment. We included the more standard inhibitor-titration curves and their effects on yield. Finally, instead of plotting the less-intuitive yield-shape curves with data averages, we highlighted on the inhibitor titration curves the yield regimes that we then compared for shape and where we included all individual data points. The experiment where we altered the infection efficiency in

a controlled way by changing the virion input and keeping the virion output constant was similarly replotted.

In Fig. 4, we now compressed the results of more than 750 infections (2 virus strains, each in 1 or 2 cell lines, treated with at least 6 different concentrations of 12 or 9 different antibodies including untreated, before or after entry) into 11 plots, each displaying two groups of datapoints (infections treated during or after entry) showing the effects of each antibody type (based on the viral antigen they target). All the detailed data separated per antibody and including the condition where antibody was present throughout the infection have been moved to Extended Data Figs. 6-8.

Fig. 6 was revised to 1) summarize our key conclusions (Fig. 6A), 2) describe our model (Fig. 6C), and 3) emphasize the significance of our findings. We challenge the current assumptions that shape is genetically tuned and emphasize the key consequences of phenotypic tuning revealed by our work on evolvability of viral populations (compare Figs. 6B and 6C).

Can the following alternative explanations be disproven?

- Speed: A high replication speed can explain lower number of filamentous virions. A fast virion assembly and budding might increase yield depending on high MOI, or viral determinants (fast polymerase) or cellular determinants (higher metabolism, less restriction proteins, not perfectly adapted to the species or osmotic stress).

We have added a control experiment that tests whether viral replication rates are related to shape in our experiments. We measured the cell-associated viral genomic RNA immediately following attachment (t_0), and after 4, 20, or 21 hours of infection (t_4 , t_{20} , and t_{21}). The full curves are shown in (Extended Data Fig. 3E-F), and the correlation of average replication rates with shape in Fig. 2F (average overall rate; $t_{20.5}/t_4$, where $t_{20.5}$ is the average value of t_{20} and t_{21}), Extended Data Fig. 3G (initial rate; t_4/t_0) and Extended Data Fig. 3H (late rate at time of shape measurements; t_{21}/t_{20}). There was no correlation between average replication rates (overall, initial, or late) and shape.

Inhibition of replication (baloxavir) can induce shape changes, but so can inhibition of membrane fusion (MEDI8852 IgG) to the same extent (Fig. 3E). The effect of fusion inhibition was corroborated by the large antibody-sweep, now summarized in Fig. 4. Finally, after unperturbed infection was established and genome replication reached steady state (Extended Data Fig. 3H), treatments had the most pronounced shape effects (Fig. 4).

- Release: A higher viral release leads to more spherical ones. Or a low release to more filamentous ones. This is closely tied to fast replication but focuses more on release. Maybe some cells/virus strains give the virus more time to assemble before release thus resulting in more filaments.

This is possible but not a mutually exclusive explanation. We predict that changes in membrane dynamics directly modulate release. Some of our data suggest that assembly is not limited by the number of genomes available. Our new viral genome RT-qPCR experiments in Extended Data Fig. 3E show that genome replication can be faster than particle assembly. In our osmotic pressure experiments, hypertonic shock elicits a fast burst of particles released (Extended Data Fig. 10D), which are just as infectious as those from untreated cells (Extended Data Fig. 10G-H). This suggests that release, and not genome packaging is the rate limiting step. We believe that for many of our late effects, the release step is precisely what is being regulated. We have added these points to our discussion section. See lines 287-290.

- Spread: More spread leads to more spherical ones (Fig 1). Antibodies prevent spread thus we have more filamentous ones in the presence of antibodies (Fig 5).

Spread can lead to more spheres by increasing MOI of subsequent infections, but only for low MOI cases where shape is sensitive to MOI – see Fig. 1 and Extended Data Fig. 2. For the basic two strains, PR8 and XUdorn in the two basic cell lines, MDCK Siat1 and Calu3, at either high or low MOI, the increase in spheres as a function of spread was limited to a single case, low-MOI PR8 infections of MDCK Siat1 cells.

We have made text changes to make it explicit when spread was permitted or disallowed in each experiment. In summary, both conditions were included in Fig. 1 and Extended Data Fig. 2. Spread was prevented in experiments in Fig. 2 (see Results text, line 109) and Fig. 3 (see Results text, line 128 and 140). Spread was only partially prevented in Fig. 4 - trypsin was not included, which effectively prevents spread in MDCK Siat 1 cells (Fig. 4A) and severely reduces spread in Calu3 cells (Fig. 4B-C) (See Extended Data Fig. 2) (see Results text, line 162). In general, for experiments in Calu3 cells, omission of trypsin is not sufficient to completely prevent spread unless ammonium chloride is also added, but it is expected to be minimal (Extended Data Fig. 2C,G, compare black open circles to pink). This only, however, affects experiments in Fig. 4B and C. It is also worth noting that the shape of virions produced by PR8 infections in Calu3 cells appears to be independent from MOI, as shown in Fig. 2E and Extended Data Fig. 4A, right, red. Overall, results were consistent when spread was blocked (PR8 in MDCK cells) and where slight spread was possible (PR8 and XUdorn in Calu3 cells).

Spread was not prevented in Fig. 5, but the potential for its contribution to those measurements was excluded in Extended Data Fig. 9E-F. Importantly, transient antibody treatments completely decouple yields and shape effects of antibody binding to infected cells, where shape but not yield changes are apparent (Fig. 5A-B). A newly added control eliminates the possibility of particle reuptake and thus spread contributing to yield and shape measurements in the transient antibody treatments (Extended Data Fig. 9E-F) (See also our next response).

- Reuptake: Antibodies could lead to reuptake of the small virions which increases the relative number of filamentous virions in the supernatant.

A newly added control eliminates the possibility of particle reuptake contributing to yield and shape measurements in the transient antibody treatments (Extended Data Fig. 9E-F). For this, we repeated the transient antibody treatment experiment where we added a fixed concentration of fluorescently labeled PR8 virus together with the treatment antibody. We then separately measured the shape of released and added virions after 2 hrs (Extended Data Fig. 9E-F). While the proportion of filaments among released virions increased, the proportion of filaments in the added virus population remained the same. Importantly, the total virion counts in the added virus population remained constant. See Results text, lines 211-221.

Do you have an explanation for why, in general, first more filamentous, then spherical and then an increase in filament production is seen? How does this fit to membrane tension? Or to replication efficiency/speed?

One way this could fit membrane dynamics idea is that those steps in the viral life cycle are dominated by membrane depletion (endocytosis), replenishment (exocytosis), and depletion again (budding when replenishment slows down late in infection. Indeed, this interpretation is consistent with combined insights from high-MOI infection time-courses of MDCK Siat1 cells by PR8 and XUd (Fig. 1D,1E, Extended Data Fig. 2B, 2F) and our new RT-qPCR data (Extended Data Fig. 3E). We detail this discussion here, but do not include it in order to remain focused on key findings that are more definitive. A big open question that remains in this context is how are effects on membrane homeostasis early in infection propagated to shape effects late in infection. This will be a subject of future inquiry.

Briefly, by 4 h.p.i. the only virus detected in the supernatant is the small amount of input virus which fell off the cells and failed to infect. The identity of the t4 virus subpopulation is confirmed by the experiments where ammonium chloride was added before infection to completely block it – the t4 virus yield in that case was the same but then remained unchanged for the duration of the time-course (Extended Data Fig. 2B, 2D, 2F, 2H red curves). 4 h.p.i. is also the time point, which revealed the greatest extent of genome replication in MDCK cells (Extended Data Fig. 3E). The subsequent 4-12hrs witnessed the greatest release in budding virions (Fig. 1D,1E, Extended Data Fig. 2B, 2F), likely a product of great membrane replenishment as new protein synthesis was at its peak. During this time the rate of genome production stabilized at a lower overall value, and with it likely the rate of membrane replenishment (Extended Data Fig. 3E). Late in infection, budding rate exceeded that of genome amplification, which likely skewed the balance in favor of budding relative to membrane replenishment.

It should become clear whether it is generally talked about the 24 h time point. Or whether the dynamic is analyzed. It gets a little confusing because of this complexity. Is the effect of attenuation on shape observable early or late or both?

Thank you for pointing this out, we have adjusted the text to make this clear. Dynamic is analyzed only in the final section of the paper titled ‘Antibody-induced shape dynamics are fast and attenuation-independent,’ and we made that clear in the opening sentence of the section, see lines 202-205. In all preceding instances that were not time-courses, measurements were done at t24 hours post infection (MOI range in Fig. 2E, all experiments in Figs. 3 and 4), and the text makes that clear in each case. See lines 117, 128, 159.

In all experiments it is very important to mention whether spread is excluded or not.

We have adjusted the text to make this clear for each experiment, detailed above.

The discussion can be longer and address a lot of the uncertainties.

We have expanded the discussion and refocused on our most important observations. Specific points are mentioned in other responses.

Specific points:

The introduction says that in animals filamentous virions are favored. How is this with the new cow-infecting viruses (just a question out of curiosity for potential follow-ups)? Instead of external pressure, this might also be due to slow replication/low yield/low release in different species?

Our statements on the shape of virions in animal infections are based on cited past published literature in the introduction. Because of the high particle concentrations required for electron microscopy, many reports on the shape of animal viruses are actually reporting the shape of animal-derived virus after one or several passages in tissue culture or in eggs. We are working on getting the needed biosafety approvals to address shape questions for H5N1 influenza in cows and for samples from human infections. However, both of those directions are outside the current scope due to both the approvals required and the optimization of such samples for flow virometry.

Fig 1C with and without spread is not defined at this time in the results text or the figure legend.

We have now defined this in the text, see next response.

Line 91 the term “spread” should explicitly stated here in the context of trypsin as it is used throughout the text when trypsin is not added.

Thank you for this comment. In this work we inhibit spread by omitting trypsin when sufficient (such as in MDCK Siat1 cells) or adding ammonium chloride at 4 h.p.i. when needed (as in Calu3 cells). We believe that specifically mentioning trypsin or NH4Cl for each experiment may be too technical for a general audience. In the current manuscript version, we describe how we prevent spread in the Fig. 1 legend (see lines 342 and 345) as well as in the methods, and point the reader here early in the text, see line 87. We also demonstrate effectiveness of inhibiting spread in Extended Data Fig. 2. Then for simplicity in the text, we simply state whether spread was allowed or disallowed, ensuring this is clearly stated where relevant.

Fig 1: On Calus we see the typical filamentous – spherical – filamentous phenotype. On MDCK cells we see more filamentous particles for PR8 at low MOI without spread, and more filamentous particles for XUdorn in general. Could the interpretation also be: it is not extrinsic factors but depends on cells or viral strain?

We believe that shape does depend on cell and virus strain, but that it is not defined by either. Additionally, if different cell lines cause the same virus to produce different shape particles, then from the perspective of viral biology, that is an extrinsic factor. See lines 95-96. None of our viral strains are always spherical or always filamentous, but demonstrate cell and strain dependencies. In this manuscript we try to isolate what factor is determining this. We agree that there are some cell lines and some strains that yield more filaments or more spheres than others, however, we hope that Figure 2 now makes it clear that those rules are not universal. Instead, we see a very clear correlation of shape with efficiency. See also lines 111-124.

Fig 2A-D: We mostly see the typical filament-sphere-filament phenotype. On cacos there is always filament increase without the typical drop. H1N1 starts with a high filament levels. Again a conclusion might be that it depends on strain or cell or a combination of both.

Cell and strain combinations certainly affect shape, but while there are general trends with various cell lines or virus strains, they all break down for certain combinations. However, what seems to be universally true (at least for the 16 virus-cell combinations, and for inhibition of early infection steps) is the relationship between viral efficiency and shape. Additionally, it is also universally true that shape is dynamic over the time-courses of infection. By moving these additional time-courses to Extended Data Fig. 3, we hope to focus the reader's attention on the quantitative analysis of the trends or lack thereof detailed in the new Fig. 2. We also removed the specific statement quoted above on the characteristic curve shapes so as not to distract from the bigger points. We instead comment that shape is dynamic,

which is a big and surprising point in-and-of-itself. The readers interested in analyzing shape curves in infection time-course more closely can do so as the complete data is now included in Extended Data 3A-D.

Fig 2E: If understood correctly, the efficiency is defined as $\text{yield/cell} = \text{yield/MOI}$. This would also correspond to a fast replication/fast release of particles. This would also make sense as fast assembly and release of virions with only limited available genome/protein material would result in small virions (=spheres).

Efficiency is viral particles produced/viral particles input. Since particles/infectious unit varies from ~13 up to ~230 with virus and cell, efficiency doesn't exactly equate to either yield/cell or yield/MOI. As mentioned above, we now measure genome replication in cells (Extended Data Fig. 3E-F), and find no correlation with shape (Fig. 2F, Extended Data Fig. 3G-H). We additionally find evidence in our osmotic shock experiments that particle release, not genome or protein incorporation, may be the limiting step in assembly (Extended Data Fig. 10D-E).

Extended Fig 3B: Isn't there a (weak) positive correlation between infectivity and shape?

There is a subtle correlation of infectivity with shape, although this may not be independent from the much stronger relationship of efficiency and shape. Infectivity (virions/IU) determines the input required to reach a given MOI. Therefore in our experiments, virion input = MOI x cell number x infectivity. Similarly, yield is directly related to the virion output of the infection. Thus efficiency (virion output/virion input) is directly proportional to yield/infectivity. So, the subtle correlation in 2C combined with the lack of correlation in 2D produces the strong correlation in 2G. This tells us that the overall efficiency of infection may determine shape, rather than attenuation at any one specific step.

Fig 3: this nicely indicates that the “proportion of filaments either decreased or remained constant with increasing MOI”. Fig E is a little confusing as it shows exactly this, but in the text describes the opposite “that increasing filaments with lower MOI”. It is both correct, but could be phrased easier. Again, could it be explained by: high MOI = fast replication/fast release = more spheres/less filaments?

We have made text and figure changes to make the flow easier to follow and understand. See specifically lines 168-170. We believe our new experiments suggest that replication speed is not a critical factor in shape, as detailed in other responses (see for example our response to the big picture comment 1: “• Speed: A high replication speed can explain lower number of filamentous virions” among others.

Fig 3F: This graph is difficult to interpret or compare as the shape on Y and yield on X axis has

never been shown before. Does yield correspond to inhibitor concentration? How does absence of inhibitor look like? Why is there always a yield in the presence of the inhibitors? This is unclear to me. But again results would fit to: low MOI/attenuated and slow replication = more filaments. (more time for a virus to assemble more and more proteins before release).

Thank you for this comment. We agree that these data could have been much more simply represented, and we now display it as shown in Fig. 3 and Extended Data Fig. 5. It should now be clear that the yield-shape relationship breaks down for different types of treatments in the same range of yields. Absence of inhibitor is now shown as the point on the Y axis (Fig. 3B-D, F, and Extended Data Fig. 5 A, C), and the shape without inhibitor shown and labeled “untreated” (Fig. 3E, G, and Extended Data Fig. 5B, D). There is always yield with inhibitors because we titrated specific doses to reduce yield but allow sufficient particle concentration to reliably measure shape.

Fig 3F: the difference between the inhibitors and what conclusion can be drawn from it is not entirely clear to me. I agree that attachment inhibition best resembles low MOI, but would this mean less spheres? Another illustration of the graph would be helpful e.g. showing shape vs inhibitor concentrations (and also absence of inhibitor). Also here: high yield/high speed = more spheres.

Thank you. We have greatly simplified these in new Fig. 3, as well as included a diagram of the viral life cycle to orient the action of the inhibitors. We ensured absence of inhibitor was included in all plots as well. It should now be clear in Figs. 3E, 3G, and Extended Data Figs. 5B,D that significant differences in filaments can arise from infections with the same yield if infection was attenuated before versus after attachment (Fig. 3E) or for different input virion numbers (Fig. 3G, and Extended Data Figs. 5B,D).

Fig 3G: This experiment would also be easier to understand when the inhibitor concentrations would be plotted.

See above comments. This is now Fig. 3F-G, Fig. 3F is inhibitor concentrations vs. yield, and 3G is shape for either untreated or the highlighted range of treatments.

Could speed be measured by genome numbers per cell?

As described above, we performed this experiment by measuring relative genome numbers in the infected cells at various times post infection to assay replication speed of the virus (Extended Data Fig. 3E-F). We found that genome replication seems to lead ahead of particle yield, and likely not be rate-limiting. Likewise, we found no obvious correlation to replication rate to virion shape (Fig. 2F, Extended Data Fig. 3G-H).

Fig 4: Is this done with or without spread? If with spread: do the antibodies inhibit spread? This would again recapitulate a “lower MOI” and explain higher filaments.

In the antibody sweep experiments summarized in new Fig. 4 with all data presented in Extended Data Figs. 6-8. We omitted trypsin but did not include NH₄Cl. For PR8 in MDCK Siat1 cells, this treatment effectively prevents spread, as confirmed by our controls in Extended Data Fig. 2. For Calu3 cells, there may be some spread, but it should be minimal without trypsin (Extended Data Fig. 2C,G, compare black open circles to pink). It is also worth noting, however, that the shape of virions produced by PR8 infections in Calu3 cells appears to be independent from MOI, as shown in Fig. 2E and Extended Data Fig. 4A, right, red.

Fig 5AB This is a very nice and important experiment. It might show that spread is not responsible for the observed effects?

Thank you, we agree that the rapid nature of these effects make it unlikely that spread is involved. We also included a new experiment that confirms absence of reabsorption of released virions and therefore spread (we used the labeled, added virus as a surrogate for released virus, and show that virion counts and shape distributions of the added virus remained constant for at least 2hrs) (Extended Data Fig. 9E-F). Additionally, MDCK Siat 1 cells are unable to activate PR8 virus, so no spread is expected in any PR8/MDCK Siat1 experiments regardless (this includes all the experiments in Figs. 4A and 5).

Fig 5CD. This again could be explained by: slower or higher release of virus/slower higher replication / meaning faster or slower assembly.

We believe that these data suggests that particle release is the rate-limiting step (as opposed to genome or protein incorporation). In hypertonic media, the increased release of particles (Extended Data Fig. 10D) without a loss in infectivity (Extended Data 10H) suggest that genomes and protein incorporation are not rate-limiting for these conditions. We also believe that this release step is when late effects are influencing shape, such as the presence of antibodies during assembly. We have added some additional discussion to address these points, see lines 287-290. We have additionally moved conditions where we oscillate between different osmolarities to the main figure (Fig. 5F) to emphasize the rapid and reversible nature of this effect.

Maybe antibodies lead to re-uptake of small virions, but not the larger ones? This is why the fraction of larger virions is found in the supernatant. This would also fit to Extended Fig 9A.

Antibody dependent internalization is a well-documented feature of entry of some viruses, although the cell lines used here are not expected to express the Fc receptors normally responsible for this effect. To test this formally, we performed transient antibody treatments as in Fig. 5A-B where we also added a fixed concentration of AF488-labeled PR8 virus that was ~50% filaments and 50% spheres. The AF488 label allowed us to measure the shape and concentration of the input virus separately from the newly produced virus. After 2 hours, we measured the concentration and shape of the AF488 virus for a range of MEDI8852 IgG concentrations and found almost no depletion of the input virus (Extended Data Fig. 9E) and no shape preference of either spheres or filaments (Extended Data Fig. 9F).

Maybe doing the antibody assay with or without trypsin or ammonium chloride could help solving this?

Thank you for the suggestion. We hope you agree that as now written, the focus of this manuscript is on rapid effects that occur within a single cycle of infection. We have included new controls, experiments, and clarifying text that exclude spread as a significant factor in our observations which are detailed in above responses.

Replies to rebuttal:

Thank you for the suggestion. Understanding whether host-targeted antibodies alter the shape of virions would help unravel the mechanism behind antibody treatments. We tested whether an α MHC antibody affects the shape of virions, seen in Extended Data Figure 5C, and saw no effects, but further experiments would be needed to answer this conclusively.

Now included in Extended Data Figure 9 we show evidence that α HA antibodies are endocytosed by infected cells. If the antibodies are inducing endocytosis, this could attribute to increased membrane tension. There are examples of α Host antibodies inducing endocytosis in certain autoimmune diseases (such as Graves' disease) or in monoclonal antibody treatments (such as Trastuzumab). However, α Host antibodies are unlikely to be physiologically relevant in influenza infections and were thus not explored further in this manuscript.

This could also indicate that only antibodies that target viral structures interfere with spheres. Meaning that it is not membrane tension but targeting the virion and reducing spread or increasing uptake or similar effects.

The question of whether antibodies targeting host proteins can induce shape changes in influenza A is a challenging question to address as there are likely no transmembrane cell proteins expressed at similar levels or at similar densities to influenza surface proteins on the infected cell. However, we would also point out the converse result present in this manuscript: M2 antibodies also have shape effects, though often this occurs without inhibiting virus

replication (Fig. 4A-C, third column). We also see filament induction without significant yield changes in certain other circumstances, such as HA base-binding antibodies in XUDorn infections of Calu3 cells (Fig.4C, second column). However, it is possible that only viral proteins have this feature, either because of expression levels or densities or that it might be an evolved viral response to pressure to reducing antibody concentration through endocytosis. Regardless, we see that the effect is general and not limited to a specific viral protein, a specific way of inhibiting assembly, or even to inhibiting assembly at all (M2 antibody effects in Fig. 4, effects of HA base binders in XUDorn infections of Calu3 cells in Fig. 4C, or transient HA antibody effects in Fig. 5).

Reviewer #2 (Remarks to the Author):

In general my points have been thoroughly addressed, and, to my interpretation, I believe the concerns of the other reviewer have also been addressed (although I will defer to their own interpretation on this).

We are glad that you feel our responses have addressed all concerns and greatly appreciate your attention and efforts towards helping us to publish this work.

I enjoyed re-reading through the paper, believe the additions have improved it from the first submission and still believe it has a lot of merit and would recommend its publication with the following minor textual addition:

My one additional point after re-review is that the authors are now using a pair of virus isolates (rather than the laboratory strains/attenuated reassortants they previously used). This includes the use of the pandemic strain A/Hong Kong/1/1968 (HK68). My understanding is there is variability in different regions about whether this qualifies as a virus that should be used at CL2 or CL3 (with most places moving towards restricting the virus to CL3-only use due to a lack of cross-reactive immunity in the population). It would be helpful, with the addition of these new viruses, if the authors included a biosafety statement at the start of the materials and methods to describe what containment level this work was done at, and what permissions were sought, and by which authorities to use these virus isolates (at either CL2 or CL3).

We have added this statement in the Methods, under Viruses.

“Work as described with IAV strains A/Puerto Rico/8/1934, A/Udorn/307/1972 containing the A/Aichi/1968 (X31) HA segment, A/Hong Kong/1/1968, and A/California/07/2009 was approved by the Institutional Biosafety Committee (IBC) at the Biosafety Level 2 (BSL-2; Registration# RD-23-IV-10).”

Reviewer Expertise:

Referee #1: virus entry, virometry

Reviewers Comments:

Reviewer #1 (Remarks to the Author):

The revised manuscript by Partlow et al. is greatly improved in terms of clarity, presentation, and supporting experiments. The studied problem is highly complex, making the evaluation and interpretation as well as the presentation of the data challenging.

First, I believe this manuscript advances the field both methodologically and scientifically. However, I still have some doubts regarding certain conclusions and interpretations. What the paper elegantly shows is that shape shifts during replication and varies depending on the cell line, viral strain, MOI, and environmental pressure. Thus, shape-shifting appears to be an adaptation of the virus to its environment (host cell and external pressures), which is not genetically encoded but variable. However, my conclusion is that it all depends on viral "efficiency," as the authors call it, and they demonstrate a nice correlation. My hypothesis would be that anything affecting efficiency also affects viral shape. This could include the cell in which the virus replicates, antibody pressure, antiviral pressure, osmotic pressure...

What I do not see is that antibody pressure, which does not affect efficiency, results in a shape shift. Thus, shape appears to be simply an effect of viral efficiency. In the end, this may help the virus remain in the host, continue spreading, and escape neutralizing antibodies. If the authors address this concern and tone down the narrative that antibody pressure directly induces filaments that evade antibodies (and put it in the discussion), it will strengthen the paper.

General points:

It seems to me that "efficiency" is the decisive factor, which varies between cells, viral strains, and environmental factors such as pressure from antibodies, replication inhibitors, or osmotic stress. This means that something binding to the virus but not changing efficiency would not result in shape-shifting. In the case of M2 antibodies, this appears to be true: only when M2 antibodies have antiviral activity does a shape

change occur. For those where they don't affect replication, the shape remains unchanged (e.g., EFig 8 O19; EFig 7 A Ab65; O19). In EFig 6, there are minor effects without inhibition, but these appear negligible compared to other settings.

To me it remains inconclusive if shape shift is a response to antibody pressure. While in the end it might help the virus spread further, it seems to be a consequence of replication and release rather than directly related to antibody pressure e.g. by membrane tension or similar effects. This should be discussed more clearly, and maybe only late in the discussion. The text should emphasize the main findings and shift away from focusing on antibody pressure.

We are thrilled that the Referee #1 was pleased with our most recent revision to improve clarity and presentation and add supporting experiments. We thank the referee for recognizing the importance of the methodological and scientific contributions of our study. We understand that the referee remained unconvinced by our conclusions that antibodies have a second, attenuation-independent effect on shape. As a result, we have either removed or toned down these conclusions throughout the manuscript. We instead solely emphasized the finding that influenza shape is influenced by the environment of infection and the infection efficiency as the key inducer of viral shape changes.

Specific points:

The terms "band" and "smear" appear suddenly in the figures without further introduction. A brief explanation in the results section would help clarify this.

Thank you for pointing this out. Even though the terms were defined in the figure legend, we agree that they were needlessly confusing. We have simplified those terms to only state if the samples were enriched for spheres or filaments. Additionally, the Results Section 1, line 60 state that samples analyzed in Fig. 1B, and Extended Data Fig. 1 were fractionated to enrich for spherical or filamentous shapes.

Fig 1 Consider using the same Y-axis scale across the panels to avoid making small shifts appear disproportionate.

Thank you for this suggestion. We have adjusted the Y-axes in panels F and G to facilitate direct comparisons to panels D and E in the extent of shape shifts.

Lines 111-115 I am not entirely convinced by these conclusions. It seems that the cell line is the most crucial factor in determining shape, with variation between viruses within each cell line. For example, MDCK cells are always more filamentous and Calu3 cells more spherical. Similarly, PR8 is typically more spherical, and Xudorn is more filamentous, as described in the literature. Perhaps normalizing within each cell line, setting one virus to 100%, and then showing how the others deviate would clarify this. While the final conclusion that no single factor is decisive is correct, cells and strains do predetermine the extent and range of % shape.

We agree with these comments in a general sense and have revised text to make key points clear. See text, lines 96-99, 109-110. For example, we clearly recognize cell of origin as a factor, but not a lone factor in determining shape. This is not surprising as some cells will be widely permissive or nonpermissive to multiple IAV strains resulting in high or low efficiencies across the board (e.g. Calu3 and MDCK cells, respectively), but we also identified exceptions (e.g. PR8 in A549 or Caco2). We disagree that normalizing % filaments data separately within each cell line would help clarify things but think that it would rather conceal critical data on the actual percent of filaments. Furthermore 100% would refer to numbers ranging between 25-60% filaments, so it seems arbitrary and potentially misleading.

Fig 2C: Including Spearman's r and p -values would strengthen the statement that there is no correlation. Visually, it seems there might be one (higher infectivity = more filaments). Again, the clustering of cells highlights that they are the most determining factor.

Thank you for suggesting performing Spearman's correlations for these data. After learning about Spearman's, we agree this is the appropriate test for evaluating our correlations, rather than a log-linear fit. We have replaced log-linear fits of our data with the r and p values produced by Spearman's analysis in Fig. 2. Indeed, there is a slight correlation ($p=0.042$) between infectivity and shape (Fig. 2C). Though the dominant correlation ($p<0.0001$) remains between efficiency and shape (Fig. 2G). We have revised the text to acknowledge absence of correlation (Fig. 2D, 2F), or presence of weak or strong correlations where appropriate, text lines 100-109.

Fig 2D How was the shape of Cal on A549 determined here (it wasn't in Figures 2A/B)? Also, why are only two Caco results shown?

Thank you for catching this, which resulted from an error in the formatting of the figure from those data. The point formatted as Cal0709 in A549 cells belongs to Cal0709 in Caco2 cells. The same error resulted in another Caco2 point being unplotted. This figure has been corrected, but the key conclusions remained unchanged.

Fig 2F Performing these correlations is challenging because, as seen in Figure 2A, shape percentage already depends on the cell type. Comparing MDCK and Calu3 % shapes, for example, will not correlate. Possibly, a variable like % "shapeshift" would be more comparable.

From our genome replication time course (Extended Data Figs. 3E-F), we performed several analyses to determine whether genome replication speed was a determinant of shape, Fig. 2F and Extended Data Figs. 3G-H. We found no such correlation as determined by Pearson's test. It is not obvious to us how to normalize the data in a way to show % shapeshift that would reveal a trend we are missing, and don't believe that this control experiment produced enough data to warrant these further analyses.

Overall, while Calu3 cells produce more spherical populations than MDCK cells, our qPCR experiments clearly showed that MDCK cells amplify viral genomes far more rapidly (Compare Extended Data Figs. 3E and 3F). Therefore, we do not believe that genome replication speed is a key determinant of shape.

Fig 2E Here % shapeshift is nicely used as a variable. It might be helpful to organize this by cell type, as in Figure 2A. Consider using % shapeshift in more of the analyses.

We have replotted data in Fig. 2E to organize it by cell type as suggested.

Regarding the suggestion to use %shapheshift in more of the analyses: In Fig. 2E we show % shapeshift in a well internally controlled context, where the shapeshift is a result of changing MOI for each cell-virus combination. In other analyses of categorical data, such as comparing XUdorn low MOI in Calu3 to PR8 high MOI in MDCK, it is not clear how a % shapeshift metric is more informative than sharing the underlying shape data. We therefore chose to continue to express our shape data as % filamentous in those other analyses.

Fig. 2G This also suggests that low vs. high virion output is cell-type dependent and could influence shape (a cell-intrinsic factor).

This is not surprising as some cells will be widely permissive or nonpermissive to multiple IAV strains resulting in high or low efficiencies across the board. So Calu3 and MDCK cells cluster on the high- or the low-efficiency ends of this plot, respectively. However, we also identified exceptions. Namely, PR8 infections are more efficient in A549 and especially Caco2 cells than infections by other strains tested in A549 and Caco2 cells, and those points instead cluster with the more efficient Calu3 infections. We have made sure that we clearly acknowledged the cell-dependence of shapes in this revision, see lines 96-99, 109-110, 208 but conclude that the cell conditions contribute to shape by either providing more permissive or less permissive conditions to IAV more generally or to certain strains specifically.

What is the explanation for the lower increase in filaments when using entry inhibitors compared to other treatments?

See our next response to a related question.

174-177 Why is the post effect stronger than the pre effect?

We think that the reason for the stronger effect of antibody POST treatments is their second, attenuation-independent effect. However, in response to other Referee#1 comments asking us to tone down these conclusions, we refrained from further commenting on this here. This biophysical effect was thus only briefly introduced in this study with all its currently possible interpretations and will be dissected in more detail in a future manuscript.

179- Is this really accurate? When there's no inhibition, there's hardly any change, only significant minor shifts. It might help to align all Y axes for easier comparison and acknowledge the actual ranges of change.

We toned down the statements about the attenuation-independent antibody effects on shape. In this example, we deleted the sentence stating that M2 antibodies induced shape effects without an effect on yield. Since these shape changes were small, we cannot exclude very weak effects on yield that fall within a measurement uncertainty. Our strongest evidence for yield independence are shape effects that are uncoupled from the magnitude of yield changes (e.g. Fig. 4, POST treatments targeting different viral epitopes), and transient treatments (e.g. Fig. 5A, and now Extended Data Fig. 9E). However, we agree that this discussion might be distracting from our other key conclusions, so we toned it down here, and will explore it in depth in a future study.

227-228 The same (even stronger) effects were seen with Baloxavir. This supports the idea that the main effect is related to efficiency. It might be interesting to study Baloxavir kinetics.

Thank you for this suggestion. The magnitude of Baloxavir effects was stronger than that of an attachment inhibitor (SbH36-26 antibody) and the same as that of a fusion inhibitor (MEDI8852 antibody) present during entry (PRE) (Fig. 3E). However, those effects were weaker than the effects of HA base binders (such as MEDI8852 antibody) in POST (Fig. 4, second column), which were discussed in the selected lines. While it will be interesting to explore the kinetics of Baloxavir effects in a future work, this experiment is unlikely to yield new information toward our current conclusions. For example, we expect that shape effects of Baloxavir treatment are directly linked to its inhibitory activity (see the results with different concentrations of MEDI8852 in PRE, Fig. 4, second column), and adding it in short pulses might reduce its overall effects resulting in weaker shape effects. Furthermore, adding Baloxavir later in infection would greatly reduce its expected effects on infection because majority of RNA amplification was complete within 4hrs of infection (Extended Data Fig. 3E). This suggested experiment would thus not be expected to add new information to an already extensive panel of inhibitor treatments performed in this study.

252 I would weaken this statement. As described, shape depends on cell type and strain but is also variable under certain conditions.

We have reworded this statement to more clearly summarize our key findings, lines 206-207.

Previous: This work reveals that influenza A virion shape is dynamic rather than a fixed property of a given strain (Fig. 6).

Revised: This work reveals that influenza A virion assembly is dynamic and rapidly tuned to the infection environment rather than a fixed property of a given strain as commonly assumed (Fig. 6A).

287-288 his again shows that anything enhancing or weakening efficiency results in more spheres or filaments, respectively.

Thank you for pointing out that the effects of osmotic treatment could also be acting via assembly attenuation. We have revised the conclusions to Fig. 5 to include this possibility, lines 201-202 (“While from these results alone we cannot distinguish whether shape changes due to membrane tension, media salinity, or

effects on virus yield...”), and instead focusing our conclusions on rapid and reversible effects of such treatment, lines 202-204 (“these experiments confirm that shape changes can occur quickly and reversibly...”).

Do the resulting filaments escape antibody neutralization more effectively? Can this be tested using antibody titration on the resulting virions?

We have previously demonstrated the advantage of filamentous virions to antibody neutralization (Li et al., 2021, *Nature Microbiology*; <https://doi.org/10.1038/s41564-021-00877-0>). For this work we included a series of controls demonstrating that the antibody-induced filaments are authentic virions: they retained their infectivity (Extended Data Fig. 9A), had comparable genome content to untreated virus (Extended Data Fig. 9B), and were not antibody-induced clumps of spheres (Extended Data Fig. 9C-D).

Have you tried enhancing efficiency with something like nucleotides, growth factors, or FCS? If so, does this shift the virus toward more spherical shapes? Temperature differences might also affect shape outcomes.

We demonstrated that permissive cell conditions yield more spheres (Fig. 2G), increasing MOI in some instances induces spherical virions (Fig. 1D, compare PR8/MDCK shape curves with or without trypsin; Fig. 2E), and hypertonic osmotic media induces spherical virions (Fig. 5E). While we could explore other conditions that increase the efficiency of infection, undertaking these experiments will delay our publication until such conditions are identified but is unlikely to yield new information not already included. For example, FCS is inhibitory for IAV infection and is expected to reduce spheres. IAV infections are routinely performed at 34C as they are less efficient at 37C, and especially above 37C under fever-like conditions. Changing temperature is thus also expected to reduce spheres.

Have you confirmed how the filaments resulting from antibody treatment look via TEM?

We have used TEM to validate our flow virometry approach for measuring virion shapes (Fig. 1 and Extended Data Fig. 1). We then went on to apply flow virometry to attenuating conditions where EM is either not practical or not feasible. From the fluorescence and VSSC distributions of spherical and filamentous virion subpopulations either in the absence or presence of antibodies, we observe no qualitative changes in the peak shapes except for the signal redistribution from

spherical toward the filamentous peaks, and from shorter filaments toward longer filaments, suggesting that filaments become more abundant and larger under attenuation.